

# Continuous and high precision atmospheric concentration measurements of COS, CO₂, CO and H₂O using a quantum cascade laser spectrometer (QCLS)

Linda M.J. Kooijmans[1], Nelly A.M. Uitslag[1], Mark S. Zahniser[2], David D. Nelson[2], Stephen. A. Montzka[3], Huilin Chen[1,4]

[1]Centre for Isotope Research (CIO), University of Groningen, Groningen, The Netherlands
[2]Aerodyne Research Inc., MA, USA
[3]NOAA Earth System Research Laboratory, Boulder, Colorado, USA
[4]Cooperative Institute for Research in Environmental Sciences (CIRES), University of Colorado, Boulder, CO, USA

*Correspondence to*: Huilin Chen (Huilin.Chen@rug.nl)

**Abstract.** Carbonyl sulfide (COS) has been suggested as a useful tracer for Gross Primary Production as it is taken up by plants in a similar way as $CO_2$. To explore and verify the application of this novel tracer, it is highly desired to develop the ability to perform continuous and high precision in situ atmospheric measurements of COS and $CO_2$. In this study we have tested a quantum cascade laser spectrometer (QCLS) for its suitability to obtain accurate and high precision measurements of COS and $CO_2$. The instrument is capable of simultaneously measuring COS, $CO_2$, CO, and $H_2O$ after including a weak CO absorption line in the extended wavelength range. An optimal background and calibration strategy was developed based on laboratory tests to ensure accurate field measurements. We have derived water vapor correction factors based on a set of laboratory experiments, and found that line interference with $H_2O$ dominates over the dilution effect for COS. This interference can be solved mathematically by fitting the COS spectral line separately from the $H_2O$ spectral line. Furthermore, we improved the temperature stability of the QCLS by isolating it in an enclosed box and actively cooling its electronics with the same thermoelectric chiller used to cool the laser. The QCLS was deployed at the Lutjewad atmospheric monitoring station (60 m, 6°21'E, 53°24'N, 1 m a.s.l.) in the Netherlands from July 2014 to April 2015. The measurements of an independent calibration standard showed a mean difference with the assigned cylinder value within 3.3 ppt COS, 0.05 ppm for $CO_2$ and 1.7 ppb for CO over a period of 35 days. The different contributions to uncertainty in measurements of COS, $CO_2$ and CO were summarized and the overall uncertainty was determined to be 7.1 ppt for COS, 0.22 ppm for $CO_2$ and 3.4 ppb for CO for one second data. The comparison of in situ QCLS measurements with measurements from flasks and a cavity ring-down spectrometer showed a difference of -3.5 ± 8.6 ppt for COS, 0.12 ± 0.77 ppm for $CO_2$ and -0.9 ± 3.8 ppb for CO.





## 1. Introduction

Carbonyl Sulfide (COS) has been suggested as a potential tracer for photosynthetic $CO_2$ uptake (Sandoval-Soto et al., 2005; Montzka et al., 2007; Campbell et al., 2008; Berry et al., 2013, Asaf et al., 2013), as it follows the same uptake pathway into plants through stomata as $CO_2$, but is not generally re-emitted by plants (Protoschill-Krebs and Kesselmeier, 1992; Protoschill-Krebs et al., 1996; Stimler et al. 2010b). COS therefore provides a means to partition Net Ecosystem Exchange into Gross Primary Production (GPP) and Respiration. As large uncertainties in the COS budget remain, field measurements of COS and $CO_2$ concentrations and fluxes from leaf- to ecosystem- and regional scale are required for the COS tracer method to be tested and validated (Wohlfahrt et al., 2012; Berkelhammer et al., 2014). Therefore, there is a need for high frequency and high precision measurements techniques of COS and $CO_2$.

Several past studies on COS have relied on discrete (flask) samples analyzed with Gas Chromatographic Mass Spectrometry (GC-MS; Montzka et al., 2007; Stimler et al., 2010b; Belviso et al., 2013). For example, the global atmospheric flask sampling network described by Montzka et al. (2007) has allowed a foundation for understanding COS concentrations over annual cycles on global scale. However, this technique does not typically allow for continuous high frequency measurements and is less appropriate for providing insight in, for example, diurnal variations of local fluxes from soil and plants. Recent developments of Quantum Cascade Laser spectrometers (QCLS) have enabled *in situ* trace gas measurements including COS. These instruments have proven to be a valuable tool for continuous high frequency measurements of COS and $CO_2$ up to a frequency of 10 Hz (Stimler et al., 2010a; 2010b; Asaf et al., 2013; Commane et al., 2013; Berkelhammer et al., 2014; Maseyk et al., 2014; Commane et al., 2015).

The required measurement precision (in this study we define precision as the standard deviation over a two minute period) for studies of exchange processes of COS and $CO_2$ between biosphere and atmosphere depend on the concentration change that these gases undergo in any given experiment. On the regional scale, COS shows seasonal variations typically between ~100 to 150 ppt at continental sites in the Northern Hemisphere (NH), and 40 to 70 ppt in the Southern Hemisphere (SH) and at marine sites (Montzka et al., 2007). $CO_2$ seasonal variations typically reach up to 15 ppm on the NH and are as low as 2 ppm at South Pole (Zhao and Zeng, 2014). For the leaf scale, COS and $CO_2$ concentration changes can be substantially larger; for example, Berkelhammer et al. (2014) showed that during branch bag measurements COS generally decreased by 180 to 240 ppt during active photosynthesis and $CO_2$ concentrations can easily change by 200 ppm, depending on the setup. In this study we will focus on atmospheric measurements of COS and $CO_2$, with the potential to constrain gross carbon fluxes. If we were to infer the gross fluxes to match the atmospheric measurements of $CO_2$ better than 1 ppm, and given the leaf-scale relative uptake ratio of COS/$CO_2$ 1.5 – 4.0 (Stimler et al., 2010b; Seibt et al., 2010; Berkelhammer et al., 2014), our goal would be to have measurement precisions of COS better than 0.3-0.8%, i.e. 1.5 – 4.0 ppt for COS at the ambient level of 500 ppt.





Measurement instruments for long-term atmospheric trace gas concentration monitoring need to meet different requirements than, for example, eddy-covariance measurements. The eddy-covariance technique requires high frequency data (>10 Hz), which limits the precision of the measurements compared to 1 Hz data, and need an averaging period of about 10 to 30 minutes. In contrast to the high frequency required for eddy-covariance measurements, lower frequency measurements (1 Hz) provide useful results over extended measurements periods, which enhances the precision of any individual measurement. Furthermore, measurements for long-term monitoring do not require fast response, thus it is not necessary to operate the instrument at high flow rates. As a matter of fact, low flow rates are preferred so that calibration cylinders can be used over a longer period. This reduces the additional logistics needed for calibration gases, such as filling, calibration and transportation of the cylinders (Xiang et al., 2014). In this research we developed a robust setup for high precision and long-term monitoring of ambient concentrations of COS, $CO_2$, CO and $H_2O$ at different heights from the Lutjewad monitoring station in Groningen, The Netherlands. To this end we employed a 'QCL Mini Monitor' from Aerodyne Research Inc. (Billerica, MA, USA) that can operate autonomously and requires little operator attention (Stimler et al., 2010a). We designed an optimal strategy for 'zero' air spectral correction and calibration for accurate measurements and we assessed the correction for water vapor interference. We will show the precision and accuracy of the instrument with over half a year of field-data and measurements of calibration standards, and we compare the measurements with other instrumentation and flask measurements.

## 2. Experimental setup

Before the actual deployment of the instrument in the field we performed laboratory tests to assess the accuracy and traceability of the QCLS measurements and to develop procedures for applying corrections as needed. Here we describe the laboratory tests and we give detailed information about the instrumentation and field setup.

### 2.1 Instrumentation

The 'QCL Mini Monitor' that we use is a Tunable Infrared Laser Direct Absorption Spectrometer (TILDAS) using a single continuous-wave quantum cascasde laser (Alpes Lasers), which is cooled with a Peltier element to -19.8°C, and using a single photodiode infrared detector (Teledyne Judson Technologies; McManus et al., 2010). The waste heat from both the laser and detector is removed with a recirculating mixture of water containing 25% ethanol, which is temperature controlled with a thermoelectric chiller, ThermoCube 300 (Solid State Cooling Inc, USA). The instrument was initially set to simultaneously measure COS, $CO_2$ and $H_2O$ at wavenumbers 2050.397, 2050.566 and 2050.638 $cm^{-1}$ respectively. We extended the range of the laser current to include measurements of CO at 2050.854 $cm^{-1}$. Figure 1 shows the simulated transmission spectrum of ambient concentrations of COS, $CO_2$, CO and $H_2O$ as obtained through the HITRAN 2012 database (Rothman et al., 2013). The precision and accuracy of the measurements will be discussed in Sect. 3.1.





The instrument consists of a 0.5 L astigmatic Herriot style multipass absorption cell (McManus et al., 2010) with an effective path length of 76 m. The cell has a temperature between 20 and 24 °C, depending on the room temperature and the temperature setting of the thermoelectric chiller. The cell is kept at a constant pressure of 53.3 hPa (40 Torr) with an inlet valve that is controlled by the TDLWINTEL program (Aerodyne Research Inc., Billerica, MA, USA) based on the measured cell pressure. The same software manages the data acquisition and spectral analysis (Nelson et al., 2004) and calculates dry air mole fractions real-time (1 Hz) through nonlinear least square spectral fits combined with the measured cell temperature and pressure, a constant path length, and the HITRAN 2012 database cross-sections as a function of wavelength. The spectral fit for CO is separated from the fit for COS, $CO_2$ and $H_2O$ as there is slight interaction of the CO peak with a second absorption line of COS. The COS fit close to the CO peak is linked to the COS peak at lower wavenumbers to improve the fitting for CO. This is achieved by fitting the spectra in two steps: first the mole fractions are determined for both COS peaks independently, second the CO concentration is recalculated with the fixed COS concentration derived from the separated COS peak in the first step.

The TDLWINTEL software has the option to store raw spectra. These spectra can later be used for re-analysis using the so called 'Playback' mode of the software. The spectral parameters (line shape and position) for the fits are taken from the HITRAN database (Rothman et al., 2013). The sample spectra are normalized with a 'zero' air spectrum to remove background spectral structures and to remove absorbance external to the multi-pass cell (Stimler et al., 2010a; Santoni et al., 2012). The 'zero' air spectrum is periodically determined when the cell is flushed with high-purity nitrogen (99.99999%), which we will now refer to as 'background' measurement. The nitrogen is first passed over a gas purifier (Gatekeeper, CE-500K-I-4R) to remove CO that is often found in such nitrogen cylinders. The frequency of the laser is locked based on the spectrum measurement of the high strength $CO_2$ line at 2050.566 as shown in Fig. 1. For automatic start-up, a gas sealed in an aluminum reference cell can be flipped into the optical beam. The reference cell was filled with 8 hPa (6 Torr) COS and 27 hPa (20 Torr) CO. Initially, we could use the peak position of COS in the reference cell to determine the frequency of the laser. However, COS did not last longer than a few months in the reference cell so thereafter the laser frequency was locked only based on the peak position of CO, which did not impact the results.

## 2.2 Calibration strategy

To allow comparison of QCLS measurements with other instrumentation and across different sites requires traceability to a primary scale. Laboratory tests were conducted to characterize the response of the instrument against real-air standards from NOAA/ESRL, which were subsequently used to transfer the calibration scale to calibration standards (ambient air high-pressure cylinders). Moreover, we performed tests to understand the frequency required for background and reference measurements to ensure reliable and accurate results.



### 2.2.1 Instrument response

To characterize the response of the instrument for COS, $CO_2$ and CO mole fractions we used six aluminum cylinders that were calibrated by NOAA/ESRL on NOAA-2004 COS scale, WMO-X2007 $CO_2$ scale (Zhao and Tans, 2006) and WMO-X2004 CO scale (Novelli et al., 2003). Five of the six standards were calibrated for $CO_2$, four for CO and two for COS. The

internal surfaces of the COS calibrated cylinders were Aculife-treated. Figure 2 shows that the response curves are linear for $CO_2$ (between 354 and 426 ppm) and CO (between 94 and 384 ppb). Furthermore, the residuals of these curves do not show signs of non-linearity. For COS conclusions about the linearity of the response are not possible given that we had only two calibrated cylinders. As the response is linear for $CO_2$ and CO, we assume that the response is linear for COS as well. The NOAA-calibrated cylinders cover a wide range of $CO_2$ and CO concentrations and response curves that were determined for

these gases were consistent when the experiment was repeated. The factor 1.038 to transfer the QCLS $CO_2$ measurements to the WMO scale is slightly different from the factor 1.05 found by Commane et al., 2013. For COS the concentration range of the NOAA/ESRL standards was smaller, between 447.8 and 486.6 ppt, and combined with the lower precision of these measurements, the response curve has large uncertainty. Therefore, repeating the experiment resulted in varying response curves, with an average slope equal to 0.99 and the minimum and maximum slope equal to 0.87 and 1.04 respectively. The

response curve shown in Fig. 2 is the curve where the two NOAA/ESRL calibrated cylinder measurements showed the lowest standard deviations: 2.7 and 3.0 ppt, with the average standard deviation over different experiments equal to 4.4 ppt. In the next section we will discuss different calibration methods to deal with the higher uncertainty of the COS response curve. Furthermore, we tested the stability of the response of COS, $CO_2$ and CO with hourly measurements of three cylinders over a period of 35 days during field measurements. The results of this will be shown in Sect. 3.1.

### 2.2.2 Calibration standards

In this study we use calibration standards, either to correct for instrument drift during field measurements and scale to the NOAA or WMO scales with the so called reference standards, or to assess the accuracy of the instrument with so called target standards. The calibration standards used in this study are high-pressure aluminum uncoated gas cylinders (Luxfer, max. 200 bar) filled with ambient air using an oil free air compresser (RIX SA-3) and are used in combination with two-

stage regulators (Scott Specialty Gases, model 14). Using the linear regression curve that we found in Fig. 2, we determined mole fractions in these calibration standards by considering response curves derived from measurements of the NOAA/ESRL standards. Results of calibrations of three cylinders with the QCLS are shown in Table 1 for $CO_2$ and CO. We calibrated the same cylinders with a cavity ring-down spectrometer (CRDS; Picarro Inc. model G2401) using the same standards linked to the WMO scale. Calibrations with the QCLS for $CO_2$ and CO agree with calibrations from the CRDS within their uncertainties, which gives confidence in our calibration method. For COS we could not compare our calibrations

with that from other instrumentation.



As we saw in Sect. 2.2.1, the response curve of COS is difficult to determine due to the lower instrument precision and the narrow COS concentration range of the NOAA/ESRL standards. These factors introduce uncertainty in the calibration of the standards, especially for those having COS mole fractions outside of this range. Besides that, the number of available calibrated cylinders used to transfer the scales to other cylinders may be limited in labs, especially for COS, as this gas is usually not one of the standard measured species. The question is therefore what a suitable method is to transfer the scale to the calibration standards for COS. To test this, we re-analyzed calibration measurements in different ways: (A) with response from the two NOAA/ESRL standards and the curve forced through a zero-point, (B) with two NOAA/ESRL standards and the curve *not* forced through zero, and (C) with a single bias correction using a NOAA/ESRL standard that has the concentration closest to the calibration standard.

Table 2 shows the assigned values on the calibration cylinders considering the different approaches. The calibration measurement was repeated three times and results are here shown as the average over the three measurement. The fact that the cylinder values need to be extrapolated causes larger deviations compared to when the curve is forced through a zero point (see e.g. cylinder #1 in Table 2). Therefore, when calibration standards have concentrations outside the range covered by the NOAA/ESRL standards, method A is preferred because it avoids extrapolation of the calibration curve by using the zero-point. However, as we saw in Sect. 2.2.1 it is difficult to accurately determine a calibration curve due to the lower precision of the measurements and the narrow COS concentration range of the NOAA/ESRL standards, which holds for both method A and B. In this study we therefore calibrate cylinders and field measurements under the assumption that the data follow a calibration curve with a slope equal to 1 (in Sect. 2.2.1 we saw that the average slope of different calibration experiments was equal to 0.99) and we apply a single bias correction, which is applied in Table 2 as method C. Theoretically, methods A and B would give the results closest to reality, here we observe that the difference of method C against methods A and B is small (on average 1.3 ppt) and the result is often in between those of methods A and B. We therefore consider the single bias correction as a good compromise when a calibration curve can not accurately be determined.

## 2.2.3. Background and reference strategy

Background measurements are required to reduce the effect of curvature of the baseline spectra and are typically done every 2, 5 or 30 minutes by other users of similar QCLS analyzers (Stimler et al., 2010a; Commane et al., 2013). When we tested the required frequency of background measurements we found that COS concentrations can shift to another level, either up or down, uncorrelated with instrument drift such as temperature changes, and even when the background measurements were done shortly after each other (every 10 minutes). In a test where we measured cylinder air over a period of five hours, alternated with background measurements every 10 minutes, we found that on average the COS concentration shifts by 5 ppt after every background measurement, but in 32 % of the cases the shift is larger than 10 ppt, with peaks up to 37 ppt. We could not correlate these shifts to changes in instrument temperature or inlet pressure, or the length of the background





measurement itself. As we could not find correlations with any parameter, it is unknown what the reason is for these shifting concentrations after background measurements. To be able to correct for the concentration shifts for accurate concentration measurements requires measurements of reference gases at least once within every background cycle. Therefore, increasing the background frequency automatically leads to larger use of a reference cylinder gas. As we also need to consider cylinder

logistics (filling, calibrating and transportation) we decided to do less frequent background measurements (once every six hours) with reference cylinder measurements in between. Lowering the background frequency to six hours did not negatively affect the measurement precision, but the accuracy of the measurements is affected due to instrument drift over the period in between the background measurements. The frequency of reference cylinder measurements that is needed to correct for the instrument drift depends on the rate with which the instrument temperature changes. To test the frequency of reference

measurements that is needed to remove drift due to temperature changes, we measured cylinder air over a period of 19 hours, alternated with a reference gas every 15 minutes. During this measurement period the electronics temperature changed by 0.1 °C due to changes of the room temperature, where the most rapid change was 0.06 °C in 30 minutes. Without correcting the data, the standard deviation of the minute averaged data is 11.8 ppt for COS, with drift up to 40 ppt. Correcting the data with reference measurements from every 15 minutes lowers the standard deviation to 4.6 ppt, correcting only every 30 and 60

minutes gives a standard deviation of 5.3 and 9.6 ppt respectively. See Table 3 for an overview of the standard deviations of COS, $CO_2$ and CO. Based on these results we can say that reference measurements every 30 minutes are sufficient to remove drift within 5.3 ppt with temperature changes up to at least 0.06 °C per 30 minutes. Field measurements have shown that with corrections from 30 minute reference measurements, drift is still sufficiently removed with temperature changes up to 0.2 °C per 30 minutes. Furthermore, improving the temperature stability of the instrument could reduce the drift such that

reference measurements are needed less frequently. The effect of improved temperature stability was tested during field operation of the QCLS at the Lutjewad station (see Sect. 2.5).

**2.3 Water vapor interference correction**

The concentration measurements of gases determined through light absorption spectrometry can be affected by water vapor in the sample air in two ways: (1) by spectroscopic effects (enhanced pressure broadening or direct spectral interference), which will directly modify the absorption spectrum and (2) through dilution of the sample air, which linearly depends on

water vapor concentrations. The water vapor interference can be prevented by drying the air before measurement. However, this requires adding a drier to the sampling inlet lines and depending on the drier it can require additional maintenance, which is not favorable for unattended autonomous measurements. As the QCLS includes measurements of $H_2O$, water vapor interferences can be accurately accounted for in the calculation of dry air mole fractions of the other gases. We determined the water vapor dependence of mole fractions reported by the instrument (TDLWINTEL software) for COS, $CO_2$ and CO

from laboratory experiments. To do this we measured dry and humidified cylinder air alternately. We humidified the cylinder air with wet silica gel in a filter, giving a water vapor profile from 2.1 % down to 0.2 % $H_2O$ in the sample air when the line was flushed (Rella et al. 2013). The interaction of silica gel with COS, $CO_2$ and CO was tested by drying the air with





magnesium perchlorate or a cryogenic system after the air had passed the wet silica gel. No interaction with COS, $CO_2$ and CO was found for silica gel, magnesium perchlorate and the cryogenic system: the average difference between the unaffected and humidified-dried silica gel air was maximum $1.0\pm1.9$ ppt for COS, $0.11\pm0.13$ ppm for $CO_2$ and $0.3\pm0.4$ ppb for CO. Furthermore, we found that a 0.3 nm molecular sieve, which is commonly used to remove water vapor, removes all

COS from cylinder air. The dry air cylinder measurements were used as a reference during the experiment to account for instrumental drift. Figure 3 shows the mole fractions of the humidified air measurements during the experiment with the TDLWINTEL water vapor correction turned off. Figure 4 shows how the wet/dry ratios of the COS, $CO_2$ and CO concentrations relate to $H_2O$ when the TDLWINTEL water correction was turned off (blue). For this figure we combined three water vapor tests, of which one of the three was done with the TDLWINTEL water correction turned off, the other two

had the TDLWINTEL water correction turned on. We ran the Playback mode of the TDLWINTEL software to get the data of these water vapor experiments for the case that the correction was turned off, when the real time data were obtained with the correction on. For CO only two of the three experiments are used in the analysis because the third showed larger scatter (colored in gray), indicating instability of the QCLS. Figure 4 shows that the COS, $CO_2$ and CO wet/dry ratios are all linearly dependent on $H_2O$. When wet air mole fractions are measured (that is, the TDLWINTEL water vapor correction is

turned off), these curves could act as a water vapor correction factor to obtain dry air mole fractions. The effect of $H_2O$ on the species is +2.9 % for COS, -1.45 % for $CO_2$, and -0.9 % for CO per % $H_2O$. The uncertainties of these curves at 1.5% $H_2O$ are 3.5 ppt for COS (at a concentration of 450 ppt), 0.11 ppm for $CO_2$ (at 400 ppm), and 0.9 ppb for CO (at 150 ppb). The fact that $CO_2$ and CO show an inverse correlation with $H_2O$ indicates that these species are primarily affected by the dilution effect of water vapor. To the contrary, COS shows a positive relation with $H_2O$. The reason for this positive

correlation is that the baseline shape of the spectra is distorted by a small $H_2O$ peak on the left or the larger $H_2O$ peak to the right of the large $CO_2$ peak (see Fig. 1). A potential solution to this would be to split the fit between the COS peak and the small $H_2O$ peak, which we will now refer to as "split fit" (in contrast to the "standard fit"). In Fig. 4 the results with the split fit are shown in green. For $CO_2$ and CO the green curve does not differ significantly from the blue curve, as would be expected. For COS the correlation with $H_2O$ is now negative, with an effect of -0.7 % per % $H_2O$. A slope of -1 % would be

expected due only to pure dilution; however, pressure broadening by water vapor affects the mole fractions as well. Within TDLWINTEL the width of the peaks are fixed based on the line shape information from the HITRAN database which is only for air-broadening with $N_2$ and $O_2$. Water broadening can be greater than air broadening. To correct for the pressure broadening effect the software modifies the width of the absorption line through the so called air "broadening coefficients" which are specific for every spectral line of a certain species. The later version of TDLWINTEL can also use the water

broadening effect by increasing the air broadening coefficients from HITRAN. The residual error of the fit, which is caused by the broadened absorption line but not properly adjusted line width, affects the mole fraction and thereby the slope of the curves in Fig. 4. This can explain why the slopes differ for the different species. Besides that, the measured $H_2O$ concentration on itself has an uncertainty, which contributes to the deviation of the slope from -0.01. For the spectral lines that we use the ratio of the water broadening to air broadening coefficients were estimated by the manufacturer to be 1.5, 1.7



and 1.5 for COS, $CO_2$ and CO respectively for application of the standard fit. The slopes that were determined with these TDLWINTEL corrections (not shown here) were equal to +3.0 % for COS, -0.19 % for $CO_2$ and +0.1 % for CO per % $H_2O$ without using the split fit. These results show that, even after correcting for water vapor interference by the TDLWINTEL software, the mole fractions are still water vapor dependent.

Using the Playback mode we tried to find the optimal water broadening coefficients to sufficiently correct for water vapor using TDLWINTEL. We did this for the three different water vapor tests and for both the standard fit and the split fit. For the standard fit we could not find optimized broadening coefficients for COS because turning the TDLWINTEL correction on caused an opposite correction and resulted in larger deviations from the assigned cylinder value due to the effect of the

$H_2O$ peak on the baseline. For the *standard* fit, the optimized broadening coefficients varied between 2.1 and 2.2 for $CO_2$ and between 1.0 and 2.0 for CO for the different experiments. For the *split* fit we did find optimized coefficients for COS between 1.0 and 1.4 for the different experiments, for $CO_2$ and CO the same results were found as for the standard fit. When the different experiments were combined the optimized broadening coefficients are equal to 1.0, 2.15 and 1.0 for COS, $CO_2$ and CO respectively, where the values for $CO_2$ and CO can be used for both the standard and split fit, and the value for COS

is only suitable for the *split* fit. Note that the uncertainty of these optimized broadening coefficients is influenced by the uncertainty of the curves as in Fig. 4 and the variation between the different experiments.

Different water vapor correction strategies can now be considered and are summarized in Table 4. An appropriate direct water correction for COS is not possible with the combination of the standard fit and the TDLWINTEL correction on.

However, a correction curve can still be applied to these data with a curve that is determined with the same broadening coefficients as is used for the original data to be corrected. Here we found that for a broadening coefficient of 1.5 with the standard fit the slope of the curve for COS is equal to 0.030. We continued to test the performance of the water vapor correction based on the standard fit and the TDLWINTEL correction off. We applied the correction curves to field measurements over a period of 35 days in March and April 2015. In Sect. 3.3 we will compare the dry air mole fractions

with measurements of a collocated CRDS for $CO_2$, CO and $H_2O$ and with dry air flask sample measurements for COS.

**2.4 Flow schematics for measurements at the Lutjewad station**

In July 2014 we deployed the QCLS in the field for measurements at the Lutjewad monitoring station in the Netherlands (60 m, 6°21'E, 53°24'N, 1 m a.s.l). We use three diaphragm pumps (KNF N-86) to keep the inlet lines well flushed between the inlet at the tower and the laboratory where the analyzer is positioned (up to 60 m length in the tower and 30 meter from the

tower to the lab-building) with a flow of 2 L min$^{-1}$. Just in front of the pumps we split the line with a tee junction to get a subsample from the tower lines. These subsamples are pulled through the sample cell with an oil-free dry scroll pump (Varian TriScroll) downstream of the cell of the QCLS. 1/2'' Synflex (Decabon) tubing is used in the tower and in front of the KNF pumps. We have tested the interaction of Teflon and Synflex with COS: no significant differences in COS mole



fractions were found when the air flow was alternately passed over stainless steel and Teflon (0.9±1.9 ppt) or Synflex (0.7±2.6 ppt). We did observe COS production from the KNF N-86 pumps, however, this did not cause problems for the measurement sampling as these pumps were placed on the bypass lines. We use a 5.0 μm Teflon filter at the inlet of the tower sample lines to prevent dust, sand and salt to come into the lines. In front of the analyzer we use another 0.5 μm

stainless steel filter (Swagelok) to prevent pollution of the sample cell. A check valve is placed between the analyzer and the vacuum pump to prevent unfiltered room air to enter the cell in case the vacuum pump suddenly stops. A needle valve was placed between the cell and the vacuum pump to control the flow of the system at 0.16 L min$^{-1}$, resulting in a 90% response time of ~15 s. We use a multi-position Valco valve (VICI; Valco Instruments Co. Inc.) to switch between sample lines from different heights in the measurement tower and to cylinder gases. The Valco valve is controlled by the TDLWINTEL

software in which one can set the sequence and duration of the valve ports to be measured. An interval is set for automatic hourly repetition of the sequence. Until January 2015 we used a solenoid valve (Parker) to switch between dry nitrogen and cylinder or ambient air. However, we found that this solenoid valve was leaking, and was thereby diluting the cylinder and ambient air measurements with dry nitrogen. On January 7, 2015 we changed the setup such that the Valco valve also controlled switching to the dry nitrogen. The Valco valve, KNF pumps, air purifier and 0.5 μm filter are built into a 19'' rack

with 1/8'' stainless steel tubing.

For our setup, every hour starts with a measurement from a reference and target standard (3 minutes each). Subsequently, the system alternates between three measurement heights where every height is measured for 8 minutes, meaning that every height is measured two times in an hour. The reference standard was measured every half hour to remove instrument drift. In

March 2015 we measured an extra reference gas once every hour, which was used to test the stability of the instrument response over a period of 35 days (the results will be shown in Sect. 3.1). Background measurements were done every six hours with dry nitrogen over 60 seconds. Before the actual background measurement is done, the cell is first flushed for 2.5 minutes to make sure that water vapor is removed from the cell by 99%.

### 2.5 Temperature stability

We noticed that under lab conditions, when the temperature is controlled within ~0.2 °C, the precision of the instrument was typically better than with the highly changing temperatures up to 2 °C at the measurement station during the course of the day. These temperature dependencies were also observed with other QCLS analyzers in Xiang et al. (2014) and Berkelhammer et al. (2014). Before we made any modifications, the electronics temperature ($T_{electr}$) varied with 0.91 °C with every degree of changing room temperature ($T_{room}$), the cell temperature ($T_{cell}$) varied with 0.11 °C/°C. Xiang et al. (2014)

showed the potential to improve the temperature stability with an active temperature control using an Oasis 3 chiller (Solid State Cooling Inc, USA) of which the set point can be controlled with the TDLWINTEL software. They improved the $T_{cell}$ variability of 0.03 °C/°C without active temperature control to 0.0005 °C/°C with active control. We improved the temperature stability of the instrument with only the ThermoCube chiller, for which we extended the cooling loop to a heat





exchanger attached to the fan of the electronics section. This way, the air going into the electronics section for ventilation is actively cooled. Moreover, we have put the analyzer in an enclosed box to add an extra layer of temperature isolation. With these modifications the temperature variability of $T_{electr}$ improved to 0.21 °C/°C, and to 0.07 °C/°C for $T_{cell}$.

## 2.6 Flask analysis of COS

Besides *in situ* measurements, flask or canister measurements can provide a valuable tool for providing information about ambient concentrations of COS as well (e.g. Montzka et al., 2004; Montzka et al., 2007; White et al., 2010; Blonquist et al., 2011). We filled flasks from cylinders so that we could analyse these flasks on the QCLS and the GC-MS at NOAA/ESRL (Montzka et al., 2007). This allowed us to assess the accuracy of the flask measurements and check the methodology for assigning values to our calibration cylinders. Four pairs of glass flasks with a volume of 2.5 L were filled to 2.5 bar with dry

air from two of our calibration standards and the two NOAA/ESRL standards. The calibration standards were calibrated for COS with the NOAA/ESRL standards as described in Sect. 2.2.2. The four cylinders contained COS concentrations between 447.8 and 486.6 ppt. The flask measurement setup using the QCLS is similar to that of routine tower measurements but with a few modifications. A pressure sensor was added to monitor the pressure in the flasks. Furthermore, a diaphragm pump with a shut-off valve was used to remove residual air from previous flask measurements in the connecting tube, and to test the

lines for leaks. In Fig. 5 the placement of the pressure sensor and diaphragm pump is indicated; they are separated from the field measurement setup as it is used for flask measurements only. Reference and target gases were introduced before measuring the actual flask sample to calibrate the measurements to the NOAA scale. We did not use measurements of a reference gas after the flask measurement as the stability of the measurements is affected by the larger pressure difference between the flask and the reference cylinder. Two measurements of 2 minutes each were done on each flask and the results

were averaged to derive the final value. An overview of the measurements of the QCLS and GC-MS is given in Fig. 6 where the flask pair measurements are averaged and are shown as the deviation from the assigned cylinder value. The repeatability within the flask pair is shown by the error bars; note that the comparison also demonstrates the uncertainties associated with the transfer to the NOAA scale (see Table 5). For the first measurement at the QCLS (orange) three of the four flask pairs are within 1.5 ppt of the assigned value and one flask pair deviates by 7 ppt. This is similar to the GC-MS measurements at

NOAA where one of the four flask pairs deviates further from the assigned values. However, this holds for another flask pair than at the QCLS. For the second QCLS measurement (blue) one flask pair has drifted on average by 12 ppt, where other flasks remained stable within 2.5 ppt. It is unclear why the two flasks have drifted as all flasks were filled, measured and stored in the same way; however, we have kept the pair to monitor the potential drift in the future and to find out what has caused the drift. Note also that the consistency of the flask measurements between April 2015 and January 2016 is depending

on the stability of the reference cylinder which was used for all flask measurements. Although we observed that COS can drift in cylinders we did not find indications that the particular reference cylinder used for this analysis drifted over the 9 month period.





We also measured dry air flask samples to test if the water vapor correction that we determined in Sect. 2.3 sufficiently removes the effect of water vapor on calculated mole fractions. Flasks were filled to ambient pressure as part of a standard flask sampling routine at the Lutjewad station (Neubert et al., 2004). The air samples are dried with a cryogenic system prior to collecting. The flasks were stored for maximum 1.5 months before being measured. The same measurement strategy as for

the NOAA/ESRL comparison was used, and for these flasks two measurements of ~1.5 minutes could be done before an inlet pressure of 0.3 bar was reached. We did not observe any dependence of measured dry air mole fractions in air from the flasks with the inlet pressure below ambient. The measurement results will be shown and compared with the *in situ* QCLS measurements in Sect. 3.3.

## 3. Results and Discussion

### 3.1 Precision and accuracy

We assessed the measurement uncertainty and accuracy with the hourly measurements of the reference and the target gases over the period from August 2014 until April 2015. As mentioned previously, each reference and target gas was measured for two minutes. The mean value of the hourly instrument-reported and uncorrected two-minute measurements are shown in Fig. 7. In Fig. 8 the standard deviation of these measurements are shown. Figure 7 also includes the electronics temperature

of the QCLS. The cylinder measurements show that concentrations can drift substantially, i.e. COS concentrations easily vary by 50 to 100 ppt. However, concentration changes are not correlated with temperature, which changed by 13 K throughout the year. Note that the concentration shift on January 7, 2015 (especially visible in $CO_2$) happened after eliminating the leaking solenoid valve that caused mixing of nitrogen into the tubing that delivers the reference and target gases. In October 2014 the span between the two cylinder measurements changed, which is again mostly visible in $CO_2$. The

reason for this change is that the regulator pressure of one of the two cylinders was slightly changed, which affected the amount of dilution of nitrogen into the sample line. Although it is known that COS mole fractions can drift in cylinders over time we did not find indications that the mole fractions drifted within the measurement period.

Figure 8 shows the two-minute standard deviations of the hourly reference gas measurements between August 2014 and

25 April 2015. It is clear from Fig. 8 that the instrument precision cannot be captured with one single value due to its variation. In the right plot of Fig. 8 the histograms of the standard deviations are shown for the periods before and after improvement of the temperature stability with the black/gray colors corresponding to that in the left figure. The data show that for COS and $CO_2$ the period after improving the temperature stability has an improved mean standard deviation compared to the period before the temperature improvement (from 6.6 to 4.8 ppt for COS and from 0.14 to 0.06 ppm for $CO_2$). However,

looking at the stability of the instrument in August and September 2014 there is no consistent relation between temperature stability and instrument precision. We have seen that other factors such as alignment influence the precision as well.



The overall uncertainty of the measurements consists of uncertainties associated with the scale transfer, water vapor corrections and the measurement repeatability. Table 5 summarizes the different uncertainty contributions for measurements of COS, $CO_2$ and CO. Note that the overall uncertainty varies due to the variation of the measurement repeatability, as we discussed in the previous paragraph. Additional to these uncertainties we observed that COS decreased in a few uncoated

aluminum cylinders at a rate of 2 to 3 ppt per month, but we did not observe this for all aluminum cylinders. Also we did not find indications that our calibration standards drifted during field measurements at the Lutjewad station. Furthermore, experience with cylinders over the past 15 years at NOAA indicates that COS in Aculife treated cylinders is typically much more stable than untreated cylinders. A potential method to improve COS calibrations is to calibrate these using a ppb-level standard accurately diluted to a range of desired COS concentrations (LaFranchi et al., 2015). Applying this method could

improve the accuracy of the calibration if the COS concentrations can be accurately and precisely provided over a broad range, thereby allowing for a more accurate determination of instrument response and a calibration curve. Besides that, this method will aid in assessing the stability of calibration standards.

Our COS measurements are reported on the NOAA-2004 scale, and can be compared to the observations from the global

network of NOAA/ESRL (e.g. Montzka et al., 2007). The same holds for $CO_2$ and CO on the WMO-X2007 $CO_2$ scale and WMO-X2004 CO scale. To test the accuracy of the measurements against the NOAA or WMO scale we analyzed the measurements of one/two target standards after application of two corrections: (1) a correction factor as obtained from response curves of $CO_2$ and CO to transfer the data to the WMO scale and (2) a bias correction using a reference standard to remove instrument drift and to calibrate the data with the NOAA scale for COS. For measurements in March and April 2015

we determined hourly response curves from measurements of two calibration standards. For every species we took the cylinders with the outer concentration values such that the response curves span a wide concentration range. To analyze the need to determine hourly response curves, we corrected the data in two ways: (1) with a single bias correction for COS and a fixed response curve for $CO_2$ and CO, i.e. the one that we determined in the laboratory (see Fig. 2), shown in the left plots of Fig. 9, and (2) with changing response curves determined from the hourly cylinder measurements, shown in the right plots of

Fig. 9. After the corrections are applied, the mean offset of the measurements is within 3.3 ppt for COS, 0.05 ppm for $CO_2$ and 1.7 ppb for CO over the period of 35 days. The slightly larger deviation from the CO assigned value for one of the two cylinders with the fixed response correction is because this cylinder has a higher concentration than the reference cylinder (237.2 against 97.8 ppb) and therefore the uncertainty due to the bias correction is larger. The other target cylinder, which has a concentration of 119.1 ppb, is closer to the reference cylinder and only shows a deviation of -0.6 ppb. For $CO_2$ it is

visible that using the fixed response curve gives a bias up to 0.2 ppm in the target measurements, which is not visible when using the hourly response curve. This bias is an effect of the fact that the response did change in the first 10 days of the period, after which it became stable. We could not relate the changing response curve to any parameter such as temperature or pressure. We did notice that the response curve was changing only in the period after the instrument was transported. A potential reason for the change could therefore be that the instrument still had to be stabilized after transportation. We did



not find indications that the response curve for $CO_2$ changed outside the period between March and April. Except for the fact that the hourly response curve corrects for the changing response curve for $CO_2$, the changing response curve does not significantly remove scatter compared to the fixed response curve for $CO_2$ and CO; that is, the standard deviation of target measurements for the fixed response curve are not substantially lower when the hourly response curve was applied. Also the

target measurements are not consistently closer to the assigned values when the hourly response curve was applied. Furthermore, the fact that the use of hourly response curves does not give lower standard deviations for target measurements of COS compared to when the single bias correction is used, indicates again that the response curves can not accurately be determined for this species, which we discussed in Sect. 2.2.1 and 2.2.2 as well. As we have not seen indications that the response curve for $CO_2$ changes outside the period between March and April, we do not see the need to frequently determine

the response curves with multiple cylinder gases. Moreover, if the response would change outside of the period in March and April as well, then the effect only reaches up to 0.2 ppm for $CO_2$. Taking into account logistical reasons (use of cylinder gases) we suggest correcting data with a single bias correction for COS, and with a fixed response curve for $CO_2$ and CO as determined once with NOAA/ESRL standards (see Sect. 2.2.1), together with a single bias correction from a reference cylinder.

## 3.2 Continuous COS, $CO_2$, CO and $H_2O$ observations from Lutjewad

The COS, $CO_2$, CO and $H_2O$ data record obtained at the 60 m level of the Lutjewad tower is presented in Fig. 10. Based on the previous sections we applied the following corrections to the data: (1) Before March 25, 2015, the TDLWINTEL water vapor correction was applied with broadening coefficients 1.5 for COS and CO and 1.7 for $CO_2$. On top of this correction we applied a linear water correction curve for COS, $CO_2$ and CO as obtained with the TDLWINTEL correction *on*. After

March 25, 2015, the TDLWINTEL correction was turned off and we applied the correction curve from Fig. 4 as obtained with the TDLWINTEL correction *off;* (2) the calibration correction curves as obtained in Sect. 2.2.1, Fig. 2, were applied to transfer the data to the WMO scales for $CO_2$ and CO; (3) a bias correction was applied to remove instrument drift and to calibrate the data with the NOAA scale for COS, here we used the same, and single, reference cylinder over the whole measurement period. (4) To correct for the dilution of nitrogen due to a leaking solenoid valve before January 7, 2015, we

determined a dilution factor by comparing $CO_2$ measurements from the QCLS and a CRDS from the same location and height, under the assumption that without dilution the two analyzers measure the same concentrations (see Sect. 3.3). The percentage of dilution was calculated for $CO_2$ and was typically between 0.4 and 4.9 %. These dilution factors were then applied to all species.

The location of the Lutjewad station along the coast of the province of Groningen in the Netherlands allows the measurement of marine background air during northerly winds, and continental air during southerly winds (Van der Laan et al., 2009). Daytime $CO_2$ concentrations are typically correlated with elevated CO concentrations, indicating the influence of local and regional fossil fuel emissions (Van der Laan et al., 2010). Even though the data do not cover a full year cycle, it



can be seen that the seasonal amplitude of $CO_2$ mole fractions is approximately 15 ppm with a minimum around the end of August. The seasonal variation of $CO_2$ for the Lutjewad measurement station is analyzed in detail by Van der Laan-Luijkx et al. (2010) and Van Leeuwen (2015). The COS mole fraction also shows a seasonal cycle of which the peak-to-peak amplitude is estimated to be 96 ppt based on the two-harmonic fit. Kettle et al. (2002) showed that vegetative uptake is the

flux with the largest seasonal cycle on the NH, and Montzka et al. (2007) showed that the seasonal amplitude of COS depends on the degree to which the sampled air is influenced by terrestrial ecosystems. It is therefore likely that the seasonal variation of COS observed at the Lutjewad site is influenced by vegetative uptake. In Fig. 11 we compare COS mole fractions from Lutjewad with that from three other sites as measured from flask samples with a GC-MS by NOAA/ESRL (Montzka et al., 2007). The flask samples cover data between 2000 and 2015 for Wisconsin, United States (LEF) and Mauna

Loa, United States (MLO) and between 2001 and 2015 for Mace Head, Ireland (MHD). These data are an update of those presented in Montzka et al. (2007) (data available at: ftp://ftp.cmdl.noaa.gov/hats/carbonyl_sulfide/). The flask measurements are plotted as function of time of the year. The high altitude MLO site is less directly influenced by terrestrial ecosystems and therefore shows only small seasonal variation, in contrast to the LEF site, which is largely influenced by (forested) continental air. The Lutjewad COS mole fractions are most consistent with measurements from MHD, which can

be expected since both stations are coastal sites and are located at similar latitudes. The seasonal amplitude of COS at MHD and Lutjewad is in between that of LEF and MLO, most likely because both sites are not solely influenced by marine or continental air but by both types of air masses. The COS mole fraction has a minimum in September and October, and is a few weeks later than the minimum of the $CO_2$ mole fraction. Montzka et al. (2007) and Blonquist et al. (2011) also observed a COS minimum later than that of $CO_2$. They reasoned that this difference is due to the fact that at the end of the growing

season COS mole fractions keep decreasing due to vegetative uptake without at the same time having a source of COS, whereas during this time of year respiration is beginning to offset assimilation in determining the ambient $CO_2$ mole fractions.

### 3.3 Measurement comparison

In Fig. 12 we compare the minute averaged QCLS measurements for $CO_2$, CO, and $H_2O$ with those made by a collocated

CRDS. The air samples were taken from the same height, but through a different inlet. The CRDS measurements of $CO_2$, and CO were performed on humid air, and were corrected for water vapor dilution and interference effects based on a set of instrument-specific correction factors determined in the laboratory of the Center for Isotope Research before field deployment (Chen et al., 2010; 2013). The comparison was only made from January 7 onwards because before that period there was the problem of nitrogen leaking into the sample air for which we used the CRDS data to correct for the dilution.

The mean differences (QCLS − CRDS) are -0.12 ± 0.77 ppm for $CO_2$, -0.9 ± 3.8 ppb for CO and -0.01 ± 0.09 % for $H_2O$. For $H_2O$ there were problems with water vapor in the sample lines in March and April, therefore we only calculated the difference for $H_2O$ over the month January. The slope of 0.98 for the correlation of the CRDS and QCLS data of both $CO_2$ and CO indicates that there is no concentration dependent offset. Also no correlation was found between water vapor and the





difference between CRDS and the QCLS $CO_2$ and CO data ($CO_2$: slope = 0.16, $R^2$ = 0.003; CO: slope = 0.48, $R^2$ = $1.9 \cdot 10^{-5}$). These results give confidence in the calibration strategy and water vapor correction presented in this study. For the period up to January 29, the TDLWINTEL water vapor correction was turned on, and on top of this correction we applied the linear water correction curves as obtained with the TDLWINTEL correction turned on. If we would not apply this linear correction

curve on top of the TDLWINTEL correction, then the median difference between the QCLS and CRDS measurements for this period would be -0.59 ppm for $CO_2$ and -1.1 ppb for CO (against -0.02 ppm and -0.9 ppb for specifically this period when the linear correction curve is applied).

The *in situ* COS measurements between March 25 and April 29, 2015 are compared with 11 dry air flask samples measured

by the same QCLS. The time delay related to the transit of air in the inlet between sampling of air for flasks and QCLS measurements is assumed to be the same, as both sampling systems have a flow rate of 2 L min$^{-1}$ and the sample tubing of the systems has the same size. The flasks are flushed for an hour before closing, but because of mixing in the flask we assume that the flask sample represents the last 15 minutes, therefore we average the *in situ* measurements over these 15 minutes. Furthermore, the flask samples have an inlet at 60 m height, but because the *in situ* 60 m measurements only cover

the period between 9 and 15 minutes before flask closure we also include 40 m measurements to cover the last 9 minutes before the flask is closed. The average difference of COS mole fractions between the 40 and 60 m level is 0.7 ± 9.7 ppt over the measurement period in March and April as determined by the QCLS so we do not expect any bias associated with including ongoing results from the 40 m height in the averages for comparisons. Peaks and valleys in COS mole fractions are well covered by both sampling techniques (flasks and *in situ*); for example the peak up to 620 ppt at April 11 is clearly

visible in both the *in situ* and flask measurements. The average difference between the *in situ* and flask measurements (*in situ* − flask) is -3.5 ± 8.6 ppt. For the comparison we neglected one flask sample where the *in situ* COS measurement over 15 minutes showed large variation (standard deviation of 17.5 ppt where the average standard deviation of the other periods is only 4.7 ppt) and thereby introduced an error in the comparison.

## 4. Conclusions

In this study we have tested a QCLS for its suitability to do accurate and high precision measurements of COS, $CO_2$, CO and $H_2O$. First, the instrument response was characterized using calibration standards and to transfer raw data to the NOAA or WMO scale. Unfortunately, the range of mole fractions in the reference and calibration tanks did not allow the COS response to be accurately determined over the entire range of measured mole fractions. Based on an analysis of different calibration methods, however, we concluded that the raw measurements and calibration standards were best calibrated using

a single bias correction for COS. From hourly paired measurements of calibration standards we observed changes in the response curve for $CO_2$ over a period of 10 days after transporting the instrument to the measurement site. However, as we have not seen indications that the response curve for $CO_2$ changed outside this period, and also taking into account logistical





reasons (use of cylinder gases) we suggest calibrating field data with a fixed response curve for $CO_2$ and CO as determined once with primary calibration cylinders. Second, we investigated the needed frequency of background and reference measurements. Based on laboratory tests we have shown that background measurements every six hours with reference measurements every 30 minutes (for removal of instrument drift) are sufficient to keep the standard deviation of cylinder

measurements within 5 ppt for COS, 0.1 ppm for $CO_2$ and 0.3 ppb for CO over a period of 19 hours. We characterized the water vapor dependence of COS, $CO_2$ and CO from laboratory experiments. Based on an assessment of the TDLWINTEL water correction we determined optimal broadening coefficients for the use of the water correction within TDLWINTEL. Besides that, we presented an alternative water correction based on linear dependence of the wet air mole fractions with $H_2O$ concentration. Furthermore, we demonstrate that a small $H_2O$ peak close to the COS peak has caused a water vapor

dependent concentration error that is larger than the direct water vapor dilution effect. This water vapor interference can be minimized by careful adjustments to the software fitting parameters and was virtually eliminated with corrections as demonstrated in Fig. 4.

The QCLS was set up for continuous *in situ* measurements at different heights at the tower of the Lutjewad monitoring

station. Hourly target measurements were used to asses the accuracy and precision of the measurements. After application of a calibration response curve for $CO_2$ and CO, and a single bias correction for removal of instrument drift and to calibrate the COS measurements to the NOAA scale, the target measurements showed a mean difference with the assigned cylinder value within 3.3 ppt for COS, 0.05 ppm for $CO_2$ and 1.7 ppb for CO over a period of 35 days. One-second precisions during reference gas flow were typically 4.3 ppt for COS, 0.04 for $CO_2$ and 0.9 ppb for CO, however, substantial variations in the

instrument precision were observed during the 7 month field campaign. The different uncertainty contributions for measurements of COS, $CO_2$ and CO were summarized and the overall uncertainty was determined to be 7.1 ppt for COS, 0.22 ppm for $CO_2$ and 3.4 ppb for CO. Furthermore, we improved the temperature stability of the QCLS by applying an additional insulation layer that is controlled by the same thermoelectric chiller as the one used for cooling the laser and detector. However, improvement of the temperature stability of the instrument did not show a consistent relation with

instrument precision. QCLS measurements were compared with independent CRDS measurements for $CO_2$ and CO, and with dry-air flask samples at the QCLS for COS, which showed a mean difference of -3.5 $\pm$ 8.6 ppt for COS, 0.12 $\pm$ 0.77 ppm for $CO_2$, -0.9 $\pm$ 3.8 ppb for CO and -0.01 $\pm$ 0.09 % for $H_2O$. The measurement record over the 7 month period was presented and compared with NOAA/ESRL flask measurements for COS at other sites in the northern hemisphere. The peak-to-peak amplitude of COS in ambient air at the Lutjewad monitoring station was estimated to be 97 ppt, which is

comparable to other coastal sites at similar latitudes in the Northern Hemisphere.



## Acknowledgements

We would like to thank B.A.M. Kers, M. de Vries, H.G. Jansen and H.A. Been for their help in preparing the system for installation in the field and for maintenance of the instrumentation in Lutjewad. We thank D. Paul for the valuable discussions and suggestions, H.A. Scheeren for providing the $CO_2$ and CO calibrations of our calibration standards and J.J.
Spriensma for her help in sorting out flask samples. We also acknowledge the preparation of gravimetric standards and working standards at NOAA by B. Hall, and the technical assistance of C. Siso. This work was supported by the Dutch Sector Plan Physics.

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



**Tables**

**Table 1** $CO_2$ and CO calibration values together with 1-$\sigma$ uncertainties of the calibration curves as obtained with the QCLS and a cavity ringdown spectrometer (CRDS) for comparison of calibration results. COS calibration values could not be compared with that from other instrumentation.

| | $CO_2$ [ppm] | | CO [ppb] | |
|---|---|---|---|---|
| | QCLS | CRDS | QCLS | CRDS |
| Cyl. #1 | $412.33 \pm 0.12$ | $412.43 \pm 0.08$ | $97.8 \pm 1.7$ | $97.6 \pm 3.0$ |
| Cyl. #2 | $398.30 \pm 0.12$ | $398.13 \pm 0.08$ | $119.1 \pm 1.7$ | $119.1 \pm 3.0$ |



**Table 2** COS calibration values for our real-air calibration standards as obtained from three different calibration approaches: (A) with response from the two NOAA/ESRL standards and a zero-point, (B) with the two NOAA/ESRL standards and the curve *not* forced through a zero-point, and (C) using a single bias correction. The NOAA/ESRL standards have COS concentrations 447.8 and 486.6 ppt. The calibration measurement was repeated three times; results are shown as the average over the three measurement and uncertainties indicate the standard deviation over the three measurements.

|  | | Cyl #1 | Cyl #2 | Cyl #3 | Cyl #4 |
|---|---|---|---|---|---|
| A. | Through NOAA/ESRL standards and 0 | 393.0 ± 2.2 | 448.8 ± 4.7 | 473.6 ± 1.5 | 504.1 ± 0.7 |
| B. | Through NOAA/ESRL standards and not 0 | 379.2 ± 8.3 | 445.5 ± 4.4 | 474.7 ± 1.9 | 510.8 ± 4.7 |
| C. | Single bias correction | 390.9 ± 2.6 | 445.8 ± 3.9 | 476.6 ± 2.4 | 506.6 ± 2.1 |



**Table 3** Standard deviation over minute averaged data of COS, $CO_2$ and CO over the 19 hour measurement period for different correction frequencies using reference measurements.

|  | uncorrected | 1 hr. corr. | 30 min. corr. | 15 min. corr. |
|---|---|---|---|---|
| COS stdev. [ppt] | 11.8 | 9.6 | 5.3 | 4.6 |
| $CO_2$ stdev. [ppm] | 0.26 | 0.12 | 0.09 | 0.08 |
| CO stdev. [ppb] | 0.88 | 0.70 | 0.33 | 0.34 |





**Table 4** Different water vapor correction strategies based on the software fitting parameters (standard or split fit) and with the TDLWINTEL correction turned on or off. In case the TDLWINTEL correction is turned on the values indicate the broadening coefficient used for the different species, and in the case it is turned off the values indicate the slope of the correction curve as determined in Fig. 4 with $y = slope * H_2O + 1$ with $H_2O$ in percent and $y$ the wet/dry ratio of the gas.

|  | TDLWINTEL | Broadening coefficient or slope | | |
|---|---|---|---|---|
|  | correction on/off | COS | $CO_2$ | CO |
| Standard fit | on | -* | 2.15 | 1.0 |
|  | off | 0.029 | -0.0145 | -0.009 |
| Split fit | on | 1.0 | 2.15 | 1.0 |
|  | off | -0.007 | -0.0148 | -0.008 |

*No broadening coefficient could be derived; however, we found that for a broadening coefficient of 1.5 with the standard fit the slope of the curve for COS is equal to 0.030, which can be applied as an extra correction on top of the TDLWINTEL correction.





**Table 5** Uncertainty contributions and the overall uncertainty for measurements of COS (447.8-486.6 ppt), $CO_2$ (354 – 426 ppm) and CO (94-384 ppb), for the range of $H_2O$ (0-2.1 %).

| Uncertainty contributions | COS [ppt] | $CO_2$ [ppm] | CO [ppb] |
| --- | --- | --- | --- |
| Repeatability of the NOAA or WMO scale | 2.1 | 0.07 | 2.0 |
| Transfer to calibration standards (1-σ) | 2.8 | 0.12 | 1.7 |
| Calibration using calibration standards* | 2.8 | 0.12 | 1.7 |
| Water vapor correction (1-σ) | 3.5 | 0.11 | 0.9 |
| Measurement repeatability (1-sec.)** | 4.3 | 0.04 | 0.9 |
| *Overall uncertainty* | *7.1* | *0.22* | *3.4* |

*Using the single bias correction (see Sect. 2.2.2) it is the same as transferring the scale to the calibration standards.

**The 50% percentile of the short-term repeatability of calibration standard measurements in March-April 2015 (Fig 8.).




**Figures**

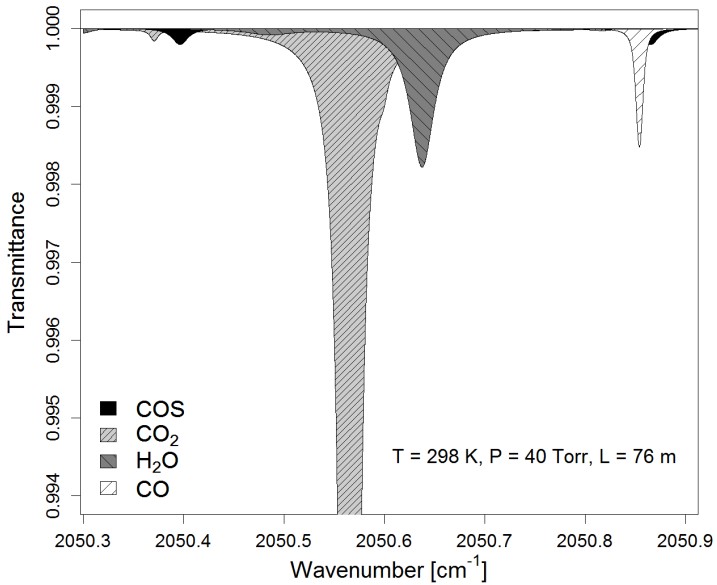

**Figure 1** Simulated transmission spectrum of ambient concentrations of COS (500 ppt), $CO_2$ (500 ppm), $H_2O$ (1.5 %) and
CO (200 ppb) with sample cell conditions: temperature 298 K, pressure 53.3 hPa (40 Torr) and the absorption path length 76
m.

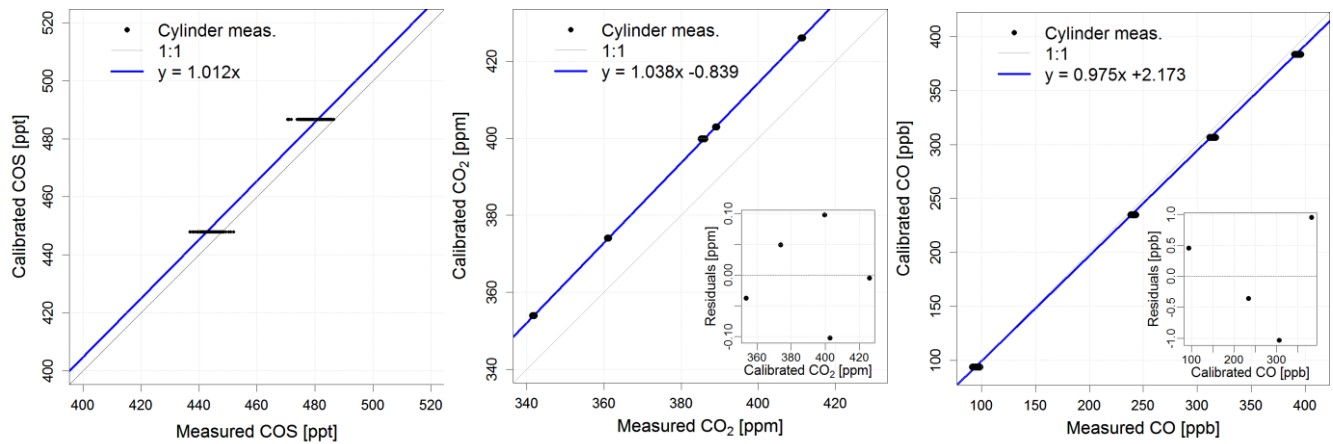

**Figure 2** Response curves determined with NOAA/ESRL standards including the residuals of the fit. The NOAA/ESRL
standards are calibrated on NOAA-2004 COS scale, WMO-X2007 $CO_2$ scale (Zhao and Tans, 2006) and WMO-2004 CO
scale (Novelli et al., 2003). The measured mole fractions shown on the *x* axes are mole fractions calculated through the
TDLWINTEL software but corrected for drift using a reference cylinder.



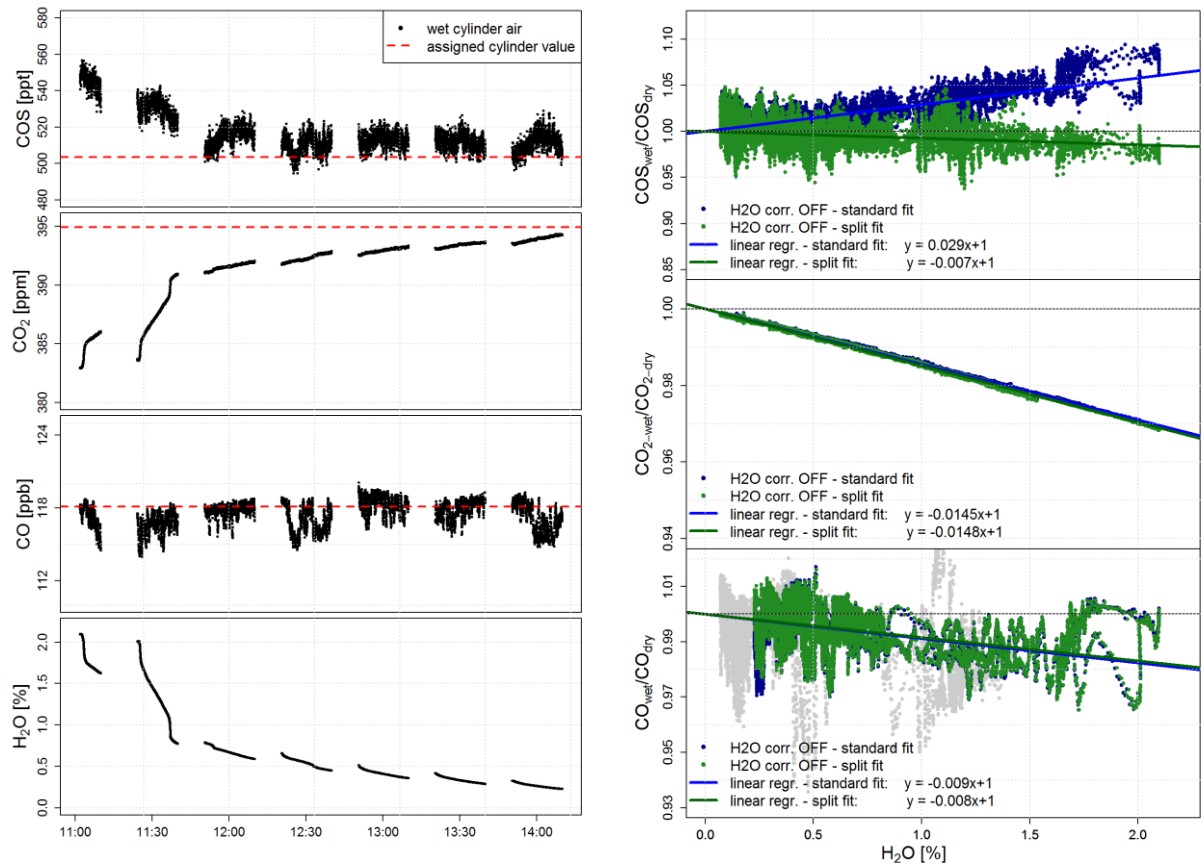

**Figure 3** (left) Water vapor experiment where cylinder air was humidified with wet silica gel and no water vapour correction was applied to the data.

5  **Figure 4** (right) Wet over dry ratio of COS, $CO_2$ and CO versus $H_2O$ as obtained from multiple water vapor experiments with humidified cylinder air. The data show the linear dependence with $H_2O$ when the TDLWINTEL water vapor correction was turned off with the standard fit (blue) and with the split fit (green).



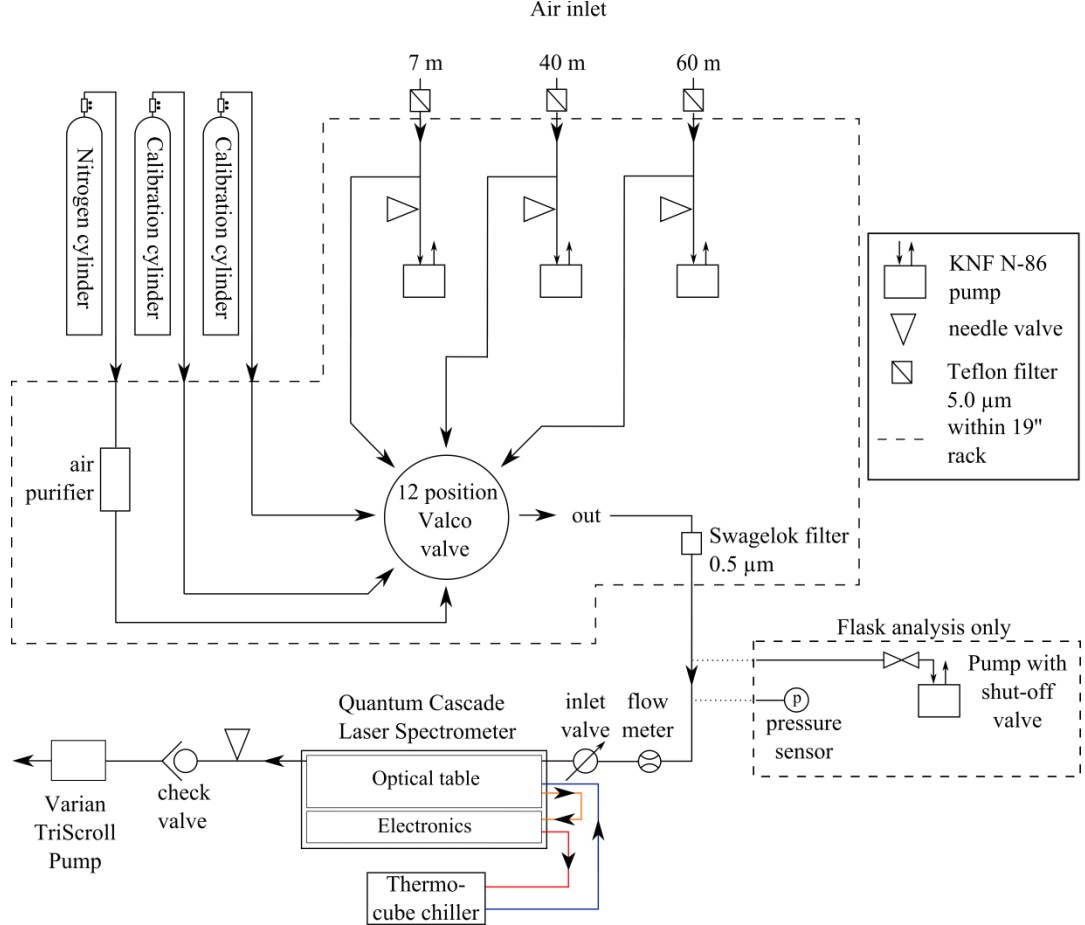

**Figure 5** Schematic overview of the instrument setup for tower profile measurements at the Lutjewad monitoring station. The pressure sensor and pump with shut-off valve were added for flask measurements only (see Sect. 2.6).





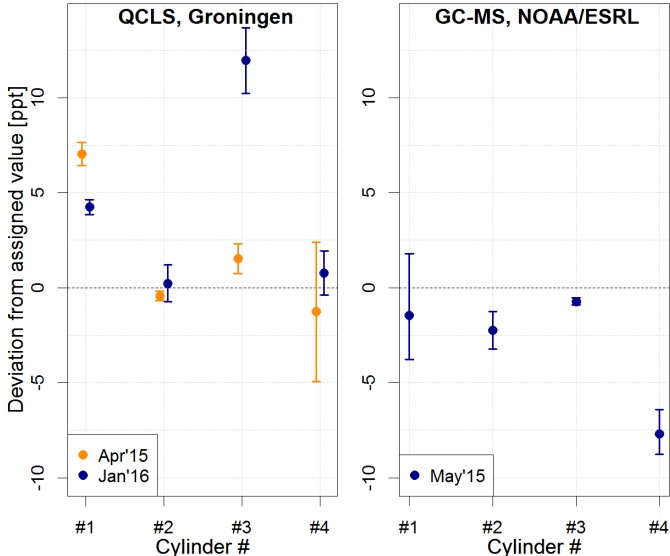

**Figure 6** Flask sample measurements at the QCLS in Groningen (left) and GC-MS by NOAA/ESRL (right). Four paired flasks were filled with dry air from the two NOAA/ESRL standards and two calibration standards, which were calibrated as in Sect. 2.2.2. The flask pair measurements are averaged and shown as the deviation from the assigned cylinder value (447.8 (#1), 486.6 (#2), 455.5 (#3), 467.6 (#4) ppt COS), the error bars show the repeatability. For the QCLS two measurements were done: in April 2015 (orange) and in January 2016 (blue). The GC-MS measurement at NOAA/ESRL was performed in May 2015.





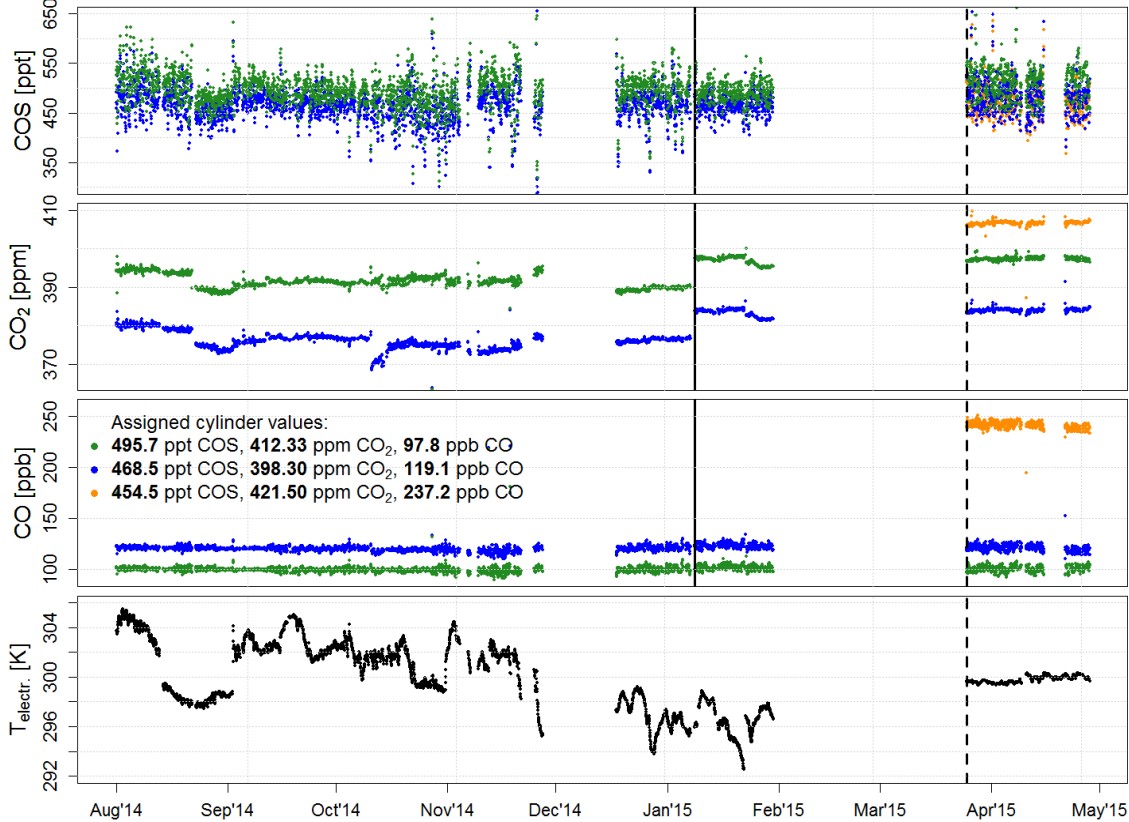

**Figure 7** Mean concentrations of hourly measurements of calibration standards conducted while the instrument was in the field from August 2014 until April 2015 together with electronics temperature. The solid vertical line at January 7, 2015 indicates the moment where we changed the setup with a solenoid valve to a Valco valve for switching to nitrogen, as the solenoid valve was found leaking. The dashed vertical line at March 25, 2015 indicates the moment where we improved the temperature stability by actively cooling the electronics section and putting the analyzer in an enclosed box, which resulted in substantially smaller temperature fluctuations than before. From this moment onwards also an extra cylinder was measured every hour to ascertain if the instrument response was stable over a period of 35 days (orange). The gap in the data record in December 2014 and February-March 2015 is because of tests with the QCLS in the laboratory.





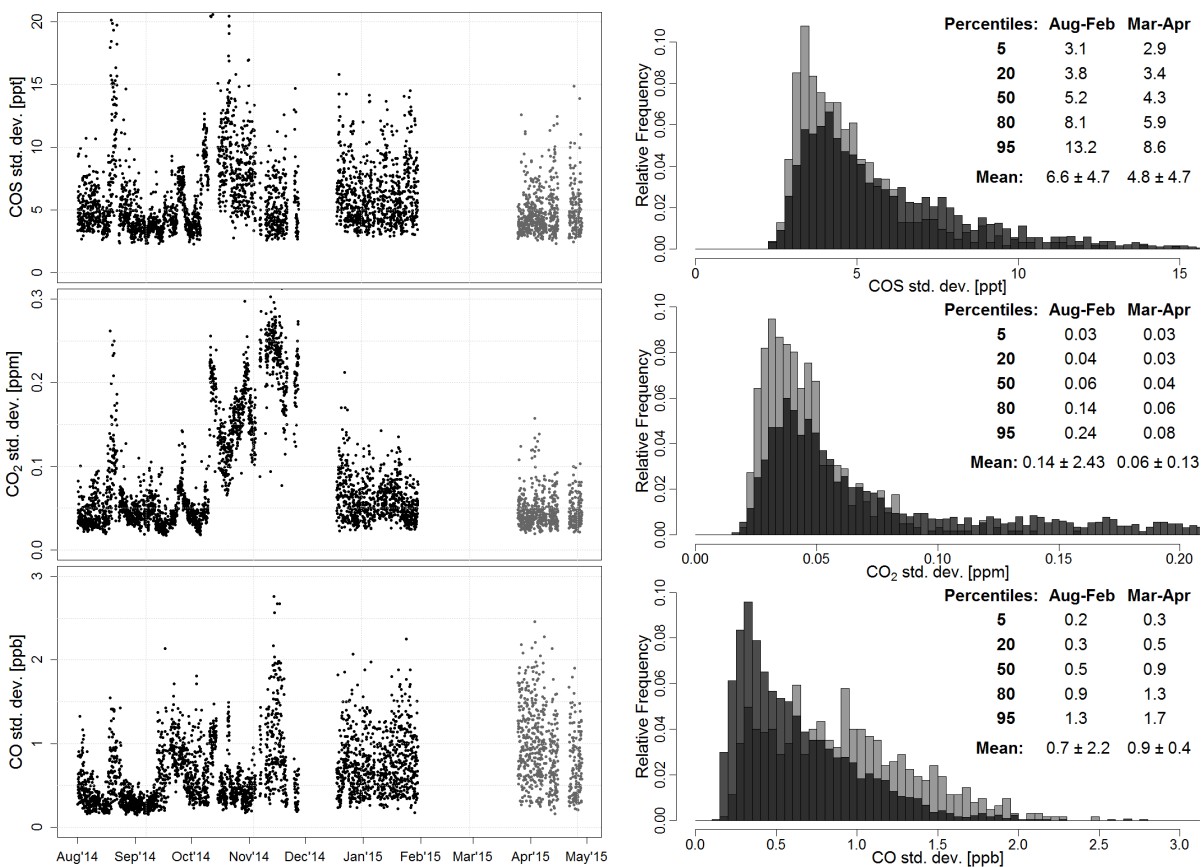

**Figure 8** Left: Standard deviation over two minutes of hourly measurements of one of the calibration standards as measured from August 2014 until April 2015. The gray (black) colored data indicate the period after (before) modifications were made to improve the temperature stability. Right: Histogram of standard deviations as shown in the left figure with the black/gray colors corresponding to the colors in the left figure. Note that the dark histogram is transparent. Also included are the mean standard deviations as well as an overview of percentiles because the data do not show a Gaussian distribution.




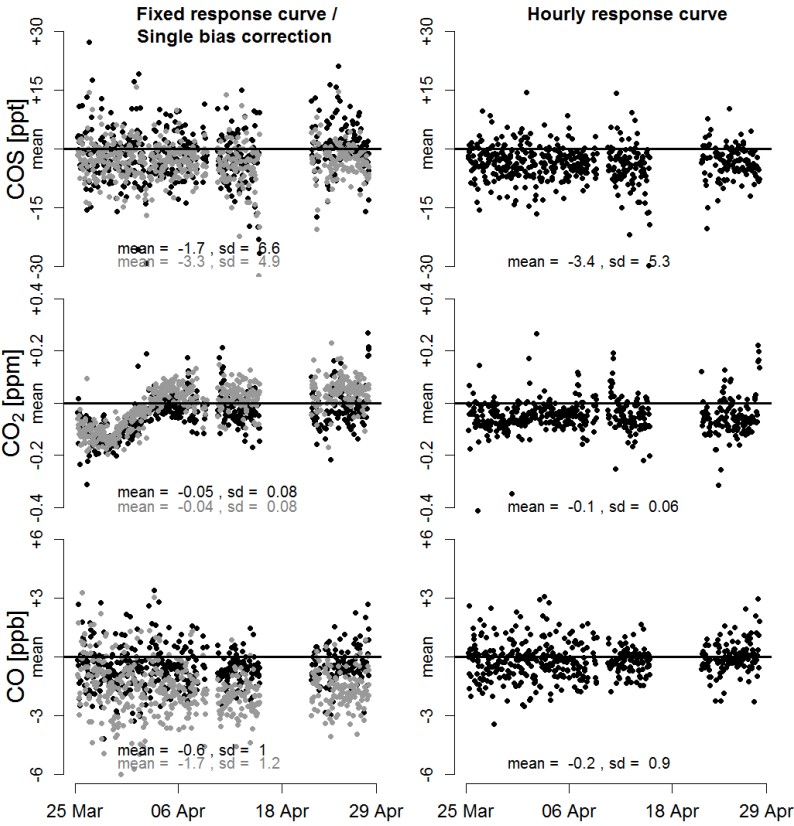

**Figure 9** Mole fraction offsets in target standards after application of (left) corrections with a fixed response curve for $CO_2$ and CO that was determined in the laboratory (see Sect. 2.2.1), and with a single bias correction for COS, and (right) corrections with changing response curves determined from the hourly measurements of calibration standards. For the fixed response curve measurements of two target cylinders are shown, for the hourly response curve only one is shown, as two of the three cylinders were needed to determine the response curve.





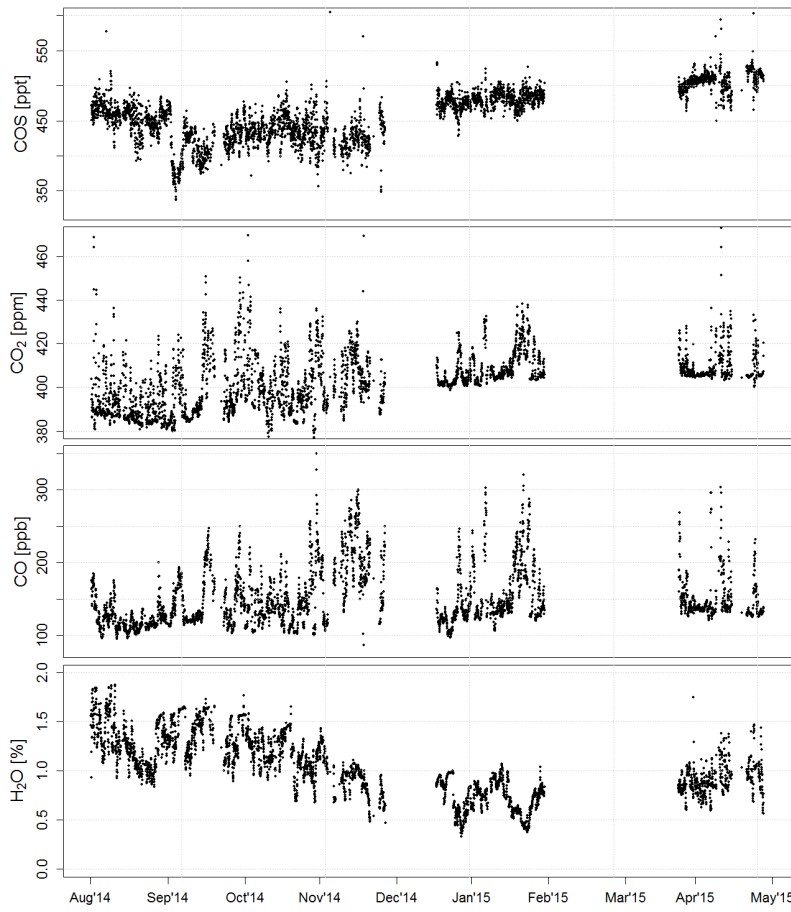

**Figure 10** Hourly averaged measurements of COS, $CO_2$, CO and $H_2O$ in ambient air at the 60 m level of the Lutjewad measurement tower as measured by the QCLS. Data are shown with corrections as described in the text.



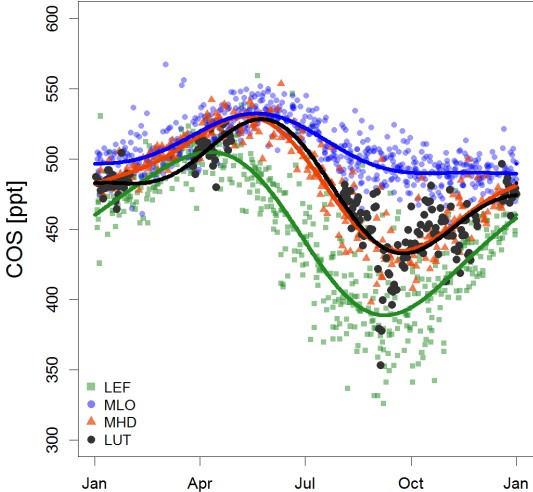

**Figure 11** COS seasonal cycle of 4 sites: Wisconsin, United States (LEF), Mauna Loa, United States (MLO), Mace Head, Ireland (MHD) and Lutjewad, the Netherlands (LUT) where the data of the latter site is presented in this study. COS mole fractions for the LEF, MLO and MHD sites were measured from flask samples at a GC-MS by NOAA/ESRL (Montzka et al., 2007). The NOAA/ESRL data are shown as flask pair means from individual sampling events. All NOAA measurements are plotted as function of time of the year and cover a period between 2000 to 2015 for LEF, MLO and MHD. *In situ* COS measurements with the QCLS at the Lutjewad site during 2014-2015 are shown as daily averages (black). A two-harmonic seasonal cycle is fit through the data.



**Figure 12** Minute averaged measurements of COS, $CO_2$, CO and $H_2O$ of the Lutjewad measurement tower between January 7 and April 29, 2015 as measured by the QCLS (blue), compared with measurements from a CRDS (orange) for $CO_2$, CO and $H_2O$, and with dry air flask sample QCLS measurements (orange) for COS.

