# Peer review of "Continuous and high precision atmospheric concentration measurements of COS, CO₂, CO and H₂O using a quantum cascade laser spectrometer (QCLS)"

_Atmospheric Measurement Techniques, 2016_

## Referee Comment (RC2) · Anonymous Referee #2 · 5 Apr 2016

Critique Summary

In general, I believe that this is a well-constructed experiment of the characterization of the Aerodyne QCLS system for measurement of ambient concentrations of COS, CO2, and CO. I believe that it is appropriate for publication in Atmos. Meas. Tech. after substantial revisions. My major criticisms are related the presentation of uncertainties and how they relate to different types of analyses that might be performed with the data set collected at Lutjewad. Presumably, the authors have written this manuscript with future analyses in mind that focus on interpreting ambient COS, CO, and CO2

data. The particular type of analysis will determine which types of uncertainty should be considered. I suggest revising the manuscript in a way that makes it clear in the Introduction what types of analyses are intended (or are possible) for this data set, while also discussing the theoretical requirements for uncertainty in these cases. The authors hint at two ways in which the data might be used – interpreting vertical gradients in COS at Lutjewad and interpreting horizontal gradients between a network of stations reporting COS (as in Figure 11). The latter use of data requires a much more stringent evaluation of accuracy since it requires the data from all stations to be on the same scale. The former, has a seemingly less stringent need for accuracy, but instead is more dependent on short-term precision in order to quantify what might be small differences in concentration from one height to the next. This is my impression, anyways, and I suggest that the authors explore this idea and report on whether there should be two different estimates of total uncertainty depending on the type of analysis (i.e. vertical profiles vs station-to-station differences).

For this review, I first provide a general critique of the authors' presentation of errors in the QCLS measurement, including a discussion of the link between uncertainty type and the intended analysis, as suggested above. Then, I provide a list of specific comments and edits for the authors to address.

General Critique

On the evaluation of target cylinders of COS, CO, and CO2 for accuracy against the NOAA/WMO scale (Figure 9 and Sec. 3.1), I would expect that the mean bias should be lower than the uncertainty listed in Table 5 as "Transfer to calibration standards" (2.8 ppt for COS). Is this correct? Or should it be compared to the quadrature sum of "Transfer to cal standards" and "Calibration using calibration standards" (4.0 ppt for COS)? In any case, this comparison needs to be made in order to evaluate whether the uncertainties listed in Table 5 (and contributing to the total uncertainties) properly envelope the bias determined from this experiment. Using COS as an example, the mean bias is -3.3 ppt and -1.7 ppt for the two cylinders (single bias correction). The standard error of the

mean should be given, as well. Even though the mean is within 1 standard deviation of 0, with so many measurements made, the mean bias should be very well known and will likely be significantly different from 0 (the standard error will more accurately express how well the mean is known than the standard deviation). The next question becomes, is this mean bias (+/- the standard error) reflected in the uncertainties listed in Table 5. Additionally, please also give the absolute concentrations for the cylinders used in this experiment. It is relevant to know whether there is an increased bias for the target cylinders at concentrations outside of the range of the NOAA/ESRL standards. I use COS as an example here, but this comment also applies to CO and CO2.

Next, I think it is important to distinguish between short-term precision and repeatability and provide some discussion about where each measure of uncertainty should be used. As written, the authors do not sufficiently make this distinction. Table 5 lists the repeatability uncertainty as the 1-second standard deviation from the histograms in Figure 8. I would not consider this value a measure of repeatability, rather, it should be the standard deviation of the mean of the repeat measurements on the target cylinders in Figure 9. I would expect the repeatability to be somewhat higher than the short-term precision because typically there is some amount of instrumental drift. The repeatability for COS is 6.6 ppt and 4.9 ppt for the single bias correction case and 5.3 ppt for the hourly response curve case. All three measures of repeatability are somewhat higher than the short-term precision, and the total uncertainty (Table 5) should probably reflect these higher values.

When the repeatability (rather than the precision) is incorporated into the total measurement uncertainty, the total uncertainty has relevance for interpreting diurnal, synoptic, seasonal, and longer-term variability in concentrations. It is also relevant when comparing the Lutjewad station measurements to measurements at other sites and with other measurements that are referenced to the NOAA/WMO scale. Therefore, an analysis of data in Figure 11 should have a total uncertainty attached that includes repeatability rather than short-term precision. On the other hand, if the goal is to interpret differences between one height and another in the vertical sampling arrangement at Lutjewad, this is where the short-term precision is relevant. In this situation, one is looking for small differences between two (essentially) simultaneous measurements. Additionally, since this type of analysis relies on a comparison of measurements from the same instrument, I would expect that there should be less reliance on the need to transfer the measurements to the NOAA/WMO scale (and the total uncertainty should therefore be lower). Is this correct? If it has been determined that a background offset correction is more important that correcting for a variable instrument response curve, then the near-simultaneous measurements at both inlet heights will be affected by the same offset correction, and, as a result, uncertainty in that offset correction will cancel when estimating the difference between measurements.

In any case, it is important to distinguish between uncertainties that are associated with a comparison of multiple measurement sites vs those associated with inlet height differences. Additionally, the water vapor correction uncertainties may also cancel out unless there is a very strong vertical H2O gradient. While this might be challenging to determine, given the potential for sampling line artifacts, some consideration of this would be worthwhile.

Finally, I question what value there is in reporting the 1-second precision. I would think that unless there is a need to understand high frequency variability in the atmosphere (such as for eddy covariance fluxes), the precision should be reported at a more relevant dwell time. Figure 10, for example, shows hourly averages. Daily averages are used in Figure 11. The 1-second precision has a negligible contribution to uncertainty in the mean on these time scales. For the vertical profile measurements, each height has a 8 minute dwell time. Variability at time-scales less than 8 minutes, therefore, is not important for determining inlet height differences. Even at the 95th percentile of standard deviations for the entire data set in Figure 8 (13.2 ppt for COS), the precision for the 8-minute averages is still very small (likely < 1ppt). The point is, that this short-term precision determines the magnitude of inlet height differences that can be

considered significant. The authors report that differences between heights 40 m and 60 m are, on average, 0.7 ppt +/- 9.7 ppt. It seems, therefore, that the mean difference between the two heights is probably not significant, but with a 9.7 ppt standard deviation, there are likely times when the differences could be interpreted as real (with high confidence) and potentially relatable to the quantification of a COS flux. I think a short discussion of these points would add a lot of value to this manuscript.

Specific comments:

Page 2, lines 31-33: Regarding the need to constrain COS to 1.5 to 4.0 ppt to constrain $CO_2$ fluxes matching a 1 ppm change. It is my understanding that LRU (leaf-scale relative uptake) scales with the ratio of $CO_2$/COS mole fractions. In this case if $CO_2$ is $\sim$400ppm and COS is 500ppt, then a 1 ppm draw-down in $CO_2$ would scale to $1.9 - 5.0$ ppt (or $0.4 - 1\%$) in COS. This reflects just a small change from the $0.3 - 4.0$ ppt range quoted, but should be noted, nonetheless.

Page 3, 1st paragraph: Is this paragraph a justification for making vertical gradient measurements of COS in order to estimate fluxes rather than via the eddy covariance method? If so, state this explicitly. It would also be relevant to discuss a few details about flux-gradient theory and the pros and cons of estimating fluxes by this method. For example, doesn't this approach also require a flux measurement of at least one species for which you also have a vertical profile for (e.g. water vapor or $CO_2$)? I understand that this is a paper focused on instrument characterization, as is entirely appropriate for AMT, but it would be good to also provide some context for the long-term vision of this measurement set-up and how COS (and therefore $CO_2$) fluxes will ultimately be estimated.

Page 5: Around this section, I started having a hard time keeping the cylinder types straight. I think it would be useful to have a Table that lists each cylinder as either a NOAA/ESRL standard, a reference standard, or a target standard, says which cylinders were brought to the field, and gives a list of what species are characterized in each.

Also relevant for the COS cylinders would be the cylinder material (coated or uncoated) for each.

Further, the COS concentrations in the target/reference cylinders shown in Figure 7 do not match any of the concentrations listed in Table 2. Are these a separate set of cylinders? Which ones are used in Figure 9 as the target cylinders and which ones are used to derive the hourly response curves? Which ones were observed to have some drift to them?

Page 6, line 29: Do you mean to say "uncorrelated with instrument parameters, such as temperature"?

Page 8, lines 16-17 (and Table 5): Could you provide more flexible uncertainties for the water vapor correction for COS, $CO_2$, and CO, instead of at a single water vapor concentration and single analyte concentration? The most straightforward way to do this, it seems, would be to give the uncertainty on the slope. For example, COS changes by 2.9% (+/-??) per % $H_2O$. Related to this, a value of 3.5 ppt is listed in Table 5 as the water vapor correction uncertainty for a range of COS concentrations and $H_2O$ concentrations (up to 2.1%). I would expect the uncertainty to scale with $H_2O$, so if the uncertainty at 1.5% is 3.5 ppt, the uncertainty at 2.1% would be 4.9 ppt. Is this accurate? If not, then why? If so, this should be acknowledged someway in Table 5 and also reflected in the total uncertainty.

Page 11-12, Sec. 2.6: I was confused by the goals of the flask analyses in this study. I was expecting that they would be used to compare the QCLS with the GC-MS measurements at NOAA; however, the discussion here neglects to actually discuss this comparison (although it is shown in Figure 6). Instead, the discussion seems to center on how well the flask measurements via QCLS reproduce the cylinder concentrations that are estimated from in situ measurements. This reduces, essentially, to a test of whether any bias is added with the flask filling procedure or from storage effects. I can understand how this would be important to confirm ahead of a comparison of GC-MS

measurements (which requires flasks) with the QCLS; however, no such comparison is presented in the text. From Figure 6 it appears as though the QCLS flasks are slightly elevated with respect to the GC-MS measurements. Is this significant? Or is it within expectations based on uncertainties in the transfer of the NOAA scale?

Page 13, lines 4-6: Here it is stated that a few uncoated cylinders drifted at a rate of 2-3 ppt per month. Which cylinders are these? How was the drift determined? In previous sections comments have been made indicating that cylinder drift was not found to impact certain experiments. Given that COS drift in cylinders appears to be a persistent concern, I think the authors should consider adding a new section somewhere that details how the presence or absence of drift was monitored and which cylinders drifted over what time periods (and how that relates to various field and laboratory experiments).

Pages 14-15, Sec, 3.2: I suggest that the authors consider putting this section last since it is not related to characterization of the instrument performance. As written, it falls in between two sub-sections that relate to instrument characterization (Sec 3.1 Precision and Accuracy and Sec. 3.3 Measurement Comparison).

Pages 15-16, Sec. 3.3: What is the point of the flask vs in situ comparison for COS in this section? I was expecting that there would be a comparison between the QCLS and GC-MS flask measurements, given that the section describes an instrument intercomparison for CO and CO2 measurements. What value does the comparison between in situ and flasks on the same instrument add for this site? Is there a longer-term record of COS from flasks at this site that one might want to harmonize with the in situ measurements going forward? Is there some other value to having both flask and in situ measurements at the same site? Is the intent to send flasks to other sites (to be measured by QCLS) for comparison to Lutjewad? Or is it simply a check on the water vapor correction? If so it should be discussed along with the laboratory experiments on the water vapor correction.

Page 16, line 6: "…against -0.02 ppm…" Should this value be -0.12 ppm?

Tables 2 and 5: It appears as though the average of the uncertainties of the 4 cylinders in Table 2 (scenario C) is how the transfer uncertainty is estimated in Table 5. If so, please state this somewhere or, perhaps, include another column in Table 2 that details the average uncertainty for each of the three scenarios. This would help clarify where some of the numbers in Table 5 come from.

Table 4: Include uncertainties on the slopes of the correction curves.

Table 5: Might it be more appropriate to rename "Calibration using calibration standards" as "Calibration of the Instrument Response"? This might clarify the meaning of this uncertainty.

Figures 3 and 4: Consider calculating the mean and standard error of each grouping of data (in between background measurements) during the water vapor experiments (in Figure 3) and deriving the linear regressing using these means, rather than the real-time data (in Figure 4). This would certainly clean up Figure 4 (for CO and COS, anyway), but it also might provide more robust fits to the dry/wet ratios.

Figure 5: This diagram added to some confusion about which cylinders were brought to the field site, since only two calibration cylinders are shown. Considering adding some representation of the actual number of cylinders to this diagram, or specifying in the caption.

Figure 6: Consider showing the mean deviation (across all 4 cylinders) for the QCLS and for the GC-MS measurements (with and without outliers included).

Figure 12: I think that the COS figure should be a separate figure, since the comparison figure on the right-hand side is not showing instrument comparability of COS, as it is for CO, CO2, and H2O. It is a bit misleading, otherwise.

---

## Editor Comment (EC1) · M. von Hobe (Editor) · 11 Apr 2016

Please also consider the following issues when revising the manuscript:

In Figure 7, the stability improvement after the temperature stabilization measures on March 25 is clearly visible. But while for CO and OCS, the data scatter nicely around the respective assigned cylinder values, the measured CO2 values are significantly (by about 13 ppm) below the assigned cylinder values. Please explain this in the revised manuscript!

In the introduction, precision is defined "as the standard deviation over a two minute

period". It would be more useful to look at precision achieved with different averaging times, because i) you have various sources of random noise with different time scales (e.g. temperature fluctuations), and ii) the time scales of the uncertainties are important in the context of different scientific applications of the data (this issue is raised by referee #2). It would be nice to construct and show an Allen plot to show how the random variability varies with time. With the amount of measurements you have made, this should be fairly easy to do.

In the same context, a one-second-precision and an overall uncertainty for one second data of 4.3 ppt and 7.1 ppt respectively are given for OCS. Looking at Figure 7, the scatter for 1-hour-averages looks larger, and it also seems peculiar that the standard deviation for 1 minute averages given in Table 3 should be larger than the standard deviation for one-second-data. Please look at this carefully and explain how you calculated the averages and corresponding standard deviations on different time scales!

Did you ever try to calibrate with higher concentration OCS standards? As far as I know, the OCS standards that can be purchased from NOAA are typically secondary standards (ambient air samples that NOAA measures and certifies). However, in your acknowledgements, you mention the preparation of gravimetric standards. Can you indicate where in your study such gravimetric standards were used? Using one or two standards with higher concentration would certainly make the response curve for OCS shown in Figure 2 even more useful than the two-point-curve.

For the H2O correction, where do you obtain quantitative information of the water broadening coefficients? If this information is not published, the experimental derivation should be described. And I find the discussion somewhat confusing, because on page 9, line 13 you state that "optimized broadening coefficients are equal to 1.0, 2.15 and 1.0 for COS, CO2 and CO". But a water vapour broadening coefficient of 1.0 would mean that the water vapour effect is the same as for air, so water vapour broadening does not play a significant role, correct? It would be nice to get a quantitative feeling how much the water vapour broadening actually affects the measured mixing ratios...

[Figure]

I understand the numbers given as °C/°C in Section 2.5 in a way that you reduced the response of your system to ambient temperature variability? Was it not possible to stabilize to a fixed temperature? This seems not so difficult...

---

## Author Comment (AC1) · 29 Jun 2016

We greatly thank the reviewers for their work. Their comments helped us to improve the structure and presentation of our manuscript considerably. We first make a few general comments regarding the major changes in the manuscript, after that we provide a point by point reply to the reviewer's comments.

Both reviewers commented on the fact that we did not present a more detailed analysis on interpretation of the COS signals (e.g. diurnal variations, land-sea differences and differences between heights). We agree that this would make an interesting study but we think it will be more appropriate to publish this in a separate manuscript. In the introduction of the revised manuscript we state more explicitly that the aim of this research is to test the performance of the instrument and in the results section we discuss the suitability of the QCLS for different potential purposes based on the uncertainty analysis results. The latter also relates to a comment of Reviewer 2 who raised the point that different types of analyses have different requirements for accuracy and precision. E.g. comparing heights at one site with the same instrumentation doesn't require high accuracy. In the revised version we discuss the required precision for the flux-gradient method in the introduction section and we added a discussion about the relevance of different uncertainty contributions for different types of analyses.

Both reviewers commented that the measurement comparison for COS in Figure 12 was misleading, as this was not an instrument comparison, while it was for CO2, CO and H2O. To solve this we moved the flask comparison between the QCLS and GC-MS from section 2.6 into the measurement comparison section. Also the different analysis types are better introduced to make the difference between the comparisons clear. Besides that, we present the flask – in situ comparison for COS (top plot of Fig. 12) in a separate figure to avoid confusion.

Furthermore, we got multiple comments regarding the timescale of the presented precisions as not only the one-second precision, but also precisions on longer timescales are relevant to reflect the influence of instrumental drift. To address this, we added a discussion in section 3.1 about two-minute precisions as obtained from Fig. 8.

In the following we will provide a point by point reply to the reviewers' comments. Our replies are written in blue, and corrections in the manuscript are written in green. References to figures, sections and pages in our reply refer to the original submitted manuscript, unless stated otherwise. In the revised manuscript changes are marked in red.

**Anonymous Referee #1**

In their study, the authors have tested a recent model of the QCL Mini Monitor spectrometer (QCLS) from Aerodyne Research Inc. for its suitability to obtain continuous and high precision atmospheric measurements of COS, CO2 and CO. They evaluated the performances of this instrument in the laboratory then in the field at the Lutjewad monitoring station in the Netherlands. The ultimate goal is to provide high-quality observations of COS needed to understand the seasonality, the long-term trend and the spatial distribution of sources and sinks of this trace gas because existing monitoring networks of CO2 and CO make use of other instruments (e.g. cavity ring-down spectroscopy (CRDS) analyzers).

**General comments.**

Although the QCLS was set up for continuous in situ measurements at different heights at the tall tower of the Lutjewad monitoring station, the authors do not demonstrate the suitability of the QCLS analyzer to perform profile measurements because data recorded at 7m height are not reported. I think that this question cannot remain open after having put much effort into assessing the performances of the instrument. If the authors consider that their data are not suitable for publication, they could instead make use of the hourly measurements of calibration standards conducted while the instrument was in the field in March-April 2015 to simulate the suitability of the QCLS analyzer to perform profile measurements (see below). The continuous COS, CO2 and CO observations from Lutjevad are not even used to determine trace gas concentrations characteristic of air masses having either dominant continental or marine influence. Examples of diurnal variations could have been provided. Is there a seasonal change in the amplitude of the diurnal variations recorded by the instrument? Instead, the authors focus on COS seasonal variations reconstructed from incomplete records and compare QCLS daily averages with flask pair means from individual daytime sampling events (NOAA/ESRL data, see Fig. 11). If COS undergoes strong diurnal variations at Lutjevad, only daytime data should be compared with NOAA/ESRL data. My overall feeling is that the potential of the QCLS analyzer is not demonstrated in the field.

We agree with the reviewer that not all the potential analysis of these data is shown (daily cycle / profile etc.). We also think that this sort of analysis would be nice to have; however, the aim of this manuscript is to evaluate and improve the performance of the instrument, and not to focus on the interpretation of the COS observations. At the same time we show the measurements obtained at the Lutjewad site, as the reviewer pointed out, to demonstrate the potential of the QCLS analyzer in the field. Accordingly, we show the measurement record and a comparison across sites, and added a discussion on the observed gradients and the suitability of the QCLS to capture these gradients. Our plan is not to present further interpretation/analysis of the observations beyond these in this manuscript, but to publish this in a separate manuscript where we can present an in-depth interpretation. In the revised version of the manuscript we state explicitly in the introduction section what the goal is:

In this manuscript we aim to evaluate and improve the performance of the instrument. [...] Furthermore, we evaluate the total uncertainty of the measurements by combining the uncertainties of scale transfer, water vapor corrections and the measurement precision. Based on the precision and accuracy that we derive from these experiments, we discuss the suitability of COS measurements on this instrument for different purposes; that is, for interpreting profile measurements and comparing concentrations across sites.

Instead of providing further interpretations of the COS observations in the field we added a discussion in section 3.1 regarding the suitability of the QCLS for different potential applications, as was also suggested by Reviewer 2.

We observed that COS concentrations between 40 and 7 m differ on average by  $1.7 \pm 5.3$  ppt during daytime and  $11.4 \pm 33$  ppt during nighttime. Given that the gradients should be larger than the uncertainties mentioned above, the daytime gradients are too small to be able to use the flux-gradient method. The suitability of the flux-gradient method is further dependent on the choice of the measurement height (gradients are larger closer to the surface), the size of the flux-gradient method is limited by the measurement uncertainty that we found here, COS profile measurements from this instrument can still be useful to derive storage fluxes, as nighttime gradients are typically larger.

Furthermore, we agree with your point that we should only compare the Lutjewad daytime data with the NOAA/ESRL data and have modified Figure 11 such that the Lutjewad data only include daytime concentrations. The data presented in the new version of Figure 11 do not differ significantly from that in the first version and therefore the main message remains the same.

Thanks to rigorous testing in the laboratory, the authors clearly demonstrate that an appropriate direct water correction for COS is not possible with the combination of the standard spectral fit and the TDLWINTEL software correction on. A potential solution to this is to modify the standard fit of the spectral bands (hardly achievable by a non-specialist) or to apply homemade water corrections with the TDLWINTEL software correction on or off. One of the correction strategies (i.e., homemade water corrections with the TDLWINTEL software correction off) was tested in the field against dry air flask sample QCLS measurements for COS. Data reported in Fig. 12 suggest that COS mole fractions are not well corrected for water interference yet. The upper panel (COS one) could benefit from having H2O displayed. The difference between the flasks and in situ measurements of COS gave  $-3.5 \pm 8.6$  ppt where the 8.6 ppt standard deviation indeed suggests large variations between the different comparisons. We did a more thorough analysis on these flask comparisons and found two errors in the flask – in situ comparison: the first was related to the transfer of scales to the flask data and the other is that we first assumed that the flask and in situ sampling have the same time delay, while in reality this delay was in the order of 10 minutes. This was corrected for in the CRDS-QCLS comparison, but not for the comparison with flasks. With these corrections included we find a difference (flask - in situ) of -9.7  $\pm$  4.6 ppt, where the flasks are larger than the in situ measurements. The difference is now more consistent between the different flasks as the standard deviation is lowered to 4.6 ppt, but the mean difference is larger. The average difference of -9.7 ppt could be caused by the uncertainty of the flask and the in situ measurements (where the latter includes the uncertainty of the water vapor correction), the sampling bias of both in situ and flask measurements, as well as the storage effect of air in the flasks. As multiple factors are influencing the comparison, we cannot simply assign the difference to an error in the water vapor correction. Furthermore, we found that after applying the water correction to the in situ data, the differences between flasks and in situ measurements does not show a correlation with  $H_2O$ , while without water correction to these is a correlation. We added the following in Section 3.2 (revised version) to discuss this:

The average difference of -9.7 ppt could be caused by the uncertainty of the flask and the in situ measurements (where the latter includes the uncertainty of the water vapor correction), the sampling bias of both in situ and flask measurements, as well as the storage effect of air in the flasks. In Table 5 we quantified the uncertainty related to the transfer of the scale to working standards and measurements, the water correction, and the measurement precision. Combining these uncertainties gives an uncertainty of 5.8 ppt for the flasks and 6.5 ppt for the in situ measurements (with the uncertainty of the water vapor correction included). The combined uncertainty of both methods is 8.7 ppt, which is only slightly smaller than the -9.7 ppt difference that we found. We were not able to test the effect of biases resulting from the sampling of flasks. We find that after applying the water correction to the in situ data, the flask – in situ difference does not show a correlation with  $H_2O$  ( $R^2 = 0.01$ ; slope = 3.9 ppt COS per % H2O), whereas without water vapor correction there is a correlation with  $H_2O$  ( $R^2 = 0.36$ ; slope = 26.3 ppt COS per %  $H_2O$ ), which demonstrates that our laboratory derived water vapor correction properly corrects for the effect of water vapor.

The authors provide evidence that the baseline measurements typically done several times per hour by former users of QCLS analyzers (background measurements are determined with high purity nitrogen) are not recommended for continuous and high precision atmospheric measurements. They developed an optimal background and calibration strategy to ensure accurate field measurements and put much effort into stabilizing the temperature of the QCLS analyzer. Unfortunately, I found little evidence in the text and the figures that improving the temperature stability by actively cooling the electronics section and putting the analyzer in an enclosed box improved the precision and accuracy of field measurements. Plots showing the temperature dependency of COS, CO2 and CO concentrations are missing. The histograms of standard deviations shown in Fig. 8 are not convincing because other parameters than temperature could have affected the noise of the instrument from August 2014 until February 2015 (instabilities of the laser for example that can be responsible for increases in SD). I noticed periods of higher stability before February 2015 (for example in September 2014) than after late March 2015 when the instrument was properly insulated.

Regarding showing data of the temperature dependence of concentrations we did look for a ppt/degree factor for the particular experiment that is described. We did not find a stable correlation between temperature changes and concentration drift for the test in section 2.2.3,

likely because the changes only reach up to 0.06 °C in 30 minutes. From later field work we did find that concentrations drift together with temperature on the order of 100 ppt (COS) per degree change in the electronics temperature. We added this information in Section 2.2:

Hourly measurements of reference standards in the field (during a later field campaign in Hyytiälä, Finland, not shown here) showed that concentrations drift on the order of 100 ppt (COS), 2 ppm (CO2) and 10 ppb (CO) per degree change of the electronics temperature in a few hours.

It is right that the temperature stability did not improve the instrument precision and we do mention this in line 29-31 of Page 12 and in the conclusion section (line 22-26, page 17). Still, further stabilization of the temperature does improve the setup in the sense that the use of cylinder measurements to correct for temperature-related drift is reduced, which we mentioned in line 19-20, page 7. In the first version of our manuscript we did mention already that also the alignment can likely affect the instrument precision. In the revised version of the manuscript we marked moments where the mirror alignment was changed, which corresponds with moments that the size of the precision changed and indicates that the mirror alignment affects the instrument precision. Besides that we marked a period where the temperature of the room temperature was characterized by rapid changes caused by an air conditioner, which also corresponds with larger standard deviations. We added the following discussion in section 3.1 about this:

For the period marked in red the room temperature was characterized by rapid changes caused by an air conditioner. This was also a period where the instrument precision was adversely affected, indicating that temperature stability influences the instrument precision. [...] What we do see is that the instrument precision changed every time that the mirror alignment was changed, indicated by blue vertical lines. During periods that the mirror alignment was not changed (January 2015 and April 2015) the instrument precision was stable. These results show that the instrument precision can largely be affected by the mirror alignment.

The QCLS measurements of COS were not compared with that of other instrumentation except when four pairs of flasks filled with dry calibration gas were sent to NOAA/ESRL headquarters and analyzed there with GC/MS. Data shown in the top panel of Fig. 12 are misleading as the flasks shown are not analyzed with a different instrument.

See also our general reply.

It is right that Fig. 12 does not show an instrument comparison for COS, that is also why we called section 3.3 "Measurement comparison" and not "instrument comparison. We decided to move the flask comparison between the QCLS and the GC-MS to the measurement comparison section (3.2 in the revised manuscript) and separated this section over three paragraphs:

1. Comparison of COS measurements by QCLS and GC-MS;

2. Comparison of flask and in situ COS measurements by QCLS;

3. Comparison of  $CO_2$ , CO and  $H_2O$  measurements by QCLS and CRDS.

The three different measurement comparisons are now introduced so that is it clear that the comparison of flask vs. in situ measurements by the QCLS is not an instrument comparison.

Moreover, we present the flask - in situ comparison for COS (top plot of Fig. 12) in a separate figure to avoid confusion.

Because this study is the first in-depth appraisal of the performances of the most recent model of the QCL Mini Monitor spectrometer (QCLS) from Aerodyne Research Inc., it should be published but after major revision.

Improvements are necessary to demonstrate that this instrument is or is not suitable to obtain continuous and high precision atmospheric measurements of COS, CO2 and CO. *See also our general reply and the corrections described in our first comment.*

Since two representatives of the Aerodyne company are in the list of co-authors it would be interesting to know whether the TDLWINTEL software will offer in the future the possibility to deselect the standard fit and use the new split fit (no more need to perform homemade water corrections). If the instrument drift between two backgrounds can be properly corrected using a reference cylinder, can the TDLWINTEL software be updated to automatically correct for drift?

Newly delivered instruments will have the software fitting parameters set as the split fit. For instruments that were delivered with the standard fit the Aerodyne representatives are willing to help setting up the split fit using remote access to the instrument software.

Online automatic correction based on a reference cylinder is not preferred, as the cylinder value at the beginning and end of the measurement period is needed to do such corrections. Measurements within the period can't be corrected yet as the direction and size of the drift is not known while the calculations are being done.

Specific comments.

Abstract.

Page 1 - line 15. Hope that the authors will be able to demonstrate the suitability of the QCLS analyzer to perform profile measurements too.

See also our general reply and the corrections described in our first comment.

Page 1 - line 27. The comparison of in situ QCLS measurements with measurements from flasks is not adequate for COS. See also page 9 - line 24, page 12 - line 7 and page 16 - line 9. Data reported in Fig. 12 suggest that COS mole fractions are not well corrected for water interference yet.

See the reply to the second general comment.

Introduction.

Page 2 - line 11. The past study of Belviso et al. (2013) relied on in situ samples analyzed on-line (not flasks) with GC/MS. I am aware of other on-line measurements of COS with GC/MS using the MEDUSA instrument (AGAGE network).

We thank the reviewer for pointing this out. We removed the Belviso et al. (2013) reference in this particular sentence and added a sentence about in situ measurements with the GC-MS technique and with a reference to Belviso et al. (2013):

Although the GC-MS technique can be used for in situ measurements (Miller et al., 2008; Belviso et al., 2013), this technique does not typically allow for high frequency measurements of 1 to 10 Hz.

Page 2 - line 30. Please clarify your calculations. How can you jump from LRU ratios (the ratio of the deposition velocities of COS and CO2) to measurement precisions of COS better than 0.3-0.8%?

It would have been clearer if the precision would be stated as 1.5-4.0 ppt COS (i.e. 0.3-0.8 %), where 0.3 is 1.5/500\*100. In fact "0.3 – 0.8 %" is no critical information so we decided to leave these numbers out to avoid confusion. Reviewer two also corrected us in the sense that the LRU scales with the CO2/COS mole fractions and that this actually translates to a COS measurement precision of 1.9 - 5.0 ppt. Taking these comments together we have corrected the sentence such that it now reads as:

If we were to infer the gross fluxes from chamber measurements with  $CO_2$  measurements better than 1 ppm, then, given the leaf-scale relative uptake ratio (LRU) of  $COS/CO_2$  1.5 – 4.0 (Stimler et al., 2010b; Seibt et al., 2010; Berkelhammer et al., 2014), our goal would be to have measurement precisions of COS better than 1.9 – 5.0 ppt for COS (calculated from LRU and scaling with the ratio of  $COS/CO_2$  mole fractions gives e.g. 1.5\*(500/400) = 1.9 at the ambient level of 500 ppt COS and 400 ppm  $CO_2$ ).

Page 3 - line 12. First clear indication that you are interested in profile measurements. See also page 10 – line 8, Fig. 5 and the legend of the figure. Again see page 10 – line 17. The goal was to provide high-quality observations of COS needed to understand the vertical distribution of this gas at Lutjewad but may be the authors consider that their data are not suitable for publication.

See also our general reply and the corrections described in our first comment.

**Experimental setup.**

Page 3 - line 13. Not sure that the first prototype of QCLS required little operator attention. I think that the recent one still requires operator attention and manual post processing of data.

The question if the QCLS requires much operator attention is indeed a point of discussion. We agree that the first prototype in Stimler et al. (2010a) may have required more operator attention than the analyzers that are nowadays available, but in fact it was stated in Stimler et al. (2010a) that this analyzer required little operator attention. To avoid discussion about this point we removed the part ".. and requires little operator attention (Stimler et al., 2010a)."

Stimler et al. 2010a is in fact Stimler et al. (2009). See also page 4 line 17.

We are not sure about that. The year of publication in the pdf version of the manuscript that is available at the Wiley Online Library states 2010. I suspect that it was first online in 2009 and got actually published in 2010. We stick to referring to Stimler et al. 2010a.

**Page 3 - line 17. Unfortunately not for COS.**

In fact we do compare with GC-MS measurements using flasks. Based on the comments from both reviewers we decided to better introduce the goal of the flask measurements and stress that these were also meant as a comparison between instruments. The following lines were therefore added in the introduction:

Besides in situ measurements, flask or canister measurements can be a valuable tool for providing information about ambient concentrations of COS as well. For example, flask measurements were used before when constructing an historical record from firn air (Montzka et al., 2004), during field campaigns (White et al., 2010; Blonquist et al., 2011) and for long-term monitoring (Montzka et al., 2007). [...] In addition to the experimental setup for continuous in situ measurements we developed a setup to analyze flasks, which we used to make a comparison with GC-MS measurements of flasks and to assess the laboratory derived correction for water vapor interference.

Page 3 - line 27. Please use the same terms throughout the manuscript. Are primary standard, secondary standard and target gas more appropriate terms? In general, it would be valuable to know how long was the duration of injections of standards.

We used the terms NOAA/ESRL standards (which are in fact calibrated standards against secondary standards at NOAA/ESRL) and calibration standards, which can be either reference or target standards (see line 26-30, page 4 and line 21-23, page 5 in the first version of the manuscript). To avoid confusion we changed the terminology of the standards as follows:

- "NOAA/ESRL calibration standards" for the ones that were calibrated by NOAA/ESRL.
- "working standards" for the ones used in the field (which we called "calibration standards" before) and which are calibrated against the NOAA/ESRL calibration standards in our lab. These working standards can either be reference or target standards depending on how they are used.

We hope that some confusion about the different cylinders is also taken away now we trace the cylinders throughout the manuscript (see also our reply to the comment on Table 1).

We added a sentence in the caption of Fig. 2 stating the length of the cylinder measurements.

Page 7 - lines 8-10. The statement "The frequency of reference cylinder: : ." is not sustained with data. Show plots of COS vs T for example as in Berkelhammer et al. (2014). See also the discussion about temperature stability page 10 - line 25.

See also our reply to an earlier comment; we added a sentence in Section 2.2 stating the typical concentration changes per degree that the electronics temperature changed.

Page 8 - line 13. Here you mention the existence of instabilities of the QCLS. It is important to know whether instabilities of the QCLS occurred when the instrument was deployed in the field (see Fig. 8). See also page 12 - line 31. Factors such as alignment and laser instabilities influence the precision as well.

Yes, the instabilities occurred both under laboratory and field conditions. For field conditions it is depicted in Fig. 8 during the periods when the instrument was less stable. We added a sentence in section 2.3 saying that we further discuss the instrument precision and its variation over time in section 3.1.

We will further discuss the precision of the measurements and its variation over time in Section 3.1.

**Result and discussion.**

Page 13 - line 1. How was the overall uncertainty calculated? Please clarify the method.

The overall uncertainty is calculated as the square root of the sum of the squared individual uncertainties. In the revised version of the manuscript we added this information in that particular paragraph:

The overall uncertainty is calculated as the quadrature sum (square root of the sum of squares) of the individual uncertainties.

**Conclusions.**

This is a nice summary but what are the main conclusions of the study? Is the instrument suitable for tower profile measurements? What perspectives can you draw?

See also our general reply. In the revised version we added a discussion about the suitability of the QCLS to do different sort of analysis (differences across sites and interpreting profile measurements). Also in the conclusion section we added a sentence about this:

We discussed the relevance of different uncertainty contributions for different types of analyses (e.g. differences across sites and interpreting profile measurements). Our setup with the QCLS provides sufficient accuracy and precision to detect gradients larger than 5.2 ppt for COS, 0.11 ppm for  $CO_2$  and 1.4 ppb for CO. For the current setup in Lutjewad, the daytime gradients were too small compared to the measurement uncertainty to be able to use the flux-gradient method.

Table 1. Three cylinders are mentioned in the text but results of two cylinders are reported in Table 1. I suggest to trace the cylinders throughout the whole manuscript.

This was also raised by Reviewer 2 and is a good suggestion. We added the following sentence in section 2.2.2.

To trace the three working standards that we used in the field we refer to these with numbers #1, #2 and #3 throughout the manuscript. Other working standards that were used in this manuscript but not in the field we refer to as "A", "B", "C" and "D".

We made this consistent throughout the manuscript: Table 1, 2 and Fig. 5, 6, 8, 9 (numbers referring to the revised manuscript).

Table 5. It is impossible to reconstruct the overall uncertainty from data gathered in Table 5.

The overall uncertainty is calculated as the square root of the sum of the squared individual uncertainties:

$$\sqrt{2.1^2 + 2.8^2 + 2.8^2 + 3.5^2 + 4.3^2} = 7.1$$

In the revised version:

$$\sqrt{2.1^2 + 2.8^2 + 2.8^2 + 2.9^2 + 4.3^2} = 6.9$$

Figure 1. Mention in the legend the existence of a small water band at about 2050.4 cm-1. *We thank the reviewer for this suggestion, we have added it in the figure caption:*

A small water band at 2050.5 cm-1 can interfere with COS at 2050.4 cm-1 and can affect the COS correction for water vapor without a split fit at 2050.45 cm-1 (Sect. 2.3).

Figure 2. Explain why response curves for CO2 and CO were corrected for drift using a reference cylinder. How important was the drift and how many response curves were corrected. The residuals were averaged isn't it?

The concentrations drifted on the order of 8 ppt for COS, 3 ppm for CO2 and 5 ppb for CO in these calibration experiments. The response curves shown in Fig. 2 are based on single experiments and the consistency of the response curves when the experiment was repeated is discussed in section 2.2.1. The residuals are indeed shown as averages. In the revised version we added error bars to show the standard deviation of the measurements. We added the following in the legend of Fig 2 to indicate the size of the drift that was corrected for with the reference cylinder:

The drift was on the order of 8 ppt for COS, 3 ppm for  $CO_2$  and 5 ppb for CO and would have given biased results if this was not corrected for.

Figure 7. It is possible to simulate a nocturnal gradient of COS and CO2 from the assigned cylinder values corresponding to the green and orange samples. During the day the atmosphere is rather well mixed so COS is high and CO2 is low (green dots). During the night the atmosphere is stratified. Assuming that the ecosystem removes COS from the atmosphere and respires CO2, COS will decrease (- 41.2 ppt) and CO2 will increase (+ 9.2 ppm, orange dots). It is difficult to visualize the COS gradient because the signal is very noisy. I suggest to zoom in and check the difference between a series of consecutive orange and green dots. Is the difference about 40 ppt? You can also calculate daily averages and check again that the difference approaches 40 ppt. For CO2 it looks fine.

We do not understand what the reviewer meant to say here. No ambient concentrations but the cylinder measurements are shown in Fig. 7, which do not represent atmospheric mixing or ecosystem fluxes of COS and CO2. If the reviewer meant to say that the difference in concentrations between the different cylinders should be constant, that is true. In fact, that is presented in Fig. 9 where one cylinder is used to correct the other cylinders. If any drift would occur in one of the cylinders then it would be visible in these plots.

Figure 8. I think you should compare what is comparable. The leaky valve was removed January 7, 2015 so my suggestion is to redraw the transparent dark histograms using data collected after this date.

The leaky valve has affected the absolute value of the measurements, as we saw in Fig. 7, however, there is no potential influence on the standard deviation of the measurements. We therefore did not change the histograms in Fig. 8.

**Figure 9. It seems that the grey dots (upper left panel) have turned to black in the upper right panel.**

Actually the right panel only shows data from one cylinder, so there is only one color. In the left panel we calibrate the COS data with a single bias correction. This correction requires one reference cylinder and two of the three cylinders are corrected with this, which are shown as grey and black. In the right panel we calibrate the COS data based on response curves that were determined hourly from two of the three cylinders, and the one remaining cylinder that is corrected with these response curves is shown in black. This is mentioned in the caption of Figure 9 ("For the fixed response curve measurements of two target cylinders are shown, for the hourly response curve only one is shown, as two of the three cylinders were needed to determine the response curve."). In the revised version we repeat this in the main text.

Figure 10. This is a poorly informative figure. I suggest to focus on diurnal variations and land-ocean differences rather than on seasonal variations.

Figure 11. This is also a poorly informative figure because the new dataset is incomplete (see also general comments above).

See also our general reply and our reply to the first general comment, there we explained that the aim of this manuscript is to study the performance of the instrument, and not to interpret the COS signals. Even though the dataset does not cover a full year, Figure 11 (13 in the revised version) showed us that the QCLS measurements capture the seasonal variation well and that the measurements are comparable with those at other coastal sites at similar latitudes. With this figure we demonstrated the potential of the QCLS analyzer in the field and we can say that the accuracy and precision of the analyzer is suitable for long-term atmospheric monitoring. For these reasons we stick to only show Fig. 11 and 12 (12 and 13 in the revised version). We did add a discussion in section 3.1 about the suitability of the QCLS for different potential purposes, including that of interpreting profile measurements.

Figure 12. The upper left panel should be removed and the upper right panel should be presented after figure 4. I think it is more interesting to display the temporal difference between CRDS and QCLS data, the standard deviation of the difference and to look for temporal trends than to plot one against the other and calculate a slope. Ordinates are missing. See also our general reply. We decided to move the flask comparison between the QCLS and the GC-MS to the measurement comparison section (3.2 in the revised manuscript) and separated this section in three paragraphs, as presented in a previous reply. The three different measurement comparisons are now introduced so that is it clear that the flask vs in situ comparison at the QCLS is not an instrument comparison.

We agree that it is worth showing the temporal difference between the CRDS and QCLS, also between the QCLS in situ and flask measurements. We replaced the slope curves for the difference plots, which also include the frequency distribution of the differences plotted as a histogram.

**Anonymous Referee #2**

**Critique Summary**

In general, I believe that this is a well-constructed experiment of the characterization of the Aerodyne QCLS system for measurement of ambient concentrations of COS, CO2, and CO. I believe that it is appropriate for publication in Atmos. Meas. Tech. after substantial revisions. My major criticisms are related the presentation of uncertainties and how they relate to different types of analyses that might be performed with the data set collected at Lutjewad. Presumably, the authors have written this manuscript with future analyses in mind that focus on interpreting ambient COS, CO, and CO2 data. The particular type of analysis will determine which types of uncertainty should be considered. I suggest revising the manuscript in a way that makes it clear in the Introduction what types of analyses are intended (or are possible) for this data set, while also discussing the theoretical requirements for uncertainty in these cases. The authors hint at two ways in which the data might be used - interpreting vertical gradients in COS at Lutjewad and interpreting horizontal gradients between a network of stations reporting COS (as in Figure 11). The latter use of data requires a much more stringent evaluation of accuracy since it requires the data from all stations to be on the same scale. The former, has a seemingly less stringent need for accuracy, but instead is more dependent on short-term precision in order to quantify what might be small differences in concentration from one height to the next. This is my impression, anyways, and I suggest that the authors explore this idea and report on whether there should be two different estimates of total uncertainty depending on the type of analysis (i.e. vertical profiles vs station-to-station differences).

For this review, I first provide a general critique of the authors' presentation of errors in the QCLS measurement, including a discussion of the link between uncertainty type and the intended analysis, as suggested above. Then, I provide a list of specific comments and edits for the authors to address.

**General Critique**

On the evaluation of target cylinders of COS, CO, and CO2 for accuracy against the NOAA/WMO scale (Figure 9 and Sec. 3.1), I would expect that the mean bias should be lower than the uncertainty listed in Table 5 as "Transfer to calibration standards" (2.8 ppt for COS). Is this correct? Or should it be compared to the quadrature sum of "Transfer to cal standards" and "Calibration using calibration standards" (4.0 ppt for COS)? In any case, this comparison needs to be made in order to evaluate whether the uncertainties listed in Table 5 (and contributing to the total uncertainties) properly envelope the bias determined from this experiment.

Using COS as an example, the mean bias is -3.3 ppt and -1.7 ppt for the two cylinders (single bias correction). The standard error of the mean should be given, as well. Even though the mean is within 1 standard deviation of 0, with so many measurements made, the mean bias should be very well known and will likely be significantly different from 0 (the standard error will more accurately express how well the mean is known than the standard deviation). The

next question becomes, is this mean bias (+/- the standard error) reflected in the uncertainties listed in Table 5. Additionally, please also give the absolute concentrations for the cylinders used in this experiment. It is relevant to know whether there is an increased bias for the target cylinders at concentrations outside of the range of the NOAA/ESRL standards. I use COS as an example here, but this comment also applies to CO and CO2.

Yes, the mean bias must be compared with the quadrature sum of "Transfer to calibration standards" and "Calibration using calibration standards" (because the target measurements are treated as real ambient air samples), which indeed makes 4.0 ppt for COS, 0.17 ppm for  $CO_2$  and 2.4 ppb for CO. We thank the reviewer for the suggestion to add the standard error, we took this into account in the comparison. We added the following in the discussion of Fig. 8 in Section 3.1.

After the corrections are applied, the mean offset of the measurements is within  $3.3 \pm 0.2$  ppt for COS,  $0.05 \pm 0.003$  ppm for CO2 and  $1.7 \pm 0.1$  ppb for CO over the period of 35 days. The standard errors indicate that the mean offset is significantly different from 0. Still, the offsets are within the expected uncertainty based on the relevant uncertainties listed in Table 5, namely the transfer to calibration standards and transfer to field measurements of which the quadrature sum is 4.0 ppt COS, 0.17 ppm CO2 and 2.4 ppb CO. The fact that the mean offsets are within the expected uncertainties indicates that these uncertainties listed in Table 5 properly envelope the uncertainties of the field measurements.

Next, I think it is important to distinguish between short-term precision and repeatability and provide some discussion about where each measure of uncertainty should be used. As written, the authors do not sufficiently make this distinction. Table 5 lists the repeatability uncertainty as the 1-second standard deviation from the histograms in Figure 8. I would not consider this value a measure of repeatability, rather, it should be the standard deviation of the mean of the repeat measurements on the target cylinders in Figure 9. I would expect the repeatability to be somewhat higher than the short-term precision because typically there is some amount of instrumental drift. The repeatability for COS is 6.6 ppt and 4.9 ppt for the single bias correction case and 5.3 ppt for the hourly response curve case. All three measures of repeatability are somewhat higher than the short-term precision, and the total uncertainty (Table 5) should probably reflect these higher values.

**See also our general reply.**

The uncertainties presented in Table 5 are the one-second uncertainties, and we would like to keep this consistent. We do agree that the precision on time scales longer than 1 second is also relevant as it reflects the influence of instrument drift. The repeatability also depends on the calibration strategies, e.g. very frequent calibrations would be able to remove drifts on the time scale of the calibration period; however, the use of the calibration gas is not affordable for the field experiments. We would like to stick to one-second uncertainties in Table 5 for consistency, and changed the name "measurement repeatability" into "measurement precision". Still, we agree that it is relevant to discuss precisions on longer timescales; therefore we added a discussion about this in section 3.1:

Furthermore, in Table 5 and Fig. 7 we present one-second precisions, but also precisions on longer timescales are relevant to reflect the influence of instrumental

drift. Typically, precision depends on the length of the averaging period, where it can either decrease for Gaussian noise or increase for instrument drift. We will see in Fig. 8 (discussed later in this section) that for two-minute averaged target measurements the standard deviation varies between 4.9 and 6.6 ppt. This reflects the typically higher values for precision on longer timescales than one second. We provide the numbers for COS here, but the same holds for CO2 and CO.

Note that the mean offset in the upper right plot of Fig. 8 is changed from -3.4 to -1.7 in the new version of the manuscript. A wrong cylinder value was used in the calculations for this particular plot.

When the repeatability (rather than the precision) is incorporated into the total measurement uncertainty, the total uncertainty has relevance for interpreting diurnal, synoptic, seasonal, and longer-term variability in concentrations. It is also relevant when comparing the Lutjewad station measurements to measurements at other sites and with other measurements that are referenced to the NOAA/WMO scale. Therefore, an analysis of data in Figure 11 should have a total uncertainty attached that includes repeatability rather than short-term precision. On the other hand, if the goal is to interpret differences between one height and another in the vertical sampling arrangement at Lutjewad, this is where the short-term precision is relevant. In this situation, one is looking for small differences between two (essentially) simultaneous measurements. Additionally, since this type of analysis relies on a comparison of measurements from the same instrument, I would expect that there should be less reliance on the need to transfer the measurements to the NOAA/WMO scale (and the total uncertainty should therefore be lower). Is this correct? If it has been determined that a background offset correction is more important that correcting for a variable instrument response curve, then the near-simultaneous measurements at both inlet heights will be affected by the same offset correction, and, as a result, uncertainty in that offset correction will cancel when estimating the difference between measurements.

In any case, it is important to distinguish between uncertainties that are associated with a comparison of multiple measurement sites vs those associated with inlet height differences.

The reviewer made a very good point here, and we are thankful for that. We agree that we should discuss the different purposes of the QCLS measurements and the different uncertainties associated with that. Indeed, the measurements do not need to be transferred to the NOAA/WMO scales if only the gradients were to be analysed based on the measurements from different heights. In fact, another reason to transfer our measurements to the NOAA/ESRL scale is that COS in cylinders tends to drift over time, and it is therefore important to achieve good accuracy by linking the measurements to a well-maintained scale. We made a number of changes (additions) in the manuscript to cover this. In the introduction section we introduce the different requirements for different types of analyses:

Besides the difference in requirements for precision between different experimental setups, the type of analyses intended for a dataset also determines the requirements for precision and accuracy of the measurements. If the intention is to compare atmospheric concentrations across sites, then accuracy is important because data from different sites must be on consistent scales. On the other hand, short-term precision is more important than accuracy when differences between heights are to be interpreted (e.g., as in estimation of fluxes from profile measurements). Following the K-parameterization formulation of the flux-gradient method (e.g. Meredith et al., 2014),  $F = -K \frac{\Delta C}{\Delta z} \rho$ , the precision required to capture the concentration differences between heights ( $\Delta C$ ) mostly depends on the size of the fluxes F, the height difference  $\Delta z$ , the turbulence conditions, which is represented by the eddy diffusivity K, and to a lesser extent by the molar density of air p. To be able to capture COS fluxes of e.g. 10 pmol  $m^{-2} s^{-1}$  over a height difference of 20 meters the measurement precision of COS should be better than 0.5 ppt under high turbulent conditions ( $K = 10 \text{ m}^{-2} \text{ s}^{-1}$ ) and 4.8 ppt under low turbulent conditions ( $K = 1 m^{-2} s^{-1}$ ). [...].Furthermore, we evaluate the total uncertainty of the measurements by combining the uncertainties of scale transfer, water vapor corrections and the measurement precision. Based on the precision and accuracy that we derive from these experiments, we discuss the suitability of COS measurements on this instrument for different purposes; that is, for interpreting profile measurements and comparing concentrations across sites.

The following discussion is added in section 3.1 (precision and accuracy):

We have now presented the total uncertainty for long-term concentration monitoring at an atmospheric station, but not all uncertainties are relevant for every type of analyses. If the data are to be compared across different sites, then the data should be on the same scale and the accuracy of the measurements is important. However, if data from the same site and the same instrumentation are compared for example to do flux-gradient analysis, then there is a less stringent need for accuracy and the shortterm precision is more important. If the uncertainties related to the transfer of the scales are not taken into account then the total uncertainty would be 5.2 ppt for COS, 0.11 ppm for  $CO_2$  and 1.4 ppb for CO. This includes both the uncertainty of the water vapor correction and the one-second measurement precision. The uncertainty of the water vapor correction cannot simply be ignored as gradients in  $H_2O$  can exist as well. We observed that COS concentrations between 40 and 7 m differ on average by  $1.7 \pm 5.3$  ppt during daytime and  $11.4 \pm 33$  ppt during nighttime. Given that the gradients should be larger than the uncertainties mentioned above, the daytime gradients are too small to be able to use the flux-gradient method. The suitability of the flux-gradient method is further dependent on the choice of the measurement height (gradients are larger closer to the surface), the size of the fluxes at a given site and the turbulence conditions. Even though the use of the flux-gradient method is limited by the measurement uncertainty that we found here, COS profile measurements from this instrument can still be useful to derive storage fluxes, as nighttime gradients are typically larger.

Additionally, the water vapor correction uncertainties may also cancel out unless there is a very strong vertical H2O gradient. While this might be challenging to determine, given the potential for sampling line artifacts, some consideration of this would be worthwhile.

We agree that when  $H_2O$  would be similar with height the water correction uncertainty would cancel out, but unfortunately we cannot simply make this assumption. We added a sentence in section 3.1 to mention this:

If the uncertainties related to the transfer of the scales are not taken into account then the total uncertainty would be 5.2 ppt for COS, 0.11 ppm for CO2 and 1.4 ppb for CO. This includes both the uncertainty of the water vapor correction and the one-second measurement precision. The uncertainty of the water vapor correction cannot simply be ignored as gradients in  $H_2O$  can exist as well.

Finally, I question what value there is in reporting the 1-second precision. I would think that unless there is a need to understand high frequency variability in the atmosphere (such as for eddy covariance fluxes), the precision should be reported at a more relevant dwell time. Figure 10, for example, shows hourly averages. Daily averages are used in Figure 11. The 1second precision has a negligible contribution to uncertainty in the mean on these time scales. For the vertical profile measurements, each height has a 8 minute dwell time. Variability at time-scales less than 8 minutes, therefore, is not important for determining inlet height differences. Even at the 95th percentile of standard deviations for the entire data set in Figure 8 (13.2 ppt for COS), the precision for the 8-minute averages is still very small (likely < 1ppt). The point is, that this short-term precision determines the magnitude of inlet height differences that can be considered significant. The authors report that differences between heights 40 m and 60 m are, on average, 0.7 ppt +/- 9.7 ppt. It seems, therefore, that the mean difference between the two heights is probably not significant, but with a 9.7 ppt standard deviation, there are likely times when the differences could be interpreted as real (with high confidence) and potentially relatable to the quantification of a COS flux. I think a short discussion of these points would add a lot of value to this manuscript.

See also our general reply and our reply to a comment earlier about short-term precision and repeatability. We agree that also longer-term precisions are relevant in this study, because, indeed, we also discuss hourly and daily averaged measurements in Fig. 10 and 11. As we said in an earlier reply we discussed precisions on timescales longer than one second in section 3.1.

We also agree with the point that the differences between 40 and 60 m can indeed be significant and that this is reflected in the 9.7 ppt standard deviation. Based on this comment we checked if these moments with significant differences affect the flask – in situ comparison and removed one flask – in situ pair from the comparison. For the moments that the flask and in situ measurements are compared the standard deviations are smaller. We added some further discussion about the 40-60 differences:

Although the mean difference of 0.7 ppt is not significant, the 9.7 ppt standard deviation points to actual differences between the 40 and 60 m concentrations. For the moments where the flask and in situ measurements are compared the difference is -4.2  $\pm$  3.4 ppt. We neglected one flask – in situ pair for which the 40 and 60 m difference was larger than 10 ppt. For the remaining data we do not expect a large bias associated with including ongoing results from the 40 m height in the averages.

Specific comments:

Page 2, lines 31-33: Regarding the need to constrain COS to 1.5 to 4.0 ppt to constrain CO2 fluxes matching a 1 ppm change. It is my understanding that LRU (leaf-scale relative uptake) scales with the ratio of CO2/COS mole fractions. In this case if CO2 is \_400ppm and COS is 500ppt, then a 1 ppm draw-down in CO2 would scale to 1.9 - 5.0 ppt (or 0.4 - 1%) in COS. This reflects just a small change from the 0.3 - 4.0 ppt range quoted, but should be noted, nonetheless.

This is a very good point and we thank the reviewer for correcting this detail! Reviewer 1 also asked for a clarification of the calculation from LRU to a COS measurement precision of 0.3-0.8 %. Taking this together we have corrected the sentence such that it now reads as:

If we were to infer the gross fluxes from chamber measurements with  $CO_2$  measurements better than 1 ppm, then, given the leaf-scale relative uptake ratio of  $COS/CO_2$  1.5 – 4.0 (Stimler et al., 2010b; Seibt et al., 2010; Berkelhammer et al., 2014), our goal would be to have measurement precisions of COS better than 1.9 – 5.0 ppt for COS (calculated from LRU and scaling with the ratio of COS/CO2 mole fractions gives e.g. 1.5\*(500/400) = 1.9 at the ambient level of 500 ppt COS and 400 ppm  $CO_2$ ).

Page 3, 1st paragraph: Is this paragraph a justification for making vertical gradient measurements of COS in order to estimate fluxes rather than via the eddy covariance method? If so, state this explicitly. It would also be relevant to discuss a few details about flux-gradient theory and the pros and cons of estimating fluxes by this method. For example, doesn't this approach also require a flux measurement of at least one species for which you also have a vertical profile for (e.g. water vapor or CO2)? I understand that this is a paper focused on instrument characterization, as is entirely appropriate for AMT, but it would be good to also provide some context for the longterm vision of this measurement set-up and how COS (and therefore CO2) fluxes will ultimately be estimated.

A potential type of analysis for this dataset is indeed to use the flux-gradient method. In the revised version we stated more explicitly that the goal is to evaluate and improve the performance of the QCLS and to explore the suitability of the instrument for the different potential analyses. In line with this we also mention the flux-gradient method in the introduction and we discuss the necessary precision of COS to be useful for gradient measurements:

On the other hand, short-term precision is more important than accuracy when differences between heights are to be interpreted (e.g., as in estimation of fluxes from profile measurements). Following the K-parameterization formulation of the flux-gradient method (e.g. Meredith et al., 2014),  $F = -K \frac{\Delta C}{\Delta z} \rho$ , the precision required to capture the concentration differences between heights ( $\Delta C$ ) mostly depends on the size of the fluxes F, the height difference  $\Delta z$ , the turbulence conditions, which is represented by the eddy diffusivity K, and to a lesser extent by the molar density of air  $\rho$ . To be able to capture COS fluxes of e.g. 10 pmol m-2 s-1 over a height difference of 20 meters the measurement precision of COS should be better than 0.5 ppt under high turbulent conditions ( $K = 10 \text{ m}^{-2} \text{ s}^{-1}$ ) and 4.8 ppt under low turbulent conditions ( $K = 1 \text{ m}^{-2} \text{ s}^{-1}$ ).

Page 5: Around this section, I started having a hard time keeping the cylinder types straight. I think it would be useful to have a Table that lists each cylinder as either a NOAA/ESRL standard, a reference standard, or a target standard, says which cylinders were brought to the field, and gives a list of what species are characterized in each. Also relevant for the COS cylinders would be the cylinder material (coated or uncoated) for each.

Further, the COS concentrations in the target/reference cylinders shown in Figure 7 do not match any of the concentrations listed in Table 2. Are these a separate set of cylinders? Which ones are used in Figure 9 as the target cylinders and which ones are used to derive the hourly response curves? Which ones were observed to have some drift to them?

This is a good point and this was also raised by Reviewer 1. We believe that an extra table is not necessary, but in the revised version we do track the cylinders throughout the manuscript. We added the following sentence in section 2.2.2.

To trace the three working standards that we used in the field we refer to these with numbers #1, #2 and #3 throughout the manuscript. Other working standards that were used in this manuscript but not in the field we refer to as "A", "B", "C" and "D".

We made this consistent throughout the manuscript: in Table 1, 2 and Fig. 5, 6, 8, 9 (numbers referring to the revised manuscript).

This should make clear that the cylinders used for Table 2 are a separate set of cylinders (A, B, C, D) than those in the field (#1, #2, #3). The cylinder numbers are now also given in Fig 8 (Fig 9 in original version of manuscript). Also when we discuss the cylinder drift in section 3.1 we state specifically which cylinder numbers it is associated with.

**Page 6, line 29: Do you mean to say "uncorrelated with instrument parameters, such as temperature"?**

Yes, that is indeed what we mean. The sentence is corrected as suggested.

Page 8, lines 16-17 (and Table 5): Could you provide more flexible uncertainties for the water vapor correction for COS, CO2, and CO, instead of at a single water vapor concentration and single analyte concentration? The most straightforward way to do this, it seems, would be to give the uncertainty on the slope. For example, COS changes by 2.9% (+/-??) per % H2O. Related to this, a value of 3.5 ppt is listed in Table 5 as the water vapor correction uncertainty for a range of COS concentrations and H2O concentrations (up to 2.1%). I would expect the uncertainty to scale with H2O, so if the uncertainty at 1.5% is 3.5 ppt, the uncertainty at 2.1% would be 4.9 ppt. Is this accurate? If not, then why? If so, this should be acknowledged someway in Table 5 and also reflected in the total uncertainty.

If we give an uncertainty on the slope this would indeed mean that that the uncertainty scales with  $H_2O$  and the absolute concentrations. However, it does not appear from Fig. 4 that the uncertainty scales with  $H_2O$ . Therefore, we are not in favour of giving an uncertainty on the slope, but rather on the concentration itself, where the uncertainty is determined based on the standard deviation of the residuals of the fit. In the revised version we do discuss in Section 3.1 that the uncertainty scale with the COS,  $CO_2$  and CO concentrations.

The overall uncertainty is calculated as the quadrature sum (square root of the sum of squares) of the individual uncertainties. The overall uncertainty that is presented in Table 5 characterizes the typical measurement uncertainty but may vary depending on the conditions. For example, the measurement precision varies, as we discussed in the previous paragraph, and the water vapor correction uncertainty scales with the COS,  $CO_2$  and CO concentrations.

Page 11-12, Sec. 2.6: I was confused by the goals of the flask analyses in this study. I was expecting that they would be used to compare the QCLS with the GC-MS measurements at NOAA; however, the discussion here neglects to actually discuss this comparison (although it is shown in Figure 6). Instead, the discussion seems to center on how well the flask measurements via QCLS reproduce the cylinder concentrations that are estimated from in situ measurements. This reduces, essentially, to a test of whether any bias is added with the flask filling procedure or from storage effects. I can understand how this would be important to confirm ahead of a comparison of GC-MS measurements (which requires flasks) with the QCLS; however, no such comparison is presented in the text. From Figure 6 it appears as though the QCLS flasks are slightly elevated with respect to the GC-MS measurements. Is this significant? Or is it within expectations based on uncertainties in the transfer of the NOAA scale?

We agree that this flask analysis in fact shows an instrument comparison, but that we did not discuss it as such, while it adds valuable information. We added the following discussion to cover this:

The flask pairs have a mean deviation from the assigned cylinder values of +3.0 ppt for the QCLS (1.7 ppt when excluding the drifting flask pair) and -3.0 ppt for the GC-MS measurements. Although these deviations are within the measurement uncertainty, the GC-MS measurements are consistently lower than those on the QCLS, with an average difference of 5.1 ppt (excluding the drifting flask pair). We were not able to find an explanation for this bias.

Page 13, lines 4-6: Here it is stated that a few uncoated cylinders drifted at a rate of 2-3 ppt per month. Which cylinders are these? How was the drift determined? In previous sections comments have been made indicating that cylinder drift was not found to impact certain experiments. Given that COS drift in cylinders appears to be a persistent concern, I think the authors should consider adding a new section somewhere that details how the presence or absence of drift was monitored and which cylinders drifted over what time periods (and how that relates to various field and laboratory experiments).

The drift was observed for two of the three cylinders used in Lutjewad, but after the measurement period. Similar drift (2-3 ppt per month) was observed at NOAA/ESRL as well. During the measurement period in Lutjewad we did not see a significant change in concentrations (determined from calibrations) between July 2014 and March 2015. However, we did see drift in two of the three cylinders when they were re-calibrated in November 2015. For the third cylinder we did not observe drift of the same size. We added a more detailed discussion of this in section 3.1:

Additional to the uncertainties presented in Table 5 we have observed that COS can decrease over time in uncoated aluminum cylinders. First, we did not find indications that our calibration standards drifted during field measurements at the Lutjewad station; calibrations in July 2014 and March 2015 showed a decrease of only 2.2 and 1.1 ppt for cylinder #1 and #2 over this 8-month period, which is well within the measurement uncertainty. However, a re-calibration in November 2015 showed a decrease of 18.2 and 24.1 ppt in these cylinders, a decrease with a rate of 2.3 and 3 ppt per month, while cylinder #3 only changed by 1.9 ppt. Cylinder #3 is not different from #1 and #2 (they are all uncoated aluminum cylinders), but cylinder #1 and #2 were stored with a pressure of 25 and 40 bar, which is lower than 130 bar for cylinder #3. We cannot confirm if the drift in the cylinders is related with the cylinder pressure; however, it is the only difference that we were able to find.

Pages 14-15, Sec, 3.2: I suggest that the authors consider putting this section last since it is not related to characterization of the instrument performance. As written, it falls in between two sub-sections that relate to instrument characterization (Sec 3.1 Precision and Accuracy and Sec. 3.3 Measurement Comparison).

We agree that section 3.2 (in the first version) fits better after the measurement comparisons of section 3.3 (in the first version). In the revised version we have moved the measurement comparison section in front of the section that shows the measurement record.

Pages 15-16, Sec. 3.3: What is the point of the flask vs in situ comparison for COS in this section? I was expecting that there would be a comparison between the QCLS and GC-MS flask measurements, given that the section describes an instrument intercomparison for CO and CO2 measurements. What value does the comparison between in situ and flasks on the same instrument add for this site? Is there a longer-term record of COS from flasks at this site that one might want to harmonize with the in situ measurements going forward? Is there some other value to having both flask and in situ measurements at the same site? Is the intent to send flasks to other sites (to be measured by QCLS) for comparison to Lutjewad? Or is it simply a check on the water vapor correction? If so it should be discussed along with the laboratory experiments on the water vapor correction.

We agree that the reason for doing flask measurements should be better introduced. In fact, multiple reasons that you give are true. For this paper the main reason to do flask measurements is to check the water vapor correction and the comparison with the GC-MS, which indeed needs to be better presented as you also indicated in your comment on Page 11-12, Sec. 2.6. However, testing the QCLS for its suitability to do flask measurements is interesting for the COS community in general, as flask measurements can be useful both during short field campaigns or for longer-term monitoring where in situ measurements from the QCLS are not possible/available. At this moment we monitor COS from the Lutjewad site with flask measurements and use the QCLS for in situ measurements during field campaigns elsewhere. For the revised version we have reconsidered the placement of the flask measurements in the manuscript and have put it into section 3.3 (in the first version, 3.2 in the revised version). We also introduce the flask measurements already in the Introduction section:

Besides in situ measurements, flask or canister measurements can be a valuable tool for providing information about ambient concentrations of COS as well. For example, flask measurements were used before when constructing an historical record from firn air (Montzka et al., 2004), during field campaigns (White et al., 2010; Blonquist et al., 2011) and for long-term monitoring (Montzka et al., 2007). [...] In addition to the experimental setup for continuous in situ measurements we developed a setup to analyze flasks, which we used to make a comparison with GC-MS measurements of flasks and to assess the laboratory derived correction for water vapor interference.

Page 16, line 6: ": : : against -0.02 ppm: : :" Should this value be -0.12 ppm?

No, this value is really -0.02 ppm. This number only compares the QCLS and CRDS measurements from January 7 until January 29, because for that period the TDLwintel correction was turned on and in that period the data needed an extra water correction on top of it (because the TDLwintel correction only is not sufficient). The -0.12 ppm value covers the whole period from January 7 until the end of April. The comparison that we make in lines 4-7 of page 16 is between having the linear water correction curve on top of the TDLwintel water correction and not having this extra water correction curve. With these numbers we want to indicate that the extra correction curve is needed to properly correct for the water effect.

Tables 2 and 5: It appears as though the average of the uncertainties of the 4 cylinders in Table 2 (scenario C) is how the transfer uncertainty is estimated in Table 5. If so, please state this somewhere or, perhaps, include another column in Table 2 that details the average uncertainty for each of the three scenarios. This would help clarify where some of the numbers in Table 5 come from.

Good point, this is indeed not mentioned. We made a footnote of this in Table 5: \*Average uncertainty over 4 cylinders in Table 2 (method 3).

Table 4: Include uncertainties on the slopes of the correction curves. *See also our reply to Page 8, lines 16-17.*

Table 5: Might it be more appropriate to rename "Calibration using calibration standards" as "Calibration of the Instrument Response"? This might clarify the meaning of this uncertainty.

Renaming this may indeed help clarifying the meaning of it. In fact, this uncertainty is the calibration of field measurements as done with the calibration standards. In the revised version we renamed "Calibration using calibration standards" to "Measurement calibration".

Figures 3 and 4: Consider calculating the mean and standard error of each grouping of data (in between background measurements) during the water vapor experiments (in Figure 3) and deriving the linear regressing using these means, rather than the real-time data (in Figure 4). This would certainly clean up Figure 4 (for CO and COS, anyway), but it also might provide more robust fits to the dry/wet ratios.

We like this suggestion and followed on this with something similar; in the revised version we derive the linear regression from averages over every 0.1 %  $H_2O$  range. By doing this we made sure that the regression is not influenced by the fact that there are more data points towards lower  $H_2O$  concentrations. The slopes derived with this regression are almost identical to those in the first version, but the uncertainty is reduced (e.g. from 3.5 to 2.9 ppt for COS). The slightly different slopes are now used consistently throughout the manuscript.

Figure 5: This diagram added to some confusion about which cylinders were brought to the field site, since only two calibration cylinders are shown. Considering adding some representation of the actual number of cylinders to this diagram, or specifying in the caption.

We agree that this helps clarifying some cylinder logistics and have modified the figure such that now 3 calibration cylinders and 1 nitrogen cylinder are shown in Figure 5. We also added cylinder numbers #1, #2, #3 to repeat that those cylinders were the ones brought to the field.

Figure 6: Consider showing the mean deviation (across all 4 cylinders) for the QCLS and for the GC-MS measurements (with and without outliers included).

Good suggestion, in the revised manuscript we discuss this in section 3.2 (which is where these flask measurements are discussed in the new version).

The flask pairs have a mean deviation from the assigned cylinder values of +3.0 ppt for the QCLS (1.7 ppt when excluding the drifting flask pair) and -3.0 ppt for the GC-MS measurements. Although these deviations are within the measurement uncertainty, the GC-MS measurements are consistently lower than those on the QCLS, with an average difference of 5.1 ppt (excluding the drifting flask pair). We were not able to find an explanation for this bias.

Figure 12: I think that the COS figure should be a separate figure, since the comparison figure on the right-hand side is not showing instrument comparability of COS, as it is for CO, CO2, and H2O. It is a bit misleading, otherwise.

This point was also raised by Reviewer 1 and we have made the COS comparison a separate figure. Also we added the instrument comparison between QCLS and GC-MS with flasks in section 3.2 (in the revised version).

**M. von Hobe (Editor)**

Please also consider the following issues when revising the manuscript:

In Figure 7, the stability improvement after the temperature stabilization measures on March 25 is clearly visible. But while for CO and OCS, the data scatter nicely around the respective assigned cylinder values, the measured CO2 values are significantly (by about 13 ppm) below the assigned cylinder values. Please explain this in the revised manuscript!

This is indeed the case, and it is because the data in Figure 7 are raw (but hourly averaged) data and are not yet corrected with the instrument response. After this correction the data are close to the assigned cylinder values, e.g. for  $CO_2$  the values are around 405 (orange), 398 (green) and 384 ppm (blue), which becomes 420, 412 and 398 ppm after correcting with the response curve (for  $CO_2$ : 1.038x-0.839, see fig 2). These corrected values are close to the assigned cylinder values. In the revised manuscript we made a note of this in the figure caption:

The data shown here are uncorrected data and are not calibrated with a response curve. The concentrations are therefore not necessarily close to the assigned cylinder values.

In the introduction, precision is defined "as the standard deviation over a two minute period". It would be more useful to look at precision achieved with different averaging times, because i) you have various sources of random noise with different time scales (e.g. temperature fluctuations), and ii) the time scales of the uncertainties are important in the context of different scientific applications of the data (this issue is raised by referee #2). It would be nice to construct and show an Allen plot to show how the random variability varies with time. With the amount of measurements you have made, this should be fairly easy to do.

We agree that also longer-term precisions are relevant in this study, because, indeed, we also discuss hourly and daily averaged measurements in Fig. 10 and 11. We agree that an Allan variance plot can nicely show the capability of averaging the data to reduce noise level; however, we find it less useful in this manuscript where we deal with changing instrument precision over time, which is not represented in Allen plots. In the revised manuscript we did add a discussion in section 3.1 on the uncertainty on timescales of 2 minutes, as obtained from Fig 8 (in the new version).

Furthermore, in Table 5 and Fig. 7 we present one-second precisions, but also precisions on longer timescales are relevant to reflect the influence of instrumental drift. Typically, precision depends on the length of the averaging period, where it can either decrease for Gaussian noise or increase for instrument drift. We will see in Fig. 8 (discussed later in this section) that for two-minute averaged target measurements the standard deviation varies between 4.9 and 6.6 ppt. This reflects the typically higher values for precision on longer timescales than one second. We provide the numbers for COS here, but the same holds for  $CO_2$  and CO.

In the same context, a one-second-precision and an overall uncertainty for one second data of 4.3 ppt and 7.1 ppt respectively are given for OCS. Looking at Figure 7, the scatter for 1-hour-averages looks larger, and it also seems peculiar that the standard deviation for 1 minute averages given in Table 3 should be larger than the standard deviation for one-second-data. Please look at this carefully and explain how you calculated the averages and corresponding standard deviations on different time scales!

The scatter in Fig. 7 is indeed larger than the reported precision and total uncertainty in Table 5. The data from Fig. 7 can be compared with the precision in Table 5 after corrections for instrument drift are applied, which is what we see and evaluate in Fig. 9.

The precision on timescales longer than one second is typically larger due to instrument drift. That explains why the one-minute average standard deviations in Table 3 are larger than the one-second standard deviations in Table 5. In the revised version we discuss the precision on longer time scales in more detail in section 3.1, see also our reply to the previous comment.

Did you ever try to calibrate with higher concentration OCS standards? As far as I know, the OCS standards that can be purchased from NOAA are typically secondary standards (ambient air samples that NOAA measures and certifies). However, in your acknowledgements, you mention the preparation of gravimetric standards. Can you indicate where in your study such gravimetric standards were used? Using one or two standards with higher concentration would certainly make the response curve for OCS shown in Figure 2 even more useful than the two-point-curve.

We had only two NOAA calibrated standards for COS in our lab, as these were the two that we purchased from NOAA. The gravimetric standards are only used at NOAA to calibrate these standards. It would indeed have been nice to calibrate the analyzer with higher (and lower) COS standards, but unfortunately we didn't have the tools available.

For the H2O correction, where do you obtain quantitative information of the water broadening coefficients? If this information is not published, the experimental derivation should be described.

Unfortunately this work is not published. The information given in lines 1-5 on page 9 regarding the water broadening coefficients estimates from the manufacturer does not add critical information. For these reasons we decided to remove this information from the manuscript.

And I find the discussion somewhat confusing, because on page 9, line 13 you state that "optimized broadening coefficients are equal to 1.0, 2.15 and 1.0 for COS, CO2 and CO". But a water vapour broadening coefficient of 1.0 would mean that the water vapour effect is the same as for air, so water vapour broadening does not play a significant role, correct? It would be nice to get a quantitative feeling how much the water vapour broadening actually affects the measured mixing ratios...

It is correct that a water broadening coefficient of 1.0 actually has no effect. The uncertainties of the water broadening coefficients actually tell more about the significance of these numbers. For the revised manuscript we discuss the uncertainties of the water broadening coefficients in Section 2.3:

As the optimization of the broadening coefficients depends on the curves in Fig. 4 (most optimal is when the slopes are equal to zero), the uncertainty of the water correction also entails an uncertainty in the broadening coefficients. We find that the uncertainties of the broadening coefficients are equal to 0.5 (COS), 0.03 (CO2) and 0.7 (CO). This means that varying the broadening coefficient of COS from 1.0 to 1.5 only changes the COS concentration by 2.9 ppt (at a concentration of 450 ppt COS). The large uncertainties of the COS and CO broadening coefficients therefore indicate that these coefficients have relatively little effect on the concentrations.

I understand the numbers given as  $_C/_C$  in Section 2.5 in a way that you reduced the response of your system to ambient temperature variability? Was it not possible to stabilize to a fixed temperature? This seems not so difficult...

Yes, it is correct that we tried to reduce the response to ambient temperature variability and indeed it would have been most optimal to stabilize the temperature, either the lab room where the system is placed, or the temperature of the system itself. Temperature-stable room conditions were very difficult to achieve, especially for field conditions where it is difficult to arrange temperature-controlled conditions. We did efforts to control the temperature in the room where the instrument was placed but only ended up with quick temperature changes from an air conditioning. The mean temperature was then constant at 25 °C, but with fluctuations of  $\pm$  0.5 degrees, which only resulted in larger instrument drift. We then tried to stabilize the instrument itself by extending the cooling loop and adding isolation. We did not manage to further control the temperature of the system itself with the given time and availability of the instrument before later field work started. However, there are new ideas in the community to further stabilize the instrument. We encourage these developments and may follow these in the future.

**Continuous and high precision atmospheric concentration measurements of COS, CO2, CO and H2O using a quantum cascade laser spectrometer (QCLS)**

Linda M.J. Kooijmans1, Nelly A.M. Uitslag1, Mark S. Zahniser2, David D. Nelson2, Stephen. A. Montzka3, Huilin Chen1,4

1Centre for Isotope Research (CIO), University of Groningen, Groningen, The Netherlands
 2Aerodyne Research Inc., MA, USA
 3NOAA Earth System Research Laboratory, Boulder, Colorado, USA
 4Cooperative Institute for Research in Environmental Sciences (CIRES), University of Colorado, Boulder, CO, USA

10

5

Correspondence to: Huilin Chen (Huilin.Chen@rug.nl)

**Abstract.** Carbonyl sulfide (COS) has been suggested as a useful tracer for Gross Primary Production as it is taken up by plants in a similar way as  $CO_2$ . To explore and verify the application of this novel tracer, it is highly desired to develop the ability to perform continuous and high precision in situ atmospheric measurements of COS and  $CO_2$ . In this study we have

- 15 tested a quantum cascade laser spectrometer (QCLS) for its suitability to obtain accurate and high precision measurements of COS and CO2. The instrument is capable of simultaneously measuring COS, CO2, CO, and H2O after including a weak CO absorption line in the extended wavelength range. An optimal background and calibration strategy was developed based on laboratory tests to ensure accurate field measurements. We have derived water vapor correction factors based on a set of laboratory experiments, and found that for COS the interference associated with a water absorption line can dominate over
- 20 the effect of dilution. This interference can be solved mathematically by fitting the COS spectral line separately from the H2O spectral line. Furthermore, we improved the temperature stability of the QCLS by isolating it in an enclosed box and actively cooling its electronics with the same thermoelectric chiller used to cool the laser. The QCLS was deployed at the Lutjewad atmospheric monitoring station (60 m, 6°21'E, 53°24'N, 1 m a.s.l.) in the Netherlands from July 2014 to April 2015. The QCLS measurements of independent working standards while deployed in the field showed a mean difference
- with the assigned cylinder value within 3.3 ppt COS, 0.05 ppm for CO2 and 1.7 ppb for CO over a period of 35 days. The different contributions to uncertainty in measurements of COS, CO2 and CO were summarized and the overall uncertainty was determined to be 6.9 ppt for COS, 0.21 ppm for CO2 and 3.4 ppb for CO for one second data. A comparison of in situ QCLS measurements with those from concurrently filled flasks that were subsequently measured by the QCLS showed a difference of  $-9.7 \pm 4.6$  ppt for COS. Comparison of the QCLS with a cavity ring-down spectrometer showed a difference of
- 30  $0.12 \pm 0.77$  ppm for CO2 and -0.9 ± 3.8 ppb for CO.

**1. Introduction**

Carbonyl Sulfide (COS) has been suggested as a potential tracer for photosynthetic  $CO_2$  uptake (Sandoval-Soto et al., 2005; Montzka et al., 2007; Campbell et al., 2008; Berry et al., 2013, Asaf et al., 2013), as it follows the same uptake pathway into plants through stomata as  $CO_2$ , but is not generally re-emitted by plants (Protoschill-Krebs and Kesselmeier, 1992;

- 5 Protoschill-Krebs et al., 1996; Stimler et al. 2010b). COS therefore provides a means to partition Net Ecosystem Exchange into Gross Primary Production (GPP) and Respiration. As large uncertainties in the COS budget remain, field measurements of COS and CO2 concentrations and fluxes from leaf- to ecosystem- and regional scale are required for the COS tracer method to be tested and validated (Wohlfahrt et al., 2012; Berkelhammer et al., 2014). Therefore, there is a need for high frequency and high precision measurements techniques of COS and CO2.
- 10

Several past studies on COS have relied on discrete (flask) samples analyzed with Gas Chromatographic Mass Spectrometry (GC-MS; Montzka et al., 2007; Stimler et al., 2010b; Belviso et al., 2013). For example, the global atmospheric flask sampling network described by Montzka et al. (2007) has allowed a foundation for understanding COS concentrations over annual cycles on global scale. Although the GC-MS technique can be used for *in situ* measurements (Miller et al., 2008;

15 Belviso et al., 2013), this technique does not typically allow for high frequency measurements of 1 to 10 Hz. for continuous high frequency measurements and is less appropriate for providing insight in, for example, diurnal variations of local fluxes from soil and plants. Recent developments of Quantum Cascade Laser spectrometers (QCLS) have enabled *in situ* trace gas measurements including COS. These instruments have proven to be a valuable tool for continuous high frequency measurements of COS and CO2 up to a frequency of 10 Hz (Stimler et al., 2010a; 2010b; Asaf et al., 2013; Commane et al., 2014; Derived hermone et al., 2015).

20 2013; Berkelhammer et al., 2014; Maseyk et al., 2014; Commane et al., 2015).

The required measurement precision (in this study we define precision as the standard deviation over a two minute period) for studies of exchange processes of COS and  $CO_2$  between biosphere and atmosphere depend on the concentration change that these gases undergo in any given experiment. On the regional scale, COS shows seasonal variations typically between

- $\sim$  100 to 150 ppt at continental sites in the Northern Hemisphere (NH), and 40 to 70 ppt in the Southern Hemisphere (SH) and at marine sites (Montzka et al., 2007). CO2 seasonal variations typically reach up to 15 ppm on the NH and are as low as 2 ppm at South Pole (Zhao and Zeng, 2014). For the leaf scale, COS and CO2 concentration changes can be substantially larger; for example, Berkelhammer et al. (2014) showed that during branch bag measurements COS generally decreased by 180 to 240 ppt during active photosynthesis and CO2 concentrations can easily change by 200 ppm, depending on the setup.
- 30 Besides the difference in requirements for precision between different experimental setups, the type of analyses intended for a dataset also determines the requirements for precision and accuracy of the measurements. If the intention is to compare atmospheric concentrations across sites, then accuracy is important because data from different sites must be on consistent scales. On the other hand, short-term precision is more important than accuracy when differences between heights are to be

interpreted (e.g., as in estimation of fluxes from profile measurements). Following the K-parameterization formulation of the flux-gradient method (e.g. Meredith et al., 2014),  $F = -K \frac{\Delta C}{\Delta z} \rho$ , the precision required to capture the concentration differences between heights ( $\Delta C$ ) mostly depends on the size of the fluxes *F*, the height difference  $\Delta z$ , the turbulence conditions, which is represented by the eddy diffusivity *K*, and to a lesser extent by the molar density of air  $\rho$ . To be able to

- 5 capture COS fluxes of e.g. 10 pmol m-2 s-1 over a height difference of 20 meters the measurement precision of COS should be better than 0.5 ppt under high turbulent conditions ( $K = 10 \text{ m}^{-2} \text{ s}^{-1}$ ) and 4.8 ppt under low turbulent conditions ( $K = 1 \text{ m}^{-2} \text{ s}^{-1}$ ). If we were to infer the gross fluxes from chamber measurements with CO2 measurements better than 1 ppm, then, given the leaf-scale relative (LRU) uptake ratio of COS/CO2 1.5 – 4.0 (Stimler et al., 2010b; Seibt et al., 2010; Berkelhammer et al., 2014), our goal would be to have measurement precisions of COS better than 1.9 – 5.0 ppt for COS (calculated from
- 10 LRU and scaling with the ratio of  $COS/CO_2$  mole fractions gives e.g. 1.5\*(500/400) = 1.9 at the ambient level of 500 ppt and 400 ppm  $CO_2$ ).

Measurement instruments for long-term atmospheric trace gas concentration monitoring need to meet different requirements than, for example, eddy-covariance measurements. The eddy-covariance technique requires high frequency data (>10 Hz),

- 15 which typically adversely affects the precision of the measurements compared to 1 Hz data, and requires an averaging period of about 10 to 30 minutes. In contrast to the high frequency required for eddy-covariance measurements, lower frequency measurements (1 Hz) provide useful results over extended measurement periods and enhance the precision of any individual measurement. Furthermore, measurements for long-term monitoring do not require fast response, thus it is not necessary to operate the instrument at high flow rates. As a matter of fact, low flow rates are preferred so that working standards can be
- 20 used over a long period. This reduces the additional logistics needed for calibration gases, such as filling, calibration and transportation of the standards (Xiang et al., 2014). Besides *in situ* measurements, flask or canister measurements can be a valuable tool for providing information about ambient concentrations of COS as well. For example, flask measurements were used before when constructing an historical record from firn air (Montzka et al., 2004), during field campaigns (White et al., 2010; Blonquist et al., 2011) and for long-term monitoring (Montzka et al., 2007). In this research we developed a
- 25 robust setup for high precision and long-term monitoring of ambient concentrations of COS, CO2, CO and H2O at different heights from the Lutjewad monitoring station in Groningen, The Netherlands. To this end we employed a 'QCL Mini Monitor' from Aerodyne Research Inc. (Billerica, MA, USA) that can operate autonomously and requires little operator attention. We designed an optimal strategy for 'zero' air spectral correction and calibration for accurate measurements and we assessed the correction for water vapor interference. In this manuscript we aim to evaluate and improve the performance
- 30 of the instrument. We will show the precision and accuracy of the instrument with over half a year of field-data and measurements of working standards, and we compare the measurements with other instrumentation. Furthermore, we evaluate the total uncertainty of the measurements by combining the uncertainties of scale transfer, water vapor corrections and the measurement precision. Based on the precision and accuracy that we derive from these experiments, we discuss the

suitability of COS measurements on this instrument for different purposes; that is, for interpreting profile measurements and comparing concentrations across sites. In addition to the experimental setup for continuous *in situ* measurements we developed a setup to analyze flasks, which we used to make a comparison with GC-MS measurements of flasks and to assess the laboratory derived correction for water vapor interference.

[revised manuscript text omitted]

**2.2.2 Working standards**

In this study we use working standards to represent high-pressure cylinders that are calibrated against NOAA or WMO

- 10 standards in our laboratory. The working standards are used for two purposes: (1) to correct for instrument drift during field measurements and link the measurements to the NOAA or WMO scales, in this case we refer to these standards as reference standards, and (2) to assess the accuracy of the measurements, in this case we refer to these standards as target standards. The working standards used in this study are aluminum gas cylinders (Luxfer, max. 200 bar) filled to high pressure with ambient air at the Center for Isotope Research of the University of Groningen using an oil free air compressor (RIX SA-3)
- 15 and are used in combination with two-stage regulators (Scott Specialty Gases, model 14). The working standards differ from the NOAA/ESRL calibration standards in the sense that they are uncoated. To trace the working standards that we used in the field we refer to these with numbers #1, #2 and #3 throughout the manuscript. Other working standards that were used in this manuscript but not in the field we refer to as "A", "B", "C" and "D". Using the linear regression curve that we found in Fig. 2, we determined mole fractions in these working standards by considering response curves derived from measurements
- 20 of the NOAA/ESRL calibration standards. Results of calibrations of three working standards with the QCLS are shown in Table 1 for  $CO_2$  and CO. We calibrated the same working standards with a cavity ring-down spectrometer (CRDS; Picarro Inc. model G2401-m) using the same standards linked to the WMO scale. Calibrations with the QCLS for  $CO_2$  and CO agree with calibrations from the CRDS within their uncertainties, which gives confidence in our calibration method. For COS we could not compare our calibrations with that from other instrumentation.

25

As we saw in Sect. 2.2.1, the response curve of COS is difficult to determine due to the lower instrument precision and the narrow COS concentration range of the NOAA/ESRL calibration standards. These factors introduce uncertainty in assigning values to the working standards that will be used on-site, especially for those having COS mole fractions outside of this range. Besides that, the number of available calibrated standards used to transfer the scales to other cylinders may be limited

30 in labs, especially for COS, as this gas is usually not one of the standard measured species. The question is therefore what a suitable method is to transfer the scale to the working standards for COS. To test this, we re-analyzed calibration measurements in different ways: (1) with response from the two NOAA/ESRL calibration standards and the curve forced

through a zero-point, (2) with two NOAA/ESRL calibration standards and the curve *not* forced through zero, and (3) with a single bias correction using a NOAA/ESRL calibration standard that has the concentration closest to the working standard.

[revised manuscript text omitted]

- 5 three water vapor tests, of which one of the three was done with the TDLWINTEL water correction turned off, the other two had the TDLWINTEL water correction turned on. We ran the Playback mode of the TDLWINTEL software to get the data of these water vapor experiments for the case that the correction was turned off, when the real time data were obtained with the correction on. For CO only two of the three experiments are used in the analysis because the third showed larger scatter (colored in gray), indicating instability of the OCLS. We will further discuss the precision of the measurements and its
- 10 variation over time in Sect. 3.1. The linear regression is derived from averages over every 0.1 % H2O range such that the regression is not influenced by the fact that there are more data points towards lower H2O concentrations. Figure 4 shows that the COS, CO2 and CO wet/dry ratios are all linearly dependent on H2O. When wet air mole fractions are measured (that is, the TDLWINTEL water vapor correction is turned off), these curves could act as a water vapor correction factor to obtain dry air mole fractions. The effect of H2O on the species is +3.0 % for COS, -1.46 % for CO2, and -0.8 % for CO per % H2O.
- 15 The standard deviation of the residuals of the fit translate to an uncertainty in concentrations of 2.9 ppt for COS (at a concentration of 450 ppt), 0.10 ppm for CO2 (at 400 ppm), and 1.1 ppb for CO (at 150 ppb). These uncertainties are depending on the COS, CO2, and CO concentrations. The fact that CO2 and CO show an inverse correlation with H2O indicates that these species are primarily affected by the dilution effect of water vapor. To the contrary, COS shows a positive relation with H2O. The reason for this positive correlation is that the baseline shape of the spectra is distorted by a
- small  $H_2O$  peak on the left of the larger  $H_2O$  peak to the right of the large  $CO_2$  peak (see Fig. 1). A potential solution to this would be to split the fit between the COS peak and the small  $H_2O$  peak, which we will now refer to as "split fit" (in contrast to the "standard fit"). In Fig. 4 the results with the split fit are shown in green. For  $CO_2$  and CO the green curve does not differ significantly from the blue curve, as would be expected. For COS the correlation with  $H_2O$  is now negative, with an effect of -0.8 % per %  $H_2O$ . A slope of -1 % would be expected due only to pure dilution; however, pressure broadening by
- 25 water vapor affects the mole fractions as well. Within TDLWINTEL the width of the peaks are fixed based on the line shape information from the HITRAN database which is only for air-broadening with N2 and O2. Water broadening can be greater than air broadening. To correct for the pressure broadening effect the software modifies the width of the absorption line through the so called air "broadening coefficients" which are specific for every spectral line of a certain species. The later version of TDLWINTEL can also use the water broadening effect by increasing the air broadening coefficients from
- 30 HITRAN. The residual error of the fit, which is caused by the broadened absorption line but not properly adjusted line width, affects the mole fraction and thereby the slope of the curves in Fig. 4. This can explain why the slopes differ for the different species. Besides that, the measured H2O concentration on itself has an uncertainty, which contributes to the deviation of the slope from -1 %. For the spectral lines that we use the ratio of the water broadening to air broadening coefficients were estimated by the manufacturer to be 1.5, 1.7 and 1.5 for COS, CO2 and CO respectively for application of

the standard fit. The slopes that were determined with these TDLWINTEL corrections (not shown here) were equal to +3.0 % for CO2 and +0.1 % for CO per % H2O without using the split fit. These results show that, even after correcting for water vapor interference by the TDLWINTEL software, the mole fractions are still water vapor dependent.

- 5 Using the Playback mode we tried to find the most optimal water broadening coefficients to sufficiently correct for water vapor using TDLWINTEL. We did this for the three different water vapor tests and for both the standard fit and the split fit. For the standard fit we could not find optimized broadening coefficients for COS because turning the TDLWINTEL correction on caused an opposite correction and resulted in larger deviations from the assigned cylinder value due to the effect of the H2O peak on the baseline. For the *standard* fit, the optimized broadening coefficients varied between 2.1 and
- 10 2.2 for  $CO_2$  and between 1.0 and 2.0 for CO for the different experiments. For the *split* fit we did find optimized coefficients for COS between 1.0 and 1.4 for the different experiments, for  $CO_2$  and CO the same results were found as for the standard fit. When the different experiments were combined the optimized broadening coefficients are equal to 1.0, 2.15 and 1.0 for COS,  $CO_2$  and CO respectively, where the values for  $CO_2$  and CO can be used for both the standard and split fit, and the value for COS is only suitable for the *split* fit. As the optimization of the broadening coefficients depends on the curves in
- Fig. 4 (most optimal is when the slopes are equal to zero), the uncertainty of the water correction also entails an uncertainty in the broadening coefficients. We find that the uncertainties of the broadening coefficients are equal to 0.5 (COS), 0.03 (CO2) and 0.7 (CO). This means that varying the broadening coefficient of COS from 1.0 to 1.5 only changes the COS concentration by 2.9 ppt (at a concentration of 450 ppt COS). The large uncertainties of the COS and CO broadening coefficients therefore indicate that these coefficients have relatively little effect on the concentrations. 
[revised manuscript text omitted]

- 15 suitability of the QCLS for measuring air from flasks. We developed a means to analyze flasks; this allowed a comparison between the QCLS and GC-MS measurements from the NOAA Boulder laboratory (Montzka et al., 2007). For this analysis we filled flasks from cylinders, such that, besides the instrument comparison, we were able to assess the accuracy of the flask measurements and check the methodology for assigning values to our working standards based on the NOAA/ESRL calibration standards. Four pairs of glass flasks with a volume of 2.5 L were filled to 2.5 bar with dry air from two of our
- 20 working standards and the two NOAA/ESRL calibration standards. The working standards were calibrated for COS by comparing their response on the QCLS to the NOAA/ESRL calibration standards as described in Sect. 2.2.2. The four cylinders contained COS concentrations between 447.8 and 486.6 ppt. The flask measurement setup using the QCLS is similar to that of routine tower measurements but with a few modifications. A pressure sensor was added to monitor the pressure in the flasks. Furthermore, a diaphragm pump with a shut-off valve was used to remove residual air from previous
- 25 flask measurements in the connecting tube, and to test the lines for leaks. In Fig. 5 the placement of the pressure sensor and diaphragm pump is indicated; they are separated from the field measurement setup as it is used for flask measurements only. Air from reference and target tanks was introduced before measuring the actual flask sample to calibrate the measurements to the NOAA scale. We did not use measurements of a reference gas after the flask measurement as the stability of the measurements is affected by the larger pressure difference between the flask and the reference cylinder. Two measurements
- 30 of 2 minutes each were done on each flask and the results were averaged to derive the final value. The results of these flask measurements will be shown in Sect. 3.2.

We also measured dry air flask samples to test if the water vapor correction that we determined in Sect. 2.3 sufficiently removes the effect of water vapor on calculated mole fractions. To that aim, flasks were filled to ambient pressure as part of a standard flask sampling routine at the Lutjewad station (Neubert et al., 2004). The air samples are dried with a cryogenic system prior to collecting. The flasks were stored for maximum 1.5 months before being measured. The same measurement

5 strategy as for the NOAA/ESRL comparison was used, and for these flasks two measurements of ~1.5 minutes could be done before an inlet pressure of 0.3 bar was reached. We did not observe any dependence of measured dry air mole fractions in air from the flasks with the inlet pressure below ambient. The measurement results will be shown and compared with the *in situ* QCLS measurements in Sect. 3.2.

**3. Results and Discussion**

**10 3.1 Precision and accuracy**

We assessed the measurement uncertainty and accuracy with the hourly measurements of the reference and the target gases over the period from August 2014 until April 2015. As mentioned previously, each reference and target gas was measured for two minutes. The mean value of the hourly instrument-reported and uncorrected two-minute measurements are shown in Fig. 6. In Fig. 7 the standard deviation of these measurements are shown. Figure 6 also includes the electronics temperature

- 15 of the QCLS. The cylinder measurements show that concentrations can drift substantially, i.e. COS concentrations easily vary by 50 to 100 ppt. However, on the long term, concentration changes are not correlated with temperature, which changed by 13 K throughout the year. The concentration shift on January 7, 2015 (especially visible in CO2) happened after eliminating the leaking solenoid valve that caused mixing of nitrogen into the tubing that delivers the reference and target gases. In October 2014 the span between the two cylinder measurements changed, which is again mostly visible in CO2. The
- 20 reason for this change is that the regulator pressure of one of the two cylinders was slightly changed, which affected the amount of dilution of nitrogen into the sample line. Although it is known that COS mole fractions can drift in cylinders over time we did not find indications that the mole fractions drifted within the measurement period. Later in this section we discuss the cylinder drift in more detail.
- 25 Figure 7 shows the two-minute standard deviations of the hourly reference gas measurements between August 2014 and April 2015. It is clear from Fig. 7 that the instrument precision cannot be captured with one single value due to its variation. For the period marked in red the room temperature was characterized by rapid changes caused by an air conditioner. This was also a period where the instrument precision was adversely affected, indicating that temperature stability influences the instrument precision. In the right plot of Fig. 7 the histograms of the standard deviations are shown for the periods before
- 30 and after improvement of the temperature stability with the black/gray colors corresponding to that in the left figure. The data show that for COS and  $CO_2$  the period after improving the temperature stability has an improved mean standard deviation compared to the period before the temperature improvement (from 6.6 to 4.8 ppt for COS and from 0.14 to 0.06

ppm for  $CO_2$ ). However, as there were also moments with better precision in August and September 2014, there is no consistent relation between temperature stability and instrument precision. What we do see is that the instrument precision changed every time that the mirror alignment was changed, indicated by blue vertical lines. During periods that the mirror alignment was not changed (January 2015 and April 2015) the instrument precision was stable. These results show that the

5 instrument precision can largely be affected by the mirror alignment.

The overall uncertainty of the measurements consists of uncertainties associated with the scale transfer, water vapor corrections and the measurement precision. Table 5 summarizes the different uncertainty contributions as well as the overall uncertainty for measurements of COS,  $CO_2$  and CO. The overall uncertainty is calculated as the quadrature sum (square root

- 10 of the sum of squares) of the individual uncertainties. The overall uncertainty that is presented in Table 5 characterizes the typical measurement uncertainty but may vary depending on the conditions. For example, the measurement precision varies, as we discussed in the previous paragraph, and the water vapor correction uncertainty scales with the COS,  $CO_2$  and CO concentrations. Furthermore, in Table 5 and Fig. 7 we present one-second precisions, but also precisions on longer timescales are relevant to reflect the influence of instrumental drift. Typically, precision depends on the length of the
- 15 averaging period, where it can either decrease for Gaussian noise or increase for instrument drift. We will see in Fig. 8 (discussed later in this section) that for two-minute averaged target measurements the standard deviation varies between 4.9 and 6.6 ppt. This reflects the typically higher values for precision on longer timescales than one second. We provide the numbers for COS here, but the same holds for  $CO_2$  and CO.
- 20 We have now presented the total uncertainty for long-term concentration monitoring at an atmospheric station, but not all uncertainties are relevant for every type of analyses. If the data are to be compared across different sites, then the data should be on the same scale and the accuracy of the measurements is important. However, if data from the same site and the same instrumentation are compared for example to do flux-gradient analysis, then there is a less stringent need for accuracy and the short-term precision is more important. If the uncertainties related to the transfer of the scales are not taken into account
- 25 then the total uncertainty would be 5.2 ppt for COS, 0.11 ppm for CO2 and 1.4 ppb for CO. This includes both the uncertainty of the water vapor correction and the one-second measurement precision. The uncertainty of the water vapor correction cannot simply be ignored as gradients in H2O can exist as well. We observed that COS concentrations between 40 and 7 m differ on average by  $1.7 \pm 5.3$  ppt during daytime and  $11.4 \pm 33.0$  ppt during nighttime. Given that the gradients should be larger than the uncertainties mentioned above, the daytime gradients are too small to be able to use the flux-
- 30 gradient method. The suitability of the flux-gradient method is further dependent on the choice of the measurement height (gradients are larger closer to the surface), the size of the fluxes at a given site and the turbulence conditions. Even though the use of the flux-gradient method is limited by the measurement uncertainty that we found here, COS profile measurements from this instrument can still be useful to derive storage fluxes, as nighttime gradients are typically larger.

Additional to the uncertainties presented in Table 5 we have observed that COS can decrease over time in uncoated aluminum cylinders. First, we did not find indications that our working standards drifted during field measurements at the Lutjewad station; calibrations in July 2014 and March 2015 showed a decrease of only 2.2 and 1.1 ppt for cylinder #1 and #2 over this 8-month period, which is well within the measurement uncertainty. However, a re-calibration in November 2015

- 5 showed a decrease of 18.2 and 24.1 ppt in these cylinders, a decrease with a rate of 2.3 and 3 ppt per month, while cylinder #3 only changed by 1.9 ppt. Cylinder #3 is not different from #1 and #2 (they are all uncoated aluminum cylinders), but cylinder #1 and #2 were stored with a pressure of 25 and 40 bar, which is lower than 130 bar for cylinder #3. We cannot confirm if the drift in the cylinders is related with the cylinder pressure; however, it is the only difference that we were able to find. Experience with cylinders over the past 15 years at NOAA indicates that COS in Aculife treated aluminum cylinders
- 10 is typically much more stable than untreated aluminum cylinders. A potential method to improve COS calibrations is to calibrate these using a ppb-level standard accurately diluted to a range of desired COS concentrations (LaFranchi et al., 2015), provided the ppb-level standard is less susceptible to drifting COS concentrations. Applying this method could improve the accuracy of the calibration if the COS concentrations can be accurately and precisely provided over a broad range, thereby allowing for a more accurate determination of instrument response and a calibration curve. Besides that, this
- 15 method will aid in assessing the stability of working standards.

Our COS measurements are reported on the NOAA-2004 scale, and can be compared to the observations from the global network of NOAA/ESRL (e.g. Montzka et al., 2007). The same holds for  $CO_2$  and CO on the WMO-X2007  $CO_2$  scale and WMO-X2004 CO scale. To test the accuracy of the measurements against the NOAA or WMO scale we analyzed the

- 20 measurements of one/two target standards after application of two corrections: (1) a correction factor as obtained from response curves of  $CO_2$  and CO to transfer the data to the WMO scale and (2) a bias correction using a reference standard to remove instrument drift and to calibrate the data with the NOAA scale for COS. For measurements in March and April 2015 we determined hourly response curves from measurements of two working standards. For every species we took the standards with the outer concentration values such that the response curves span a wide concentration range. To analyze the
- 25 need to determine hourly response curves, we corrected the data in two ways: (1) with a single bias correction for COS and a fixed response curve for  $CO_2$  and CO, i.e. the one that we determined in the laboratory (see Fig. 2), shown in the left plots of Fig. 8, and (2) with changing response curves determined from the hourly working standard measurements, shown in the right plots of Fig. 8. For the fixed response curve, measurements of two target standards are shown, for the hourly response curve only one is shown, as two of the three working standards were needed to determine the response curve. After the
- 30 corrections are applied, the mean offset of the measurements is within  $3.3 \pm 0.2$  ppt for COS,  $0.05 \pm 0.003$  ppm for CO2 and  $1.7 \pm 0.1$  ppb for CO over the period of 35 days. The standard errors indicate that the mean offset is significantly different from 0. Still, the offsets are within the expected uncertainty based on the relevant uncertainties listed in Table 5, namely the transfer to working standards and measurement calibration of which the quadrature sum is 4.0 ppt COS, 0.17 ppm CO2 and 2.4 ppb CO. The fact that the mean offsets are on the same order of magnitude as the expected uncertainties indicates that

these uncertainties listed in Table 5 properly envelope the uncertainties of the field measurements. The slightly larger deviation from the CO assigned value for one of the two working standards with the fixed response correction is because this standard has a higher concentration than the reference standard (237.2 against 97.8 ppb) and therefore the uncertainty due to the bias correction is larger. The other target standard, which has a concentration of 119.1 ppb, is closer to the reference

- 5 standard and only shows a deviation of -0.6 ppb. For CO2 it is visible that using the fixed response curve gives a bias up to 0.2 ppm in the target measurements, which is not visible when using the hourly response curve. This bias is an effect of the fact that the response did change in the first 10 days of the period, after which it became stable. We could not relate the changing response curve to any parameter such as temperature or pressure. We did notice that the response curve was changing only in the period after the instrument was transported. A potential reason for the change could therefore be that
- 10 the instrument still had to be stabilized after transportation. We did not find indications that the response curve for  $CO_2$  changed outside the period between March and April. Except for the fact that the hourly response curve corrects for the changing response curve for  $CO_2$ , the changing response curve does not significantly remove scatter compared to the fixed response curve for  $CO_2$  and CO; that is, the standard deviation of target measurements for the fixed response curve are not substantially lower when the hourly response curve was applied. Also the target measurements are not consistently closer to
- 15 the assigned values when the hourly response curve was applied. Furthermore, the fact that the use of hourly response curves does not give lower standard deviations for target measurements of COS compared to when the single bias correction is used, indicates again that the response curves cannot accurately be determined for this species, which we discussed in Sect. 2.2.1 and 2.2.2 as well. As we have not seen indications that the response curve for  $CO_2$  changes outside the period between March and April, we do not see the need to frequently determine the response curves with multiple standards. Moreover, if
- 20 the response would change outside of the period in March and April as well, then the effect only reaches up to 0.2 ppm for  $CO_2$ . Taking into account logistical reasons (use of cylinder gases) we suggest correcting data with a single bias correction for COS, and with a fixed response curve for  $CO_2$  and CO as determined once with NOAA/ESRL calibration standards (see Sect. 2.2.1), together with a single bias correction from a reference standard.

**3.2 Measurement comparison**

- 25 In this section we make different measurement comparisons. First, we compare COS flask measurements using the QCLS with those analyzed with the GC-MS at NOAA/ESRL. Second, we compare QCLS *in situ* and flask measurements of COS in order to test if the water vapor correction that we determined in Sect. 2.3 sufficiently removes the effect of water vapor on calculated mole fractions. Third, we compare the *in situ* CO2, CO and H2O measurements from the QCLS with those of a collocated CRDS to assess the calibration strategy and water vapor correction presented in this study. Based on the previous
- 30 sections we applied the following corrections to the *in situ* QCLS data: (1) Before March 25, 2015, the TDLWINTEL water vapor correction was applied with broadening coefficients 1.5 for COS and CO and 1.7 for CO2. On top of this correction we applied a linear water correction curve for COS,  $CO_2$  and CO as obtained with the TDLWINTEL correction turned *on*. After March 25, 2015, the TDLWINTEL correction was turned off and we applied the correction curve from Fig. 4 as obtained

with the TDLWINTEL correction *off;* (2) the calibration correction curves as obtained in Sect. 2.2.1, Fig. 2, were applied to transfer the data to the WMO scales for  $CO_2$  and CO; (3) a bias correction was applied to remove instrument drift and to calibrate the data with the NOAA scale for COS, here we used the same, and single, reference standard over the whole measurement period.

5

For the QCLS and GC-MS instrument comparison an overview of the measurements is given in Fig. 9 where the flask pair measurements are averaged and are shown as the deviation from the assigned cylinder value (see also Sect. 2.6). The precision within the flask pair is shown by the error bars. The comparison demonstrates the uncertainties associated with the transfer to the NOAA scale (see Table 5). For the first measurement at the QCLS (orange) three of the four flask pairs are within 1.5 ppt of the assigned value and one flask pair deviates by 7 ppt. This is similar to the GC-MS measurements at NOAA where one of the four flask pairs deviates further from the assigned values. However, this holds for another flask pair than at the QCLS. For the second QCLS measurement (blue) one flask pair has drifted on average by 12 ppt, where other flasks remained stable within 2.5 ppt. It is unclear why the two flasks have drifted as all flasks were filled, measured and stored in the same way; however, we have kept the pair to monitor the potential drift in the future and to find out what has caused the drift. Moreover, the consistency of the flask measurements between April 2015 and January 2016 is depending on

- 15 caused the drift. Moreover, the consistency of the flask measurements between April 2015 and January 2016 is depending on the stability of the reference standard which was used for all flask measurements. Although we observed that COS can drift in cylinders we did not find indications that the particular reference standard used for this analysis drifted over the 9 month period. The flask pairs have a mean deviation from the assigned cylinder values of +3.0 ppt for the QCLS (1.7 ppt when excluding the drifting flask pair) and -3.0 ppt for the GC-MS measurements. Although these deviations are within the
- 20 measurement uncertainty, the GC-MS measurements are consistently lower than those from the QCLS, with an average difference of 5.1 ppt (excluding the drifting flask pair). We were not able to find an explanation for this bias.

The next comparison in Fig. 10. considers the *in situ* COS measurements between March 25 and April 29, 2015 and 11 dry air flask samples measured by the same QCLS. The time delay related to the transit of air in the inlet between sampling of air

- 25 for flasks and QCLS measurements is assumed to be the same, as both sampling systems have a flow rate of 2 L min-1 and the sample tubing of the systems has the same size. The flasks are flushed for an hour before closing, but because of mixing in the flask we assume that the flask sample represents the last 15 minutes, therefore we average the *in situ* measurements over these 15 minutes. The flask samples have an inlet at 60 m height, but because the *in situ* 60 m measurements only cover the period between 9 and 15 minutes before flask closure we also include 40 m measurements to cover the last 9 minutes
- before the flask is closed. The average difference of COS mole fractions between the 40 and 60 m level is  $0.7 \pm 9.7$  ppt (40 60 m) over the measurement period in March and April. Although the mean difference of 0.7 ppt is not significant, the 9.7 ppt standard deviation points to actual differences between the 40 and 60 m concentrations. For the moments where the flask and in situ measurements are compared the difference is  $-4.2 \pm 3.4$  ppt. We neglected one flask *in situ* pair for which the 40 and 60 m difference was larger than 10 ppt. For the remaining data we do not expect a large bias associated with including

ongoing results from the 40 m height in the averages. Peaks and valleys in COS mole fractions shown in Fig. 10 are well covered by both sampling techniques (flasks and *in situ*); for example, the peak up to 620 ppt at April 11 is clearly visible in both the *in situ* and flask measurements. The average difference between the *in situ* and flask measurements (*in situ* – flask) is  $-9.7 \pm 4.6$  ppt. Besides neglecting one flask measurement where the 40 - 60 m difference was larger than 10 ppt, we

- 5 neglected one flask sample where the *in situ* COS measurement over 15 minutes showed large variation (standard deviation of 17.5 ppt where the average standard deviation of the other periods is only 4.7 ppt) and thereby introduced an error in the comparison. The average difference of -9.7 ppt could be caused by the uncertainty of the flask and the *in situ* measurements (where the latter includes the uncertainty of the water vapor correction), the sampling bias of both *in situ* and flask measurements, as well as the storage effect of air in the flasks. In Table 5 we quantified the uncertainty related to the transfer
- 10 of the scale to working standards and measurements, the water correction, and the measurement precision. Combining these uncertainties gives an uncertainty of 5.8 ppt for the flasks and 6.5 ppt for the *in situ* measurements (with the uncertainty of the water vapor correction included). The combined uncertainty of both methods is 8.7 ppt, which is only slightly smaller than the -9.7 ppt difference that we found. We were not able to test the effect of biases resulting from the sampling of flasks. We find that after applying the water correction to the *in situ* data, the flask *in situ* difference does not show a correlation
- 15 with  $H_2O$  ( $R^2 = 0.01$ ; slope = 3.9 ppt COS per %  $H_2O$ ), whereas without water vapor correction there is a correlation with  $H_2O$  ( $R^2 = 0.36$ ; slope = 26.3 ppt COS per %  $H_2O$ ), which demonstrates that our laboratory derived water vapor correction properly corrects for the effect of water vapor.

Last, we compare the minute averaged QCLS measurements for CO2, CO, and H2O with those made by a collocated CRDS in Fig. 11. The air samples were taken from the same height, but through a different inlet. The CRDS measurements of CO2, and CO were performed on humid air, and were corrected for water vapor dilution and interference effects based on a set of instrument-specific correction factors determined in the laboratory of the Center for Isotope Research before field deployment (Chen et al., 2010; 2013). The comparison was only made from January 7 onwards because before that period there was the problem of nitrogen leaking into the sample air for which we used the CRDS data to correct for the dilution.

- There are no temporal patterns visible in the difference plot on the right of Fig. 11. The mean differences (QCLS CRDS) are  $-0.12 \pm 0.77$  ppm for CO2,  $-0.9 \pm 3.8$  ppb for CO and  $-0.01 \pm 0.09$  % for H2O. For H2O there were problems with water vapor in the sample lines in March and April, therefore we only calculated the difference for H2O over the month January. A correlation of the CRDS and QCLS data of both CO2 and CO (slope = 0.98) indicates that there is no concentration dependent offset. Also no correlation was found between water vapor and the difference between CRDS and the QCLS CO2
- and CO data (CO2: slope = 0.16,  $R^2 = 0.003$ ; CO: slope = 0.48,  $R^2 = 1.9 \cdot 10^{-5}$ ). These results give confidence in the calibration strategy and water vapor correction presented in this study. For the period up to January 29, the TDLWINTEL water vapor correction was turned on, and on top of this correction we applied the linear water correction curves as obtained with the TDLWINTEL correction turned on. If we would not apply this linear correction curve on top of the TDLWINTEL correction, then the median difference between the QCLS and CRDS measurements for this period would be -0.59 ppm for

 $CO_2$  and -1.1 ppb for CO (against -0.02 ppm and -0.9 ppb for specifically this period when the linear correction curve is applied).

**3.3 Continuous COS, CO2, CO and H2O observations from Lutjewad**

The COS, CO2, CO and H2O data record obtained at the 60 m level of the Lutjewad tower is presented in Fig. 12. The data corrections used for these data were summarized in the previous section. Besides of these corrections we corrected for the dilution of nitrogen due to a leaking solenoid valve before January 7, 2015. We determined a dilution factor by comparing CO2 measurements from the QCLS and a CRDS from the same location and height, under the assumption that without dilution the two analyzers measure the same concentrations (see Sect. 3.2). The percentage of dilution was calculated for CO2 and was typically between 0.4 and 4.9 %. These dilution factors were then applied to all species.

10

The location of the Lutjewad station along the coast of the province of Groningen in the Netherlands allows the measurement of marine background air during northerly winds, and continental air during southerly winds (Van der Laan et al., 2009). Daytime CO2 concentrations are typically correlated with elevated CO concentrations, indicating the influence of local and regional fossil fuel emissions (Van der Laan et al., 2010). Even though the data do not cover a full year cycle, it can be seen that the seasonal amplitude of  $CO_2$  mole fractions is approximately 15 ppm with a minimum around the end of

- 15 can be seen that the seasonal amplitude of  $CO_2$  mole fractions is approximately 15 ppm with a minimum around the end of August. The seasonal variation of  $CO_2$  for the Lutjewad measurement station is analyzed in detail by Van der Laan-Luijkx et al. (2010) and Van Leeuwen (2015). The seasonal cycle of COS is captured well by the QCLS and the peak-to-peak amplitude is estimated to be 96 ppt based on the two-harmonic fit. Kettle et al. (2002) showed that vegetative uptake is the flux with the largest seasonal cycle on the NH, and Montzka et al. (2007) showed that the seasonal amplitude of COS
- 20 depends on the degree to which the sampled air is influenced by terrestrial ecosystems. It is therefore likely that the seasonal variation of COS observed at the Lutjewad site is influenced by vegetative uptake. In Fig. 13 we compare COS mole fractions from Lutjewad with that from three other sites as measured from flask samples with a GC-MS by NOAA/ESRL (Montzka et al., 2007). The flask samples cover data between 2000 and 2015 for Wisconsin, United States (LEF) and Mauna Loa, United States (MLO) and between 2001 and 2015 for Mace Head, Ireland (MHD). These data are an update of those
- 25 presented in Montzka et al. (2007) (data available at: ftp://ftp.cmdl.noaa.gov/hats/carbonyl\_sulfide/). The flask measurements are plotted as function of time of the year. The high altitude MLO site is less directly influenced by terrestrial ecosystems and therefore shows only small seasonal variation, in contrast to the LEF site, which is largely influenced by (forested) continental air. The Lutjewad COS mole fractions are most consistent with measurements from MHD, which can be expected since both stations are coastal sites and are located at similar latitudes. The seasonal amplitude of COS at MHD
- 30 and Lutjewad is in between that of LEF and MLO, most likely because both sites are not solely influenced by marine or continental air but by both types of air masses. The COS mole fraction has a minimum in September and October, and is a few weeks later than the minimum of the  $CO_2$  mole fraction. Montzka et al. (2007) and Blonquist et al. (2011) also observed a COS minimum later than that of  $CO_2$ . They reasoned that this difference is due to the fact that at the end of the growing

season COS mole fractions keep decreasing due to vegetative uptake without at the same time having a source of COS, whereas during this time of year respiration is beginning to offset assimilation in determining the ambient  $CO_2$  mole fractions.

**4. Conclusions**

- 5 In this study we have tested a OCLS for its suitability for making accurate and high precision measurements of COS, CO2, CO and H2O. First, the instrument response was characterized using calibration standards and to transfer raw data to the NOAA or WMO scale. Unfortunately, the range of mole fractions in the calibration standards did not allow the COS response to be accurately determined over the entire range of measured mole fractions. Based on an analysis of different calibration methods, however, we concluded that the raw measurements and working standards were best calibrated using a
- 10 single bias correction for COS. From hourly paired measurements of working standards we observed changes in the response curve for  $CO_2$  over a period of 10 days after transporting the instrument to the measurement site. However, as we have not seen indications that the response curve for CO2 changed outside this period, and also taking into account logistical reasons (use of cylinder gases) we suggest calibrating field data with a fixed response curve for CO2 and CO as determined once with calibration standards. Second, we investigated the needed frequency of background and reference measurements. Based on
- 15 laboratory tests we have shown that background measurements every six hours with reference measurements every 30 minutes (for removal of instrument drift) are sufficient to keep the standard deviation of working standard measurements within 4.6 ppt for COS, 0.08 ppm for CO2 and 0.34 ppb for CO over a period of 19 hours. We characterized the water vapor dependence of COS, CO2 and CO from laboratory experiments. Based on an assessment of the TDLWINTEL water correction we determined optimal broadening coefficients for the use of the water correction within TDLWINTEL. Besides
- 20 that, we presented an alternative water correction based on linear dependence of the wet air mole fractions with H2O concentration. Furthermore, we demonstrate that a small H2O peak close to the COS peak has caused a water vapor dependent concentration error that is larger than the direct water vapor dilution effect. This water vapor interference can be minimized by careful adjustments to the software fitting parameters and was virtually eliminated with corrections as demonstrated in Fig. 4.
- 25

The QCLS was set up for continuous in situ measurements at different heights at the tower of the Lutjewad monitoring station. Hourly target measurements were used to assess the accuracy and precision of the measurements. After application of a calibration response curve for CO2 and CO, and a single bias correction for removal of instrument drift and to calibrate the COS measurements to the NOAA scale, the target measurements showed a mean difference with the assigned cylinder

30 value within 3.3 ppt for COS, 0.05 ppm for CO2 and 1.7 ppb for CO over a period of 35 days. One-second precisions during reference gas flow were typically 4.3 ppt for COS, 0.04 ppm for CO2 and 0.9 ppb for CO, however, substantial variations in the instrument precision were observed during the 7 month field campaign. We improved the temperature stability of the

QCLS by applying an additional insulation layer that is controlled by the same thermoelectric chiller as the one used for cooling the laser and detector. However, improvement of the temperature stability of the instrument did not show a consistent relation with instrument precision. We showed that variations in the precision were largely driven by mirror alignment. The different uncertainty contributions for measurements of COS, CO2 and CO were summarized and the overall

- 5 uncertainty was determined to be 6.9 ppt for COS, 0.21 ppm for CO2 and 3.4 ppb for CO. We discussed the relevance of different uncertainty contributions for different types of analyses (e.g. differences across sites and interpreting profile measurements). Our setup with the QCLS provides sufficient accuracy and precision to detect gradients larger than 5.2 ppt for COS, 0.11 ppm for CO2 and 1.4 ppb for CO. For the current setup in Lutjewad, the daytime gradients were too small compared to the measurement uncertainty to be able to use the flux-gradient method. OCLS measurements were compared
- 10 with independent CRDS measurements for CO2 and CO, and with dry-air flask samples at the QCLS for COS, which showed a mean difference of  $-9.7 \pm 4.6$  ppt for COS,  $0.12 \pm 0.77$  ppm for CO2,  $-0.9 \pm 3.8$  ppb for CO and  $-0.01 \pm 0.09$  % for H2O. After correction for water vapor there was no correlation between the offsets and water vapor for all species. The measurement record over the 7 month period was presented and compared with NOAA/ESRL flask measurements for COS at other sites in the northern hemisphere. The peak-to-peak amplitude of COS in ambient air at the Lutjewad monitoring
- 15 station was estimated to be 97 ppt, which is comparable to other coastal sites at similar latitudes in the Northern Hemisphere.

**Acknowledgements**

We would like to thank B.A.M. Kers, M. de Vries, H.G. Jansen and H.A. Been for their help in preparing the system for installation in the field and for maintenance of the instrumentation in Lutjewad. We thank D. Paul for the valuable discussions and suggestions, H.A. Scheeren for providing the  $CO_2$  and CO calibrations of our working standards and J.J.

20 Spriensma for her help in sorting out flask samples. We also acknowledge the preparation of gravimetric standards and working standards at NOAA by B. Hall, and the technical assistance of C. Siso. Finally, we would like to thank the reviewers for their valuable comments and suggestions. This work was supported by the Dutch Sector Plan Physics.

[revised manuscript text omitted]
: (1) with response from the two NOAA/ESRL calibration standards and a zero-point, (2) with the two NOAA/ESRL calibration standards and the curve *not* forced through a zero-point, and (3) using a single bias correction. The NOAA/ESRL calibration standards have COS concentrations 447.8 and 486.6 ppt. The calibration measurement was repeated three times; results are shown as the average over the three measurement and uncertainties indicate the standard deviation over the three

5

measurements.

|    |                     | Cyl <mark>A</mark> | Cyl B       | Cyl <mark>C</mark> | Cyl D       |
|----|---------------------|--------------------|-------------|--------------------|-------------|
| 1. | Through NOAA/ESRL   | $393.0~\pm$        | $448.8~\pm$ | $473.6 \pm$        | 504.1 $\pm$ |
|    | standards and 0     | 2.2                | 4.7         | 1.5                | 0.7         |
| 2. | Through NOAA/ESRL   | $379.2~\pm$        | $445.5~\pm$ | $474.7~\pm$        | $510.8~\pm$ |
|    | standards and not 0 | 8.3                | 4.4         | 1.9                | 4.7         |
| 3. | Single bias         | $390.9~\pm$        | $445.8~\pm$ | $476.6 \pm$        | $506.6 \pm$ |
|    | correction          | 2.6                | 3.9         | 2.4                | 2.1         |

uncorrected 1 hr. corr. 30 min. corr. 15 min. corr. COS stdev. [ppt] 11.8 9.6 5.3 4.6 CO2 stdev. [ppm] 0.26 0.12 0.09 0.08 CO stdev. [ppb] 0.88 0.70 0.34 0.33

**Table 3** Standard deviation over minute averaged data of COS,  $CO_2$  and CO over the 19 hour measurement period for different correction frequencies using reference measurements.

**Table 4** Different water vapor correction strategies based on the software fitting parameters (standard or split fit) and with the TDLWINTEL correction turned on or off. In case the TDLWINTEL correction is turned on the values indicate the broadening coefficient used for the different species, and in the case it is turned off the values indicate the slope of the correction curve as determined in Fig. 4 with  $y = slope * H_2O + 1$  with  $H_2O$  in percent and y the wet/dry ratio of the gas.

|              | TDLWINTEL         | Broadening coefficient or slope |         |        |
|--------------|-------------------|---------------------------------|---------|--------|
|              | correction on/off | COS                             | $CO_2$  | СО     |
| Standard fit | on                | _*                              | 2.15    | 1.0    |
|              | off               | 0.030                           | -0.0146 | -0.008 |
| Split fit    | on                | 1.0                             | 2.15    | 1.0    |
|              | off               | -0.008                          | -0.0149 | -0.008 |

5

\*No broadening coefficient could be derived; however, we found that for a broadening coefficient of 1.5 with the standard fit the slope of the curve for COS is equal to 0.031, which can be applied as an extra correction on top of the TDLWINTEL correction.

| Uncertainty contributions                          | COS [ppt] | CO 2 [ppm] | CO [ppb] |
|----------------------------------------------------|-----------|-----------------------|----------|
| Repeatability of the NOAA or WMO scale             | 2.1       | 0.07                  | 2.0      |
| Transfer scale to working standards $(1-\sigma)^*$ | 2.8       | 0.12                  | 1.7      |
| Measurement calibration**                          | 2.8       | 0.12                  | 1.7      |
| Water vapor correction $(1-\sigma)$                | 2.9       | 0.10                  | 1.1      |
| Measurement precision (1-sec.)***                  | 4.3       | 0.04                  | 0.9      |
| Overall uncertainty                                | 6.9       | 0.21                  | 3.4      |

**Table 5** Uncertainty contributions and the overall uncertainty for measurements of COS (447.8-486.6 ppt),  $CO_2$  (354 – 426 ppm) and CO (94-384 ppb), for the range of  $H_2O$  (0-2.1 %).

**\*Average uncertainty over 4 cylinders in Table 2 (method 3).**

\*\*Using the single bias correction (see Sect. 2.2.2) it is the same as transferring the scale to the working standards.

5 \*\*\*The 50% percentile of the short-term precision of working standard measurements in March-April 2015 (Fig 7.).

Figures

---

## Author Comment (AC2) · 29 Jun 2016

Our reply to the reviewers' comments and the revised manuscript are attached.

Please also note the supplement to this comment:
http://www.atmos-meas-tech-discuss.net/amt-2016-50/amt-2016-50-AC2-supplement.pdf